Manuscript prepared for Biogeosciences
with version 2015/09/17 7.94 Copernicus papers of the LaTeX class copernicus.cls.
Date: 15 December 2018

# Biogeochemical response of the Mediterranean Sea to the transient SRES–A2 climate change scenario

Camille Richon[1,*], Jean-Claude Dutay[1], Laurent Bopp[2], Briac Le Vu[2], James C. Orr[1], Samuel Somot[3], and François Dulac[1]

[1]LSCE/IPSL, Laboratoire des Sciences du Climat et de l'Environnement, CEA-CNRS-UVSQ, Gif-sur-Yvette, France
[2]Laboratoire de Météorologie Dynamique, LMD/IPSL, Ecole Normale Supérieure - PSL Research Univ., CNRS, Sorbonne Université, Ecole Polytechnique, Paris, France
[3]CNRM, Université de Toulouse, Météo-France, CNRS, Toulouse, France
[*]Now at: Department of Earth, Ocean and Ecological Sciences, School of Environmental Sciences, University of Liverpool, Liverpool L69 3GP, UK

*Correspondence to:* Camille Richon (crichon@liverpool.ac.uk)

**Abstract.** The Mediterranean region is a climate change hot-spot. Increasing greenhouse gas emissions are projected to lead to a substantial warming of the Mediterranean Sea as well as major changes in its circulation, but the subsequent effects of such changes on marine biogeochemistry are poorly understood. Here, our aim is to investigate how climate change will affect nutrient concentra-
tions and biological productivity in the Mediterranean Sea. To do so, we perform transient simulations with the coupled high resolution model NEMOMED8/PISCES using the high–emission IPCC SRES-A2 socio-economic scenario and corresponding Atlantic, Black Sea, and riverine nutrient inputs. Our results indicate that nitrate is accumulating in the Mediterranean Sea over the 21st century, while phosphorus shows no tendency. These contrasting changes result from an unbalanced nitro-
gen–to–phosphorus input from riverine discharge and fluxes via the Strait of Gibraltar, which lead to an expansion of phosphorus–limited regions across the Mediterranean. In addition, phytoplankton net primary productivity is reduced by 10 % in the 2090s in comparison to the present state, with reductions of up to 50 % in some regions such as the Aegean Sea as a result of nutrient limitation and vertical stratification. We also perform sensitivity tests to separately study the effects of climate and
biogeochemical input changes on the future state of the Mediterranean Sea. Our results show that changes in nutrient supply from the Strait of Gibraltar and from rivers and circulation changes linked to climate change may have antagonistic or synergistic effects on nutrient concentrations and surface primary productivity. In some regions such as the Adriatic Sea, half of the biogeochemical changes simulated during the 21st century are linked with external changes in nutrient input while the other
half are linked to climate change. This study is the first to simulate future transient climate change effects on Mediterranean Sea biogeochemistry, but calls for further work to characterize effects from atmospheric deposition and to assess the various sources of uncertainty.

## 1 Introduction

The Mediterranean Sea is enclosed by three continents, and is surrounded by mountains, deserts, rivers, and industrialized cities. This evaporative basin is known as one of the most oligotrophic marine environments in the world (Béthoux et al., 1998). Because of its high anthropogenic pressure and low biological productivity, this region is likely to be highly sensitive to future climate change impacts (Giorgi, 2006; Giorgi and Lionello, 2008).

Records of the past evolution of the Mediterranean circulation show that the Mediterranean has undergone abrupt changes in its circulation patterns over ancient times. In particular, high stratification events, characterized by the preservation of organic matter in the sediments, known as sapropels, have been recorded several times through geological history. The most recent of such event occurred 10 000 years ago and lasted about 3 000 years. This accumulation of organic matter in the sediments is interpreted as the result of a strong stratification of the water column leading to suboxic deep layers (e.g. Rossignol-Strick et al., 1982; Rohling, 1991, 1994; Vadsaria et al., 2017). In more recent times, abnormal winter conditions have led to changes in deep water formation, such as the Eastern Mediterranean Transient (EMT) event that occurred during the early nineties and had chemical impacts such as an increase in the Levantine basin salinity (see Theocharis et al., 1999; Lascaratos et al., 1999; Nittis et al., 2003; Velaoras and Lascaratos, 2010; Roether et al., 2014). Also, changes in the North Ionian Gyre circulation triggered the so–called Bimodal Oscillating System (BiOS) that influenced phytoplankton bloom in the Ionian Sea by modifying water transport that led to modified nutrient distribution and altered local productivity (Civitarese et al., 2010). These events show that a semi–enclosed basin with short residence time of water (around 100 years, see Robinson et al., 2001) such as the Mediterranean Sea is highly sensitive to climate conditions and that perturbations of these conditions can modify the circulation, ultimately leading to changes in its biogeochemistry. Future climate projections with high-emission scenarios for greenhouse gases simulate warming and reduced precipitation over the Mediterranean region (Giorgi, 2006; IPCC, 2012) leading to warmer and saltier seawater (Somot et al., 2006; Adloff et al., 2015). As a result of these changes, the Mediterranean thermohaline circulation (MTHC) may significantly change with a consistent weakening in the western basin and a less certain response in the eastern basin for such high emission scenarios (Somot et al., 2006; Adloff et al., 2015). In all simulations under the A2 scenario, Adloff et al. (2015) find an increased stratification index in 2100. This increase will likely weaken the vertical mixing and may reduce nutrient supply to the upper layer of the Mediterranean, a supply that is essential for phytoplankton to bloom (d'Ortenzio and Ribera d'Alcalà, 2009; Herrmann et al., 2013; Auger et al., 2014).

Primary productivity in the ocean is influenced by its circulation and vertical mixing that brings available nutrients to phytoplankton (Harley et al., 2006). Changes in physical processes such as modification of vertical mixing can have dramatic effects on plankton community dynamics and ultimately on the productivity of the entire oceanic food web (Klein et al., 2003; Civitarese et al.,

2010). Few studies have investigated the sensitivity of the oligotrophic Mediterranean Sea to future climate change (e.g. Herrmann et al., 2014, for the northwestern Mediterranean). Lazzari et al. (2014) investigated the effects of the A1B SRES (Special Report on Emissions Scenarios) moderate climate change scenario on the Mediterranean biological productivity and plankton communities. They performed short (10–year) non–transient simulations at the beginning and the end of the 21st

century and found a decreasing trend of phytoplankton biomass in response to this climate change scenario. Macias et al. (2015) simulated a "baseline" of expected consequences of climate change alone on the Mediterranean primary productivity. Under the RCP4.5 and RCP8.5 scenarios, their simulated integrated primary productivity over the eastern Mediterranean basin increased as a result of changes in density (decreased stratification). However, those results depend on non–transient

simulations and present–day nutrient inputs. The response of the Mediterranean biogeochemistry to transient climate and biogeochemical change scenarios has not been evaluated.

  As a semi–enclosed oligotrophic basin, the Mediterranean is highly sensitive to external nutrient sources. Those sources include coastal runoff, river discharge (Ludwig et al., 2009), inputs from the Atlantic via the Strait of Gibraltar (Gómez, 2003; Huertas et al., 2012), and atmospheric deposition

(see e.g. Dulac et al., 1989; Christodoulaki et al., 2013; Gallisai et al., 2014; Guieu et al., 2014; Richon et al., 2017, 2018). Other potentially important sources of nutrient supply include direct wastewater discharge (Powley et al., 2016) and transfer by submarine groundwater (Rodellas et al., 2015). However, these two potential sources have yet to be well quantified. This study aims to assess the biogeochemical response of the Mediterranean to a "business–as–usual" climate change sce-

nario during the 21st century, while distinguishing effects from climate change, nutrient input from rivers, and changes in nutrient transport across the Strait of Gibraltar. Thus we used the high resolution coupled physical–biogeochemical model NEMOMED8/PISCES to simulate the evolution of biogeochemical tracers (e.g., nutrients, chlorophyll–a concentration, plankton biomass and primary production) under the SRES A2 climate change scenario over the 21st century (IPCC and Working

Group III, 2000). The choice of that scenario was driven by the availability of daily 3–D forcings for the biogeochemical model (physical forcings such as ocean currents, temperature and salinity, see Adloff et al., 2015). Although results from single simulation scenario must be used with caution, it is not currently feasible to make a more extensive assessment because of the computational requirements to perform large ensembles with NEMOMED8/PISCES offline model and because the 3-D

daily ocean transient forcing data are not available from simulations made with other scenarios.

  This article is organized as follows: the coupled model, forcings and the different simulations are first described. We briefly evaluate the biogeochemical model in Section 3.1 and present the evolution of the physical and biogeochemical forcings in Section 3.2. In section 3.3, we expose the temporal evolution of the main nutrients, their budgets in present and future conditions and discuss

their impact on the biogeochemistry of the Mediterranean Sea in section 4.

## 2 Methods

### 2.1 The ocean model

The ocean general circulation model used in this study is NEMO (Madec, 2008) in its regional configuration for the Mediterranean Sea (NEMOMED8 Beuvier et al., 2010). The NEMOMED8 grid has a horizontal resolution of 1/8° stretched in latitude (i.e., with a resolution from 9 km in the North to 12 km in the South of the domain). The model has 43 vertical levels with varying thicknesses (from 6 m in the surface layer to 200 m in the deepest layer). The Atlantic boundary is closed at 11°W and tracers are introduced in a buffer zone between 11°W and 6°W.

Air–sea fluxes (momentum, heat, water) and river discharge used to force NEMOMED8 are prescribed by the atmospheric Regional Climate Model ARPEGE–Climate (Déqué et al., 1994; Gibelin and Déqué, 2003) using a global and stretched grid, which has a 50–km horizontal resolution over the area of interest.

### 2.2 The SRES–A2 scenario simulation

ARPEGE–Climate is itself driven by greenhouse gases (GHG) and aerosol forcings following the observations (up to year 2000) and the SRES–A2 scenario afterwards and by SST (Sea Surface Temperature) coming from a previously run CNRM–CM coupled GCM (General Circulation Model) simulation (Royer et al., 2002). In addition, the ocean component of CNRM–CM (a low resolution NEMO version) provides the near–Atlantic conditions (3–D potential temperature and salinity) for NEMOMED8. The various forcings and the modeling chain from the GCM to the ocean regional model are described in details in Somot et al. (2006) and Adloff et al. (2015).

The NEMOMED8 simulation (ocean physics and forcings) used here corresponds to one of the simulations used by Adloff et al. (2015), i.e. their simulations labeled HIS (historical period 1961 to 1999) and A2 (A2 scenario period 2000–2099) as listed in their Table 1. This physical simulation was previously used to study impacts of climate change on Mediterranean Sea ecosystems (Jordà et al., 2012; Hattab et al., 2014; Albouy et al., 2015; Andrello et al., 2015).

The main physical changes (SST, SSS -Sea Surface Salinity-, surface circulation, deep convection and thermohaline circulation, vertical stratification, sea level) are detailed in Adloff et al. (2015). Briefly, changes in temperature and precipitation in the A2 scenario lead to increased evaporation in the basin by 2100. Freshwater input from rivers and the Black Sea decrease along with total precipitation, which in turn contributes to a substantial increase in net transport through the Strait of Gibraltar (+0.018 Sv by the end of the century).

Average sea surface temperature of the Mediterranean rises by up to 3°C by the end of the century. However, that warming is not homogeneous across the basin, with regions such as the Balearic, Aegean, Levantine and North Ionian undergoing greater warming (>3.4 °C) probably due to the addition of the atmosphere-originated quasi-homogeneous warming being combined with local changes

in surface currents. The salinity increases by 0.5 (practical salinity units) on average across the basin. These changes in hydrological characteristics generate substantial changes in the circulation and in particular the vertical mixing intensity. Under the A2 scenario, the Mediterranean basin is projected to become more stratified by 2100. Consequently, deep-water formation is generally reduced.

 ## 2.3   The biogeochemical model

Here, the physical model NEMOMED8 is coupled to the biogeochemical model PISCES (Aumont and Bopp, 2006), already used for investigations in the Mediterranean basin (Richon et al., 2017, 2018). This Monod–type model (Monod, 1958) has 24 biogeochemical compartments including 2 phytoplankton (nanophytoplankton and diatoms) and 2 zooplankton size classes (microzooplank-
ton and mesozooplankton). Phytoplankton growth is limited by the external concentration of five different nutrients: nitrate, ammonium, phosphate, silicic acid and iron. In this version of PISCES, elemental ratios of C:N:P in the organic matter are fixed to 122:16:1 following Takahashi et al. (1985). There is no explicit bacterial compartment but bacterial biomass is calculated using zoo-plankton biomass (see Aumont and Bopp, 2006, for details). Organic matter is divided in 2 forms:
dissolved organic carbon (DOC) and particulate organic carbon. The biogeochemical model was run in offline mode (see e.g. Palmieri et al., 2015): biogeochemical quantities are passive tracers, they are transported following an advection–diffusion equation using dynamical fields (velocities, mixing coefficients...) pre-calculated in a separate simulation with only the dynamical model NEMOMED8.

## 2.4   Boundary and initial physical and biogeochemical conditions

External nutrient supply for the biogeochemical model includes inputs from the Atlantic Ocean and from Mediterranean rivers. We did not include submarine groundwater discharge and direct wastew-ater discharge as there is to date no climatology for these sources. Atlantic input is prescribed from water exchange through the Strait of Gibraltar in the NEMO circulation model along with the con-centrations of biogeochemical tracers in the buffer zone. Nutrient concentrations in the buffer zone
are prescribed from a global ocean climate projection using the A2 simulation values from IPSL–CM5–LR (Dufresne et al., 2013) performed within the framework of the CMIP5 project (Taylor et al., 2012). Nutrient concentrations in the buffer–zone are relaxed to these values with a time con-stant of one month.

Nutrient inputs from rivers, including $NO_3^-$, $PO_4^{3-}$ (hereafter noted $NO_3$ and $PO_4$), and dissolved
organic carbon (DOC) are derived from Ludwig et al. (2010). Dissolved inorganic carbon (DIC) and Si are derived from Ludwig et al. (2009). For the 21st century, we use the estimations for nutrient discharge proposed by Ludwig et al. (2010) following the "Order from Strength" scenario from the Millenium Ecosystem Assessment (MEA) (Cork et al., 2005), which gives nitrate, phosphate and DOC discharge per sub–basin in 2030 and 2050. Yearly values are obtained by linear interpolation
between 2000 and 2030 and between 2030 and 2050, after which they are held constant until the end

of the simulation in 2100. Seasonal variability coming from four of the largest rivers for Mediterranean and Black Sea (Rhône, Po, Ebro and Danube) is also included. According to Ludwig et al. (2010), the future trends in nutrient discharge from the major rivers of the Mediterranean stay within the interannual variability of the past 40 years. The "Order From Strength" scenario is based on hypotheses of very little efforts made towards mitigation of climate change. Moreover, Ludwig et al. (2010) point out some substantial changes in the nutrient and water budget in specific regions. In particular, according to their scenario, the northern part of the Mediterranean has decreasing trends in nitrate discharge whereas it is increasing in the southeastern Levantine basin.

There is, to our knowledge, no transient scenario for the evolution of atmospheric deposition over the Mediterranean Sea. However, in order to evaluate the potential effects of aerosol deposition on the future Mediterranean Sea, we used deposition fields of total nitrogen deposition ($NO_3$ + $NH_4$) from the global model LMDz–INCA (Hauglustaine et al., 2014) and phosphate deposition from natural dust modeled with the regional model ALADIN–Climat (Nabat et al., 2015a) respectively (see Richon et al., 2017, and references therein for the description and evaluation of the atmospheric models). The atmospheric deposition fields represent present–day aerosol deposition fluxes (1997–2012 and 1980–2012 for total nitrogen and dust deposition respectively) that are repeated over the 1980–2099 simulation period.

Initial nutrient concentrations in the Mediterranean come from the SeaDataNet database (Schaap and Lowry, 2010) and initial nutrient concentrations in the buffer zone are prescribed from the World Ocean Atlas (WOA) (Locarnini et al., 2006). Salinity and temperature are initialized from the MEDATLAS II climatology of Fichaut et al. (2003).

All simulations began from a restart of a historical run that started in January 1965 following a spin–up of more than 115 years made with a loop over the 1966-1981 period for the physical forcings and the river nutrient discharge.

## 2.5 Simulation set–up

All simulations were made for 120 years. The control run CTRL was made with forcing conditions corresponding to the 1966–1981 period looped over the simulation period. This period was chosen in order to avoid including in the CTRL the years with excessive warming such as the 1980s and 1990s (see Figure 12 in Adloff et al., 2015, for the surface temperature evolution from 1960). The scenario simulation is referred to as HIS/A2 as in Adloff et al. (2015). The HIS refers to the historical period (in our case between 1980 and 1999), while A2 is the name of the 2000–2099 scenario simulation.

In order to separately quantify the effects of climate and biogeochemical changes, we made 2 additional control simulations: (1) CTRL_R with climatic and Atlantic conditions corresponding to present–day conditions (no scenario for climate change or nutrient fluxes through the Strait of Gibraltar) and river nutrient discharge following the scenario evolution, and (2) CTRL_RG with present-day climatic conditions, but river nutrient discharge and Atlantic buffer–zone concentrations

following the scenario conditions. Table 1 describes the different simulations. The different effects are computed by taking differences between simulations. The effects of nutrient input from exchange across the Strait of Gibraltar and from riverine discharge independent of climate effect are derived

by taking CTRL_R minus CTRL and CTRL_RG minus CTRL_R. Similarly, to derive the effects of climate change and nutrient input change on nutrient budgets, we use the difference between HIS/A2 and CTRL. To derive the effects of climate change only, we calculate the difference between HIS/A2 and CTRL_RG.

We made two supplementary simulations, one with total nitrogen deposition (HIS/A2_N) and another with total nitrogen and natural dust deposition (HIS/A2_NALADIN). These simulations include climate change and nutrient fluxes from rivers and via the Strait of Gibraltar that follow the scenario conditions. The results from these simulations should be considered as exploratory. Nonetheless, they provide insight into the potential effects of future aerosol deposition.

## 3 Results

### 3.1 Evaluation of the NEMOMED8/PISCES model

NEMOMED8 has already been used in a number of regional Mediterranean Sea modeling studies either in hindcast mode (Beuvier et al., 2010; Herrmann et al., 2010; Sevault et al., 2014; Soto-Navarro et al., 2015; Dunić et al., 2016) or scenario mode (Adloff et al., 2015). It produces the main

characteristics of the Mediterranean Sea circulation. Evaluation of the HIS simulation provided in Adloff et al. (2015) shows that the main physical characteristics of the Mediterranean are produced, in spite of a too cold upper layer (1°C colder than observations) and too little stratification in comparison to observations. In particular, the HIS simulation matches closely the observed thermohaline circulation in the Adriatic and Ionian basins (see Adloff et al., 2015).

The regional NEMOMED physical model has already been coupled to the biogeochemical model PISCES on a 1/12° grid horizontal resolution (Palmieri et al., 2015; Richon et al., 2017, 2018), but no future climate simulation has yet been performed. As a first study coupling NEMOMED8 with PISCES, we compared the main biogeochemical features of our simulations with available data. Figure 1 shows the surface average chlorophyll concentrations in the top 10 meters of the

CTRL and HIS/A2 simulations, and from satellite estimations from MyOcean Dataset (http://marine. copernicus.eu/services-portfolio/access-to-products/?option=com_csw&view=details&product_id= OCEANCOLOUR_MED_CHL_L4_NRT_OBSERVATIONS_009_041). Whenever we refer to chlorophyll, we always mean chlorophyll–a (hereafter noted chl-a). The model correctly reproduces the main high–chl-a regions such as the Gulf of Lion and coastal areas. However, Figure 1 shows an

underestimation of about 50 % of the surface chl-a concentrations by the model in these productive areas. The west–to–east gradient of productivity is also reproduced by the model with values that

agree with satellite estimates (approximately 50 % decrease in average chl-a concentration between the western and the eastern basin in the satellite data and 30 to 50 % in the model). Moreover, this Figure shows that chl-a produced by the CTRL is stable over time. The model fails, however, to

reproduce the observed chl-a–rich areas in the Gulf of Gabes and at the mouth of the Nile. This discrepancy is probably linked with insufficient simulated nutrient discharge from coastal runoff in these regions. Moreover, several studies (see e.g. Claustre et al., 2002; Morel and Gentili, 2009) show that satellite estimates have a systematic positive bias in the coastal regions because of the presence of particulate matter (for instance, sediments). The general bias observed in the Mediter-

ranean is linked with organic matter and the presence of dust particles in seawater which cause light back scattering. Figure 2 provides an evaluation of the average normalized chl-a surface concentration evolution over the entire basin for the period 1997–2005. This Figure shows that the normalized chl-a surface concentration in the model is close to the estimates provided by the SeaWiFs satellite data (Bosc et al., 2004). Even though the interannual variability of the model is 50 % smaller than in

the satellite product, the model captures the increase in chl-a concentration between 2002 and 2005 (approximately 15 % of increase in the model and 30 % in the satellite data). The evaluation of the model against two datasets shows that the model yields satisfying estimates of surface chl-a.

The vertical distribution of nitrate and phosphate over a section crossing the Mediterranean from East to West as well as chl-a and nutrient concentration profiles at the DYFAMED station are shown

in Appendix (Figures A1 and A3). These figures show that the model produces some seasonal and interannual variability of the nutricline depth and intensity. However, the nutricline depth and DCM depth are consistently overestimated by the model in comparison to the data. The nutricline intensity seem to be underestimated by about 50 % and its depth is overestimated. However, nutricline depth deepens from 100–120 m to 180–200 m between the western and the eastern basins (see Fig-

ure A3). The average chl-a concentration observed at the DYFAMED station in the top 200 m is 233 $\pm$ 146 ng L$^{-1}$ (average over the 1991–2005 period), while the model value for the HIS/A2 simulation over the same period is 159 $\pm$ 87 ng L$^{-1}$ (Figure A1).

In spite of some underestimation of nutrient concentrations that are probably linked with the features of the simulated intermediate and deep waters characteristics, the PISCES model reproduces

the main characteristics of the Mediterranean biogeochemistry, including a salient west–to–east gradient in nutrient concentrations, low surface nutrient concentrations and a deep chl-a maximum (DCM). These performances lend credence to our efforts to investigate the evolution of the Mediterranean biogeochemistry under the A2 climate change scenario with the same modeling platform.

### 3.2 Evolution of temperature and salinity

Average surface temperature and salinity (SST and SSS) evolution in the entire basin during the CTRL and HIS/A2 simulations are shown in Figure 3, which confirms results from Adloff et al. (2015) and shows that the CTRL simulation is stable over time. Beyond this basin–wide average

variation in SST and SSS, a more detailed analysis reveals much greater variability depending on the region (Somot et al., 2006; Adloff et al., 2015). For instance, the Balearic Sea is more sensitive to warming than the rest of the western basin, and the eastern basin has a more intense warming than the western basin (up to 3°C warming in the eastern basin and in the Balearic Sea). Also, the surface salinity in the Aegean Sea increases more than the other regions.

### 3.3 Evolution of the nutrient budgets in the Mediterranean Sea

The nutrient budgets of the semi–enclosed Mediterranean basin are highly dependent on external sources (e.g. Ludwig et al., 2009, 2010; Huertas et al., 2012; Christodoulaki et al., 2013). We first looked at the evolution of phosphate and nitrate fluxes in and out of the Mediterranean during our simulations. Then, in order to map the effects of climate change and external nutrient flux evolution on the Mediterranean nutrient balance, we calculated mass budgets of inorganic nitrate and phosphate during the simulated period. Finally we calculated the evolution of nutrient concentrations in different layers of the Mediterranean in order to point out the different effects of climate and nutrient fluxes on surface, intermediate and deep waters. The nutrient budgets account for changes in Atlantic input, river discharge and sedimentation. Nitrate can also accumulate in the Mediterranean waters through $N_2$ fixation by cyanobacteria, but this process accounts for less than 1 % of the total nitrate budget (Ibello et al., 2010; Bonnet et al., 2011; Yogev et al., 2011), and is neglected here.

In this Section, we refer to the period 1980-1999 as the beginning of the century, to the period 2030–2049 as the middle of the century and to the period 2080–2099 as the end of the century.

### 3.3.1 Fluxes of nutrients through the Strait of Gibraltar

The Mediterranean is connected to the global ocean by the narrow Strait of Gibraltar. Water masses transport through this strait contribute substantially to its water and nutrient budgets (e.g. Gómez, 2003; Huertas et al., 2012). The Mediterranean is a remineralization basin that has net negative fluxes of inorganic nutrients (i.e. organic nutrients enter the basin through the Gibraltar Strait surface waters and inorganic nutrient leave the Mediterranean through the deep waters of the Gibraltar Strait Huertas et al., 2012). Figure 4 shows the evolution of incoming and outgoing nitrate and phosphate fluxes through the Strait of Gibraltar in the HIS/A2 and in the CTRL simulations. We observe similar trends in phosphate and nitrate fluxes in the model. This commonality is linked to the Redfieldian behavior of the primary production in PISCES. The nutrient fluxes through the Strait of Gibraltar result from both the evolution of water fluxes, from NEMOMED8, and the evolution of nutrient concentrations in the buffer zone, from the A2 scenario from Dufresne et al. (2013).

In the HIS/A2 simulation, the incoming flux of nitrate decreases from 50 to 35 Gmol month$^{-1}$ while that for phosphate drops from 2.5 to 1.6 Gmol month$^{-1}$ until the middle of the century despite a period of increased incoming fluxes of both these nutrients in the 1990s. After 2050, fluxes increase

to reach values higher than the control in the last 25 years of simulations (Figure 4). By 2100, incoming nutrient fluxes have increased in the A2 scenario simulation by 13 % (2080–2099 minus 1980–1999 periods). This increase is statistically significant, with a linear regression having a positive slope with correlation coefficient greater than 0.75 and a p–value $< 0.001$. Furthermore, this increase follows a decrease of over 20 % in incoming nutrient fluxes between the 1980–1999 and the 2030–2049 periods. Most of the decrease is observed between 2030 and 2040 with a decrease of 15 and 1 Gmol month$^{-1}$ for nitrate and phosphate respectively during this decade.

Outgoing fluxes through the Strait of Gibraltar follow the same trends as incoming fluxes: total outgoing nitrate and phosphate fluxes decrease from 1980 to 2040 with flux values getting closer to zero and then increase until the end of the century. We observe a decreasing trend in the nitrate outgoing flux in the control from -129 in to -110 Gmol month$^{-1}$ representing about 15 % flux decrease. Over the simulation, outgoing nutrient fluxes increase less than incoming nutrient fluxes. The increase in outgoing nitrate and phosphate flux is less than 5 % (Figure 4). The relative changes in incoming and outgoing fluxes indicate an increase in the net incoming flux of about 5 %. The net flux at the beginning of the century is around -83 Gmol month$^{-1}$ for nitrate and -3 Gmol month$^{-1}$ for phosphate. At the end of the century, the fluxes are about -80 Gmol month$^{-1}$ and -2.5 Gmol month$^{-1}$ for nitrate and phosphate respectively. Also, these net fluxes are close to the CTRL net fluxes.

### 3.3.2 River fluxes of nutrients

River discharge is the main external source of phosphate for the eastern part of the basin (Krom et al., 2004; Christodoulaki et al., 2013). Figure 5 shows the total discharge of phosphate and nitrate from rivers to the Mediterranean Sea.

Phosphate discharge decreases by 25 % between the beginning and the end of the simulation period. As suggested by Ludwig et al. (2010), phosphate discharge in the A2 period stays lower than in the HIS period, in spite of a small discharge enhancement between 2030 and 2049.

Nitrate riverine discharge in the HIS/A2 simulation is substantially higher than in the CTRL simulation by 30 to 60 Gmol month$^{-1}$. The total river discharge of nitrate into the Mediterranean Sea has increased continuously from the 1960s (see the CTRL values for the years 1966–1981). According to the HIS/A2 simulation, total river nitrate discharge is 24 % larger during 2080-2099 than during 1980-1999.

### 3.3.3 Sedimentation

Sedimentation removes nutrients from the Mediterranean Sea. In this version of PISCES, the loss of nitrogen and phosphorus to the sediment is calculated from the sinking of particulate organic carbon (POC) to the sediment (linked through the Redfield ratio). Sediment fluxes of phosphorus and nitrogen during the simulations are shown in Figure 6.

The nutrient loss to the sediment decreases rapidly during the HIS simulation (1980–1999) and

remains low during the 21st century although it exhibits substantial interannual variability in the

sedimentation fluxes. By the end of the 21st century, sedimentation of phosphorus and nitrogen are

almost 50 % lower relative to the 1980 fluxes.

### 3.3.4   Phosphate and nitrate budgets under climate and biogeochemical changes in the

Mediterranean

Tables 2 and 3 summarize the average phosphate and nitrate water column content in all simulations

for the 3 time periods described earlier.

Total phosphate content in the entire Mediterranean grew in our HIS/A2 simulation by 6 % over

the 21st century, as determined by the difference between CTRL and HIS/A2 simulations between

1980-1999 and 2080–2099. The increase is larger in the eastern basin than in the western basin. In

particular, there is an 8 % increase in phosphate content in the Ionian–Levantine sub–basin. Nutrient

content in the HIS/A2 simulation is affected by both climate and nutrient fluxes from external sources

(rivers and fluxes via the Strait of Gibraltar). The effects of changes in riverine input of phosphate

are derived from the CTRL_R simulation (see also Figure 5). The difference of phosphate content

between CTRL and CTRL_R are substantial over the first half of the century. We observe 3 % de-

crease in phosphate content in the entire Mediterranean between 1980–1999 and 2030–2049 due to

river input changes (difference between CTRL_R and CTRL). Changes in phosphate fluxes thought

the Strait of Gibraltar seem to have limited effect on the global Mediterranean phosphate content

as revealed by the small difference between the beginning and the end of simulation CTRL_RG.

Conversely, climate change enhances the basin-wide phosphate content by 10 % in 2080–2099 rela-

tive to 1980–1999 (HIS/A2 minus CTRL_RG). Thus future changes in climate and external nutrient

supply may have opposite effects on nutrient concentrations in the Mediterranean.

Table 3 shows that by 2100, in HIS/A2 the combined effects of climate change, riverine, and Atlantic

nutrient input changes over the 21st century lead to a 17 % basin-wide increase in nitrate content

when compared to CTRL . Changes in river discharge lead to 9 % larger nitrate content by the end

of the century (2080–2099) compared to the beginning of the simulation period (1980–1999). The

most important effects of river input changes are observed in the Adriatic basin (over 50 % nitrate

accumulation by the end of the century). The largest effects from river input changes are found in the

Adriatic basin (>50 % nitrate accumulation by the end of the century). Across the Mediterranean,

there is only a weak (<1 %) effect on nitrate content from changes in the fluxes through the Strait of

Gibraltar. In the western basin, the comparison of CTRL_R and CTRL_RG reveals a 3 % decline in

nitrate content during the 2030–2049 period followed by an increase, reaching +1 % in 2080–2099 relative to 1980–1999. Finally, climate change enhances basin-wide nitrate content by 7 % between 1980–1999 and 2080–2099 period (HIS/A2 minus CTRL_RG). Thus river inputs and climate change are the main causes for changes in basin-wide nitrate content during the 21st century.

These global nutrient budgets reveal that climate change and external nutrient fluxes to the Mediterranean can influence its nutrient content in different sometimes even in opposing directions. In particular, river inputs have large effects on nutrient content in the eastern basin, while input through the Strait of Gibraltar has limited effects on the nutrient content even in the western basin.

### 3.3.5 Continuous evolution of phosphate and nitrate concentrations

In order to observe the continuous evolution of nutrient concentrations in different layers over the 21st century, we plotted the evolution of phosphate and nitrate concentrations for the entire simulation period in the western and eastern basins in the surface (0–200 m), intermediate (200–600 m) and deep (> 600 m) layers (Figures 7 and 8). With the separation between western and eastern basin being the Sicily Strait, the eastern basin includes the Ionian, Levantine, Adriatic and Aegean basins.

**Phosphate**

As shown in Figure 7a, and 7b, until mid century, the phosphate concentration in HIS/A2 decreases by about $0.015$ mmol m$^{-3}$ in the surface layer and by $0.017$ mmol m$^{-3}$ in the intermediate layer of the western basin. After 2050, phosphate concentration increases again until the end of the century, reaching concentrations that are similar to those in 1980 in the surface layer but at intermediate depths reaching values that are about $0.01$ mmol m$^{-3}$ higher than in that reference year. Figures 7a and 4b show that the evolution of surface phosphate concentration in the western basin in CTRL_RG and HIS/A2 are similar. The Pearson correlation coefficient between western basin phosphate concentration in HIS/A2 and incoming phosphate fluxes through the Strait of Gibraltar is 0.85, p–value $< 0.01$. Thus the phosphate concentration in the western basin appears to be influenced by phosphate inputs through the Strait of Gibraltar. However, Figure 7a shows that the difference in surface phosphate concentrations in the western Mediterranean in the CTRL_RG and CTRL_R simulations differ substantially only after about 2070. Thus the similar evolution of phosphate concentration in HIS/A2 and the incoming phosphate fluxes through the Strait of Gibraltar throughout the simulation period may be linked to changes in physical conditions. In this very dynamic Mediterranean region, changes in physical conditions linked with climate change precondition the western basin to become more sensitive to nutrient fluxes thought the Strait of Gibraltar.

A slight accumulation of about $0.015$ mmol m$^{-3}$ of phosphate is simulated in the HIS/A2 simulation in the deep western basin. The large difference between the HIS/A2 simulation and the control runs (over $0.010$  mmol m$^{-3}$ by the end of the century) shows that the evolution of the Mediterranean physics linked with climate change is primarily responsible for the changes in phosphate concentra-

tion in the intermediate and deep western basin. Climate change also decreases sediment fluxes (see

Figure 6) and increases stratification, thus isolating most of the phosphate pool from the surface.

Phosphate concentrations in the eastern part of the basin are lower than in the western part. Figure 7 illustrates the 50 % lower phosphate concentration in the surface layer, the roughly 30 % lower concentration in the intermediate layer and the 15 to 20 % lower concentration in the deep layer. In the surface layer, phosphate concentration decline in the beginning of the simulation from 0.022

mmol m$^{-3}$ in 1980 to less that 0.015 mmol m$^{-3}$ in 2000 and remains low during the 21st century (Figure 7d). There is, however, a large interannual variability in surface phosphate concentration with peaks up to 0.025 mmol m$^{-3}$ in 2060. But the HIS/A2 simulation values are consistently below the CTRL concentrations showing an important effect of climate change on surface phosphate reduction. We observe in Figures 7e and 7f an accumulation of phosphate in the intermediate and

deep layers of 17 and 13 % respectively, with large decennial variability of phosphate concentration in the deep eastern basin. In both of these layers, HIS/A2 concentrations are higher than the CTRL concentrations (over 0.015 mmol m$^{-3}$ higher by the end of the century). These results show that the evolution of phosphate concentrations in the eastern Mediterranean throughout the 21st century is mainly driven by climate change. Indeed, Figure 7 shows that average PO$_4$ concentrations in CTRL,

CTRL_R and CTRL_RG are similar for all periods in the eastern basin.

**Nitrate**

In the surface western basin, nitrate evolutions in the HIS/A2 and CTRL_RG simulations are similar, confirming the regulating effects of fluxes through the Strait of Gibraltar (Figure 8d). We observe an

accumulation of nitrate in the HIS/A2 simulation in the intermediate and deep Mediterranean waters between 9 and 20 % (Figures 8b, 8c, 8e and 8f). In the intermediate layer, nitrate concentration in the HIS/A2 simulation decreases from 4.1 to 3.6 mmol m$^{-3}$ between 1980 and 2030. After 2030, it increases again up to 4.3 mmol m$^{-3}$. In the deep western basin, we observe a slight decrease in nitrate concentration in the controls from 4.4 to about 4.2 mmol m$^{-3}$. In HIS/A2, nitrate concentration

follows the decrease of the controls until approximately 2020, then, there is a slight accumulation from 4.5 to 4.8 mmol m$^{-3}$ until the end of the century. Thus, physical changes linked with climate change have little effect on nitrate concentrations in the deep western Mediterranean until approximately 2020. Significant differences between HIS/A2 and CTRL appear in the middle of the 21st century.

In the eastern basin, the impacts of river discharges of nitrate seem to have large influence on the nitrate accumulation as shown by the similar evolution of HIS/A2 and CTRL_R simulations (Figures 8d, 8e and 8f). Figure 8d shows the contrasted effects of climate and biogeochemical changes. The strong difference between CTRL_R and CTRL concentrations at the beginning of the simulation (almost 0.4 mmol m$^{-3}$) indicates that riverine nutrient discharge has a strong influence on surface

nitrate concentrations in the eastern basin and is responsible for an important part of the eastern Mediterranean nitrate budget (see also Table 3). But the strong difference between CTRL_R and HIS/A2 at the end of the century indicates that vertical stratification leads to a decrease in surface layer nitrate concentrations, probably linked both with lower winter mixing and nutrient consumption by phytoplankton. In the intermediate and deep layers, the evolution of physical conditions has

a similarly large impact on the nitrate concentrations in the eastern basin as shown by the difference between CTRL_R and HIS/A2 (see also Table 3). In particular, nitrate concentrations increase by about 0.5 mmol m$^{-3}$ between 1980 and 2099 in the deep eastern basin. Approximately 50 % of this accumulation is due to river discharge. The large differences between the CTRL simulations and the HIS/A2 show that modification of circulation resulting from climate change have substantial impacts

on the deep and intermediate nutrient concentrations.

Evaluating separately the evolution of nutrient concentrations in different layer of the Mediterranean Sea shows that external nutrient fluxes primarily affect the surface in the western basin whereas climate change affects the entire water column. Also, climate and nutrient fluxes may have opposite

effects on surface nutrient concentration. This leads to different trends in nutrient concentrations in the surface layer and in the intermediate and deep layers. In particular, surface nitrate in the eastern basin is observed to increase as a result of increased river discharge, but climate change effects lower concentrations in HIS/A2 (see figure 8d). On the other hand, climate and river discharge of nitrate have similar effects on the intermediate and deep eastern layers, leading to the simulated increase in

nutrient content (Tables 2 and 3).

### 3.4  Surface nutrient concentrations, primary productivity and nutrient limitations

Figures 9 and 10 show the average surface concentrations of nitrate and phosphate in the beginning of the century (1980–1999) and the relative concentration differences for the end of the century in the HIS/A2 and CTRL simulations. Figure 9 confirms the previous results showing an accumulation

of nitrate in large zones of the basin by the end of the century, except for the southwestern part of the western basin (Alboran Sea), the Gulf of Lion, the North Levantine around the Rhodes Gyre and Crete and small areas in the southeastern Levantine, Tyrrhenian and Algerian basins.
On the contrary, Figure 10 shows that the surface phosphate concentration is decreasing over most of the Mediterranean basin except near the mouth of the Nile, the Ionian, Algerian, Tyrrhenian, be-

tween Crete and Cyprus and in the Alboran Sea. The specific concentrations observed next to the Nile mouth are linked to an inversion of the N:P ratio in this river in our scenario (i.e. an increase in P discharge and a decrease in N discharge). The distribution of surface phosphate concentration at the end of the century (2080–2099) shows that all P–rich areas of the eastern basin at the beginning of our simulations are depleted by the end of the simulation. For instance, the P-rich areas around

Crete and Cyprus no longer exist in the 2080–2099 period (Figure 10). Moreover, Figure 11 shows

that these areas match zones of high productivity. All the most productive zones of the beginning of the century are reduced in size and intensity by the end of the century. For instance, there is a 10 to 40 % decrease in primary production in the Gulf of Lion and around the Balearic Islands, more than 50 % reduction in the North Adriatic basin, in the Aegean Sea and in the eastern Levantine basin around Cyprus. There is also a reduction in primary productivity from 40-50 gC m$^{-2}$ year$^{-1}$ to 20-30 gC m$^{-2}$ year$^{-1}$ around Cyprus. These mesoscale changes may be linked with changes in local circulation (e.g., mesoscale eddies). These observations show that the evolution of the Mediterranean biogeochemistry is influenced by both meso- and large-scale circulations patterns. The simulated basin-wide vertical integral of primary production over the euphotic layer (0–200 m) declines by 10 % on average between 1980–1999 and 2080–2099. However, there is a productivity decrease of more than 50 % in areas such as the Aegean Sea and the Levantine Sea (Figure 11). In general, the differences in surface biogeochemical variables between 1980–1999 and 2080–2099 are weaker in the western basin because of the strong regulating impact of nutrient exchange through the Strait of Gibraltar. The large scale reduction of surface primary productivity may be a cause for the observed reduction in sedimentation (see Figure 6).

Figure 12 presents the limiting nutrient calculated using PISCES half–saturation coefficients (see Aumont and Bopp, 2006). The limiting nutrient is derived from the minimal value of limitation factors. In the Monod–type model PISCES, nutrient–based growth rates follow a Michaëlis–Menten evolution with nutrient concentrations. In the present period, most of the productive areas are N and P colimited in the simulation (Figure 12). This includes regions such as the Gulf of Lion, the South Adriatic Sea, the Aegean Sea and the northern Levantine basin. Future accumulation of nitrogen in the basin modifies the nutrient balance causing most eastern Mediterranean surface waters to become P–limited. The total balance of phosphate is more negative in the future than in the present period whereas we observe an inverse situation for nitrate. Therefore, phosphate becomes the major limiting nutrient in most of the regions where productivity is reduced such as the Aegean Sea, the northern Levantine basin and the north eastern Ionian Sea.

### 3.5 Modifications of the Mediterranean deep chlorophyll maximum and chl-a budget

Figure 13 shows the average depth of the simulated deep chlorophyll maximum (DCM) for the period 1980–1999 and for the period 2080–2099.

The DCM depth changes little during the simulation, even though salinity and temperature change. The DCM deepens slightly in some regions such as the North Ionian and the South of Crete. Although the DCM depth changes little in the future, the intensity of subsurface productivity is reduced

(see Figure 11).

Figure 14 shows the average vertical profiles of chl-a at the DYFAMED station (43.25° N, 7.52° E) and the average profiles for the western and eastern basins for the 1980–1999 and 2080–2099 periods. The subsurface chl-a maximum persists through to the end of the century. At the DYFAMED station, the average DCM depth remains unchanged, while the surface chl-a concentration is decreased by about 25 ng $L^{-1}$, which is negligible. Thus the average chl-a profile at DYFAMED changes little throughout the simulation. However, at that station there is approximately 40 % variability in the chl-a concentration profile and depth of DCM for some month (not shown). In the western basin, the subsurface maxima at present and in the future are located at the same depth (100–120 m), but the average chl-a concentration in the DCM increases by about 15 to 20 ng $L^{-1}$ during the simulation. However, where there are small changes in the average chl-a profiles in the western basin, those are often accompanied by greater local changes in the depth and intensity of the DCM (Figure 13). In the eastern basin, subsurface chl-a concentration is reduced by about 50 ng $L^{-1}$ and the subsurface chl-a maximum deepens from 100–120 m to below 150 m.

In the oligotrophic Mediterranean, the majority of the chl-a is produced within the DCM. There is an 8.9 % reduction in integrated chl-a production between the 1980–1999 and 2080–2099 due to circulation changes combined with changes in fluxes through the Strait of Gibraltar and riverine inputs. Table 4 reports total chl-a budgets in the 1980–1999, 2030–2049 and 2080–2099 periods of all the simulations in all Mediterranean subbasins (Figure 2 from Adloff et al., 2015). It reveals that chl-a budget is stable over the CTRL simulation but decreases in all Mediterranean subbasins over the HIS/A2 simulation. The decrease in chl-a is larger in the eastern regions, in particular in the Adriatic and Aegean Seas (-17 and -19 % respectively). In the western basin, the decline in chl-a is smaller (-5.1 %). The chl-a budget is probably maintained by the enhanced nutrient fluxes through the Strait of Gibraltar (chl-a in CTRL_RG does not significantly decrease in the western basin).

About 85 % of the future reduction in chl-a in HIS/A2 is explained by the effects of climate change (HIS/A2 minus CTRL_RG). However, the effects from increased nutrient inputs through the Strait of Gibraltar, decreased riverine phosphate inputs and increased riverine inputs of nitrate seem to have opposing effects to climate and circulation changes on chl-a production. In particular, in the western basin, reductions in riverine discharge of nutrients reduce chl-a by 3.5 % (see CTRL_R values), whereas changes in fluxes through the Strait of Gibraltar enhance chl-a (only 1 % decrease in chlorophyll concentration in CTRL_RG in the western basin).

### 3.6 Plankton biomass evolution

Most of the biological activity in the marine environment is found within the euphotic layer which is confined to the upper 200 m. Figure 15 shows the evolution of nanophytoplankton and diatom concentrations (in terms of carbon content mmolC m$^{-3}$) over the top 200 m in the western and eastern basins throughout the duration of all simulations. In HIS/A2, the biomass of both phytoplankton classes declines during the simulation (-0.01 mmolC m$^{-3}$ for nanophytoplankton and -0.03 to -0.04 mmolC m$^{-3}$ for diatoms). Generally, diatoms appear more sensitive to climate change as their biomass decreases more sharply than does that for nanophytoplankton. As shown in Figure 15c, diatom concentrations in the western basin appear sensitive to changes in nutrient input across the Strait of Gibraltar as indicated by the large difference between CTRL_RG and CTRL_R (0.04 mmolC m$^{-3}$). However, the reduction in diatom concentration found with HIS/A2 indicates that it is primarily influenced by climatic drivers.

The same general evolution is found for zooplankton as seen for phytoplankton (Figure 16). In HIS/A2 in all basins there is a decrease in microzooplankton concentration during 1980–2000 (from 0.165 to 0.114 mmol m$^{-3}$), after which it remains stable and consistently below the CTRL values until the end of the simulation in 2100. In the eastern basin, there is a large reduction in mesozooplankton levels. The average mesozooplankton concentration in the eastern part of the Mediterranean declines by almost 60 % in 2099 in comparison to that in 1980. However, the average mesozooplankton concentration over the 2080–2099 period is only slightly lower than the average concentration over the 1980–1999 period (0.10 and 0.11 mmol m$^{-3}$ respectively) because the decline in concentration occurs within the first years of the HIS/A2 simulation. In the western basin, there is a marked decline in mesozooplankton concentrations between 1980 and 2040 (0.05 mmol m$^{-3}$). After 2040, the surface concentration of mesozooplankton increases regularly to values that are similar to those at the beginning of the simulation. This evolution is similar to those of nutrient concentrations in surface waters of the western basin (Figure 7). In the PISCES model, zooplankton and particularly mesozooplankton are especially sensitive to the variations of external climatic and biogeochemical conditions, being the highest trophic level that is represented. Owing to their bottom–up control, zooplankton canalize all changes at the basin scale and ultimately displays the largest response. This behavior is similar to the trophic amplification observed by Chust et al. (2014) and Lefort et al. (2015).

Altogether, the analysis of plankton biomass evolution during the simulation period suggests that primary and secondary production in the eastern basin are more sensitive to climate change than in the western basin. The eastern basin is more isolated from the open Atlantic Ocean than western basin as it receives less nutrients from the Atlantic and from coastal inputs. The eastern basin is also deeper and less productive than the western basin (Crispi et al., 2001). The eastern basin exhibits a

decline in the phytoplankton biomass that is similar to the decline in the phosphate concentration. Biological production is mainly P–limited in this basin (see also Figure 12). Therefore, the constant low concentrations of phosphate observed throughout this century limit biological production and keep plankton biomass at low levels.

### 3.7   Effects of aerosol deposition on surface primary productivity

Figure 17 shows the relative effects of total nitrogen and natural dust deposition on surface primary production in 1980–1999 and 2080–2099. As shown in Richon et al. (2017), dust deposition enhances surface primary productivity in the southern part of the basin in 1980–1999 whereas nitrogen deposition enhances primary productivity in the northern part of the basin. As our HIS/A2 simulation shows a decrease in surface $PO_4$ concentrations, thus accentuating phosphate limitation over the Mediterranean basin by the end of the 21st century, the relative impact of phosphate deposition from dust would be enhanced in the 2080–2099 period relative to the 1980–1999 period. Conversely, nitrogen atmospheric deposition has very little effect on Mediterranean primary production at the end of the simulation period because most of the basin is not N–limited.

## 4   Discussion

### 4.1   Sources of uncertainties

The study represents the first transient long term simulations of the Mediterranean Sea with a coupled physical–biogeochemical high resolution model. It provides a first glimpse of the sensitivity of the Mediterranean Sea biogeochemistry to climate change and to the evolution of external nutrient fluxes. As for all modeling studies, our conclusions are subject to some limitations that we attempt to underline in this section.

#### 4.1.1   Climate change scenario

Although the physical model adequately represents the MTHC (Adloff et al., 2015), there are many uncertainties linked with climate change projections. Some are discussed in Somot et al. (2006), in particular, the need to use different IPCC scenarios for climate change projections and MTHC changes. Adloff et al. (2015) apply an ensemble of SRES scenarios and boundary conditions to the Mediterranean Sea and suggest that the choice of atmospheric and Atlantic conditions has a strong influence on the MTHC. This influence is mainly linked with the evolution of stratification index and vertical mixing. Overall, the increase in stratification in our A2 climate change scenario leads to different evolutions of nutrient concentrations between the surface and the intermediate and deep waters with surface waters becoming more sensitive to external nutrient sources (Figures 7 and 8). On the other hand, Macias et al. (2015) found that primary productivity slightly increased as a result

of decreased stratification in the climate change scenarios RCP 8.5 and RCP 4.5. The A2 scenario that we used was the only one available with 3–D daily forcings, as necessary for coupling with the PISCES biogeochemical model. However, Adloff et al. (2015) showed that other SRES scenarios such as the A1B or B1 may lead to a future decline in the vertical stratification with probably dif-

ferent consequences on the Mediterranean Sea biogeochemistry. Our study is thus only a first step for transient modeling of the Mediterranean Sea biogeochemistry. It should be complemented by new simulations that explore the various sources of uncertainty (model choice, internal variability, scenario choice) once appropriate forcings become available for multiple models as expected from the Med–CORDEX initiative (Ruti et al., 2016).

Freshwater runoff in the physical model may also influence the circulation and nutrient concentra-tions at the river mouth. Adloff et al. (2015) evaluated the changes in total freshwater runoff in the HIS/A2 simulation. Their Table 2 shows that the total freshwater runoff to the Mediterranean is lower than the Ludwig et al. (2009) estimate (by about 30 %). They found an approximately 27 % decrease in total runoff by the end of the 21st century. This trend is consistent with the deceasing

trend found by Ludwig et al. (2010). However, the 2050 estimates of freshwater runoff from Ludwig et al. (2010) are only 13 % lower than the 1970 and 2000 estimates. The freshwater runoff decrease in the physical model from Adloff et al. (2015) is more important than in the nutrient runoff model from Ludwig et al. (2010). This decrease may result in higher nutrient concentrations at the river mouth. We are also aware that the future evolution of river discharges into the Mediterranean Sea is

highly uncertain and depends on the choices for the scenario and the model (Sanchez-Gomez et al., 2009; Dubois et al., 2012; Adloff et al., 2015).

### 4.1.2 Uncertainties from the PISCES model

The evaluation of the CTRL simulation showed that NEMOMED8/PISCES is stable over time in spite of a slight drift in nitrate concentrations (see Figure 8). Nutrient concentrations in the interme-

diate and deep layers were underestimated in comparison to measurements (see Appendix). Nutrient concentrations were underestimated by up to 50 %, in particular in the deep eastern basin. Moreover, nitrate fluxes from coastal discharge in CTRL are lower than in HIS/A2. The low riverine discharge and the imbalance in sources and sinks explains the loss of nitrate in the CTRL (see Figures 8 and 9). Organic forms of nutrients are not directly available to phytoplankton in this version of PISCES,

and are not included in our nutrient budgets. Powley et al. (2017) show that organic forms of nu-trient are an important part of the Mediterranean elemental budgets. Therefore, we appear to be missing part of the N and P budgets in our calculations. The simulated chl-a vertical profiles at the DYFAMED station show a reasonable representation of the subsurface productivity maximum of the Mediterranean in spite of a mismatch in the subsurface chl-a maximum depth between model

and measurements. Model values were not corrected to match data, hence, the uncertainties in the representation of present–day biogeochemistry by the PISCES model may be propagated into the

future.

In the version of PISCES used in this study, variations in nitrate and phosphate are linked by the Redfield ratio (Redfield et al., 1963). The Redfield hypothesis of a fixed nutrient ratio used for
plankton growth and excretion holds true for most parts of the global ocean, but may not be true for oligotrophic regions such as the Mediterranean Sea (e.g. Béthoux and Copin-Montégut, 1986). Moreover, changes in nutrient balance influence the nutrient limitations as shown by Figure 12. Yet, the results simulated with the Redfieldian hypothesis are coherent with the observed variations of nutrient supply to the Mediterranean Sea and yield realistic biological productivity. Nonetheless results
concerning nutrient limitation could change in a non Redfieldian biogeochemistry model.

### 4.1.3  External nutrient sources

Climate change may impact all drivers of biogeochemical cycles in the ocean. In the case of semi–enclosed seas like the Mediterranean, the biogeochemistry is heavily influenced by external sources of nutrients (namely rivers, Atlantic and atmospheric inputs, see Ludwig et al., 2009; Krom et al.,
2010) and modification of the physical ocean (e.g. vertical mixing, horizontal advection, see Santinelli et al., 2012). Nutrient fluxes from external sources (rivers, aerosols and fluxes through the Strait of Gibraltar) may evolve separately depending on future socio-economic choices and climate feedbacks. In this study, different scenarios were used for river inputs ("Order from Strength" from Ludwig et al., 2010, based on the Millenium Ecosystem Assessment report) and for Atlantic nutri-
ent concentrations (SRES/A2 from Dufresne et al., 2013). It is important to accurately represent the incoming nutrient fluxes to the Mediterranean and their potential evolution with regards to climate change, as they have important influence on the Mediterranean Sea biogeochemistry.

**Fluxes through the Strait of Gibraltar**

Results from our CTRL_RG simulation show that the increase in nitrate and phosphate incoming
fluxes through the Strait of Gibraltar leads to higher surface concentrations in the western Mediterranean. The Atlantic nutrient concentrations are derived from a global version of the same model used our simulations (NEMO/PISCES) and forced under the same A2 climate change scenario). Therefore, there is no incompatibility issue between for the forcing and model.

**Riverine nutrient fluxes**

Additionally, our CTRL_R simulation shows that the increase in riverine nitrate fluxes leads to the accumulation of nitrate in the surface Mediterranean, in particular in the eastern basin and in the Adriatic. For the riverine nutrient inputs, scenarios from the MEA report are based on different assumptions from the IPCC SRES scenarios used to compute freshwater runoff in the HIS/A2 simulation. Freshwater discharge from Ludwig et al. (2010) is based on the SESAME model reconstruc-
tion and differs from freshwater runoff in the ARPEGE–Climate model used to force our physical

model. This may lead to incoherences between water and nutrient discharges, but the nutrient discharges from Ludwig et al. (2010) are the only ones that are available. Furthermore, the SESAME model is not coupled with NEMO/PISCES. Associated discrepancies and the uncertainties linked with the use of inconsistent scenarios in our simulation should be addressed by developing a more integrated modelling framework to study the impacts of climate change on the Mediterranean Sea biogeochemistry. As there is no consensus nor validated scenario for nutrient fluxes from riverine runoff in the Mediterranean, we chose to use one scenario from Ludwig et al. (2010). This scenario has the advantage of being derived from a coherent modeling framework. However, the Ludwig et al. (2010) nutrient discharge transient scenario does not represent the interannual variability of nutrient runoff from rivers. Moreover, according to these authors, the socio–economic decisions made in the 21st century will influence nitrate and phosphate discharge over the Mediterranean. It is difficult to forecast these decisions and the resulting changes in nutrient fluxes are uncertain.

**Potential effects of aerosol deposition**

The biogeochemistry of the Mediterranean is significantly influenced by aerosol deposition (e.g. Krom et al., 2010; Dulac et al., 1989; Richon et al., 2018, 2017; Guieu et al., 2014). The future evolution of the multiple aerosol sources surrounding the Mediterranean will likely influence the response of the Mediterranean to climate change.

Results from the HIS/A2_NALADIN simulation show that enhanced phosphate fluxes from aerosols may limit the surface decrease of phosphate concentrations and limit phosphorus limitation. However, in the HIS/A2_NALADIN simulation, the surface Mediterranean is still P-limited in most of the Mediterranean because the atmospheric nutrient fluxes are low in comparison to riverine nutrient fluxes from rivers and the nutrient flux through the Strait of Gibraltar (see Richon et al., 2017). Therefore, it appears unlikely that changes in aerosol deposition from natural dust would greatly influence future Mediterranean biogeochemistry. However, there are multiple sources of aerosols that are not included in atmospheric models, e.g., anthropogenic, volcanic and volatile organic compounds (e.g. Wang et al., 2014; Kanakidou et al., 2016). Their combined influence could perhaps constitute an important nutrient flux to the Mediterranean, thus altering the evolution of its biogeochemistry. Moreover, aerosols affect radiative forcing over the Mediterranean and may impact the climate conditions (Nabat et al., 2015b). Thus, efforts should be made to accurately represent this nutrient source in Mediterranean models to assess the effect on Mediterranean Sea biogeochemistry with regards to climate change.

Our results show that the state of the Mediterranean biogeochemistry at the end of the 21st century is the result of the combined evolutions of both climate and external nutrient fluxes. Therefore, it is very difficult to predict the future evolution of the Mediterranean based on the evolution of one of these components only. This is why it is important, in the case of semi–enclosed basins, to produce reliable estimates of the evolution of all the components influencing the biogeochemistry.

## 4.2  Climate versus biogeochemical changes effects

Figure 18 summarizes the fluxes of phosphate and nitrate in and out of the Mediterranean considered in this study.


In general, the sum of nitrogen net fluxes into the Mediterranean basin (Riverine, Gibraltar Strait and sedimentary sources and sinks) increases by 39 % at the end of the century in the scenario (HIS/A2) whereas it is increased by 23 % in the control (CTRL) in comparison to the beginning of the simulation. The balance between inputs and outputs of phosphorus increases by 9 % in the scenario and by 11 % in the control. These results suggest a substantial accumulation of nitrogen in the Mediterranean basin over the century when phosphorus fluxes can be considered roughly stable. The strong decrease in sedimentation (Figure 6) occurring in spite of enhanced nutrient flux from the Atlantic and an enhanced nitrate river flux may be linked to the decrease in vertical water fluxes (upwelling and downwelling). This would explain the accumulation of phosphate and nitrate in the deep layers of the Mediterranean Sea (Figures 7c, 7f,8c and 8f).



Results from our transient simulations show that nutrient concentrations may evolve differently depending on the region and the depth in response to climate change and external nutrient inputs. In the surface western Mediterranean, the effects of climate change and enhanced nutrient fluxes via Gibraltar both concur to the increase in nutrient concentrations (Figures 7a and 8a). In the surface eastern basin, river fluxes of nitrate and stratification have opposing effects on nitrate concentrations whereas phosphate concentrations are mainly driven by climate change effects (Figures 7d and 8d). The difference between HIS/A2 and CTRL_RG phosphate concentrations in the intermediate and deep layers (Figures 7b, 7e, 7c, 7f) indicates that variations of phosphate concentrations during the 21st century are primarily driven by climate change while nitrate concentration is equally sensitive to changes in biogeochemical forcings (Figures 8b, 8e, 8c, 8f).



Results from our different control simulations indicates the extent to which the choice of the biogeochemical forcing scenario may influence the future evolution of the Mediterranean Sea biogeochemistry. Results from Table 4 (CTRL_RG minus CTRL_R) indicate that the increase in nutrient input through the Strait of Gibraltar at the end of the century is responsible for a 2.5 % increase in chl-a concentration in the western basin by the end of the century. Herrmann et al. (2014) simulated an increase in chl-a concentration associated with climate change effects in a small region of the north western basin by the end of the century. Thus, the separate effects of climate change and external nutrient inputs may have synergetistic effects on the evolution of the western Mediterranean chl-a.


In some parts of the eastern basin, the effects from riverine nutrient fluxes on chl-a appear more important than those from climate change (see Table 4). In the Adriatic Sea, Table 3 shows that riverine nitrate discharge is responsible for 41 % increase in nitrate concentration over the simulation period (2080–2099 minus 1980–1999). In the CTRL_RG simulation, nitrate concentrations are similar to those in the CTRL_R simulation, indicating no influence of fluxes through the Strait of


Gibraltar in the Adriatic Sea. Finally, nitrate concentrations in the HIS/A2 simulation are close to the CTRL_R values showing that most of the nitrate evolution in the Adriatic Sea is linked with riverine discharge. Lazzari et al. (2014) also conclude that the river mouth regions are highly sensitive because the Mediterranean Sea is influenced by external nutrient inputs. The choice of river runoff scenario will likely influence the evolution of nutrient concentrations and the biogeochemistry in many coastal regions such as the Adriatic Sea (see also Spillman et al., 2007).

To our knowledge, this is the first attempt to study the basin–scale biogeochemical evolution using a transient business–as–usual (A2) climate change scenario. Lazzari et al. (2014) tested the effects of several land–use change scenarios on the A1B SRES climate change scenario over 10–years time slices. They found a general decrease in phytoplankton and zooplankton biomass (about 5 %), which is lower than in our severe climate change scenario. Considering only changes in climate, Herrmann et al. (2014) and Macias et al. (2015) studied the transient biogeochemical evolution of the Mediterranean Sea (Herrmann et al., 2014, only studied a small region in the north western Mediterranean) studied the transient biogeochemical evolution of the Mediterranean Sea under different climate change scenarios, with the former study focusing on a small region in the northwestern Mediterranean. Both studies found that chlorophyll concentration and plankton biomass increase slightly due to changes in vertical stratification. In our simulations, average phytoplankton biomass decreases by 2 to 30 % (see Figure 15) and average zooplankton biomass decreases by 8 to 12 % (see Figure 16). However, our transient simulations revealed non linear trends in plankton biomass evolution as a results of the influence of external nutrient fluxes. Chust et al. (2014) have shown that regional seas and in particular the Aegean and Adriatic were sensitive to trophic amplification. Our results appear to agree, showing signs of trophic amplification (see Figures 15 and 16). Assessing the sensitivity of the Mediterranean to trophic amplification would require more simulations focused on the evolution of Mediterranean planktonic biomass under different climate change scenarios.

The modifications of chl-a production and plankton biomass are linked to changes in nutrient limitation (Figure 12). Finding no clear definition of nutrient co–limitation, we consider that N and P are co-limiting when the difference in limitation factors is less than 1 %. This definition of nutrient co–limitation applies well to the Mediterranean case because of its very low nutrient concentrations. Our results are confirmed by some studies (Thingstad et al., 2005; Tanaka et al., 2011). However, our nutrient limitations are calculated from 20–years average nutrient concentrations and nutrient limitation may vary greatly during the seasonal cycle (Marty et al., 2002; Diaz et al., 2001). It has also been hypothesized by Luna et al. (2012) that the warm temperature of the deep Mediterranean enhance nutrient recycling via prokaryotic metabolism. Therefore, a part of the nutrient accumulation we observed may be linked with the increase in temperature.

**Conclusion**

This study aims at assessing the transient effects of climate and biogeochemical changes on the
Mediterranean Sea biogeogeochemistry under the high–emission SRES A2 scenario. The NEMOMED8/PISCES
model adequately reproduces the main characteristics of the Mediterranean Sea: the west–to–east
gradient in productivity, the main productive zones, and the presence of a DCM. Hence, it appears
reasonable to use it to study the future evolution of the biogeochemistry of the Mediterranean basin
in response to increasing atmospheric $CO_2$ and resulting climate change. Our study is the first to
offer a continuous simulation over the entire period of the future IPCC scenario (A2), between 2000
and 2099.

Its results illustrate how future changes in physical and biogeochemical conditions, including warm-
ing, increased stratification, and changes in Atlantic and river inputs, can lead to a significant accu-
mulation of nitrate and a decrease in biological productivity in the surface, thus affecting the entire
Mediterranean ecosystem.

Our results also illustrate how climate change and nutrient inputs from riverine sources and fluxes
through the Strait of Gibraltar have contrasting influences on the Mediterranean Sea productivity. In
particular, the biogeochemistry in the western basin displays similar trends as that for nutrient input
across the Strait of Gibraltar. Therefore, it appears critical to correctly represent the future varia-
tions of external biogeochemical forcings of the Mediterranean Sea as they may have an equally
important influence on surface biogeochemical cycles as does climate. The biogeochemistry of the
eastern basin is more sensitive to vertical mixing and river inputs than is the western basin, which is
regulated by input through the Strait of Gibraltar. Increased future stratification also reduces surface
productivity in the eastern basin.

Although this study does account for the changes in fluxes through the Strait of Gibraltar and river-
ine inputs, some potentially important sources are missing such as direct wastewater discharge,
submarine groundwater input and atmospheric deposition. These additional nutrient sources are
poorly known, with a general lack of both measurements and models as needed to build com-
prehensive datasets for past and future evolution of these nutrient sources. The HIS/A2_N and
HIS/A2_NALADIN simulations presented in this study include continued present–day nitrogen and
phosphate deposition. Although these atmospheric fluxes have been evaluated previously and were
shown to represent correctly the deposition fluxes, there is no guarantee that these fluxes will remain
constant over the next century. Results indicate that the future sensitivity of the Mediterranean to
atmospheric deposition depends on the surface nutrient limitation, which may in part be influenced
by aerosol deposition. However, there is to our knowledge no available transient scenario for the 21st
century evolution of atmospheric deposition and no ensemble simulations to assess the future evo-
lution of the Mediterranean Sea under different climate change scenarios. A new generation of fully
coupled regional models have been developed and used to study aerosols climatic impacts (Nabat
et al., 2015b). These models include a representation of the ocean, atmosphere, aerosols and rivers

and could eventually be used to make consistent future climate projections at the regional scale of the Mediterranean.

## Acknowledgments

The authors would like to thank Florence Sevault for the physical simulations of NEMOMED8, Pierre Nabat for the dust deposition with ALADIN-Climat, Yves Balkanski and Rong Wang for the
N deposition with the LMDz-INCA model and Wolfgang Ludwig for the river inputs. This work was funded by CEA (C. Richon PhD grant) and is part of the MISTRALS project. Simulations were made using HPC resources from the French GENCI program (grant x2015010040).

## Appendix A: Evaluation of the NEMOMED8/PISCES model

The comparison of modeled surface chl-a concentration with satellite estimates has revealed that the
model correctly simulates the main characteristics observed in the Mediterranean Sea (Figures 1 and 2). Comparison with in situ observation provides more refined estimates.

Figure A1 presents the average chl-a profiles at the DYFAMED station (43.25°N, 7.52°E) compared with measured concentrations for the month of February (low stratification, high productivity)
and May (end of spring bloom, beginning of stratification and DCM appearance). There are few data points below 200 m. The model produces the characteristic of the deep chl-a maximum generated in May, even if it is too deep. The colors show that the model represents some interannual variability in chl-a production in spite of a consistent bias.

The overestimated DCM depth may be due to the overestimation of nitracline and phosphacline as shown by figure A2.

The vertical distribution of nitrate and phosphate concentrations along a west–to–east transect is shown in Figure A3. The model produces the salient west–to–east gradient of nutrient concentra-
tions. Concentrations in the surface layer appear realistic. The nutricline is located between 100 and 150 m in the western basin and deepens to around 180 to 200 m in the eastern basin. Although the model represents the spatial variability of the nutricline, it is too smooth, leading to the underestimation of deep water concentrations (by about 30 to 50 %).

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

| Name | Dynamics (NEMO years) | Buffer zone concentrations | River inputs | N deposition | P deposition |
|------|----------------------|---------------------------|--------------|--------------|--------------|
| CTRL | 1966–1981 | 1966–1981 | 1966–1981 | No | No |
| CTRL_R | 1966–1981 | 1966–1981 | 1980–2099 | No | No |
| CTRL_RG | 1966–1981 | 1980–2099 | 1980–2099 | No | No |
| HIS/A2 | 1980–2099 | 1980–2099 | 1980–2099 | No | No |
| HIS/A2_N | 1980–2099 | 1980–2099 | 1980–2099 | 1997–2012 | No |
| HIS/A2_NALADIN | 1980–2099 | 1980–2099 | 1980–2099 | 1997–2012 | 1980–2012 |

**Table 1.** Description of the simulations. The years indicate the forcing years throughout the 120 years of simulation. The cycles are repeated in the CTRL simulations.

| Simulation | Period | Whole Med. | Western | Eastern | Ionian-Levantine | Adriatic | Aegean | Atlantic buffer zone |
|---|---|---|---|---|---|---|---|---|
| HIS/A2 | 1980–1999 | 551 | 241 | 310 | 305 | 1.5 | 4.0 | 535 |
| | 2030–2049 | 570 (+3.4) | 240 (0) | 329 (+6.1) | 324 (+6.2) | 1.4 (0) | 3.6 (-10) | 543 (+1.5) |
| | 2080–2099 | 598 (+8.5) | 251 (+4.1) | 346 (+11.6) | 341 (+11.8) | 1.5 (0) | 3.5 (-12.5) | 562 (+5.0) |
| CTRL | 1980–1999 | 545 | 238 | 307 | 302 | 1.6 | 4.0 | 532 |
| | 2030–2049 | 553 (+1.5) | 238 (0) | 314 (+2.3) | 309 (+2.3) | 1.6 (0) | 4.2 (+5.0) | 532 (0) |
| | 2080–2099 | 560 (+2.6) | 241 (+1.3) | 319 (+3.9) | 313 (+3.6) | 1.7 (+6.3) | 4.2 (+5.0) | 532 (0) |
| CTRL_R | 1980–1999 | 547 | 239 | 309 | 303 | 1.5 | 4.2 | 534 |
| | 2030-2049 | 536 (-2.0) | 232 (-2.9) | 304 (-1.6) | 299 (-1.3) | 1.4 (-6.7) | 3.5 (-17) | 534 (0) |
| | 2080-2099 | 538 (-1.6) | 230 (-3.8) | 309 (0) | 303 (0) | 1.5 (0) | 3.7 (-12) | 534 (0) |
| CTRL_RG | 1980–1999 | 548 | 239 | 309 | 303 | 1.5 | 4.2 | 535 |
| | 2030-2049 | 536 (-2.2) | 233 (-2.5) | 303 (-1.9) | 298 (-1.7) | 1.4 (-6.7) | 3.5 (-17) | 544 (+1.7) |
| | 2080-2099 | 540 (-1.5) | 235 (-1.7) | 306 (-1.0) | 301 (-0.7) | 1.4 (-6.7) | 3.6 (-14) | 562 (+5.0) |

**Table 2.** Simulated integrated phosphate content ($10^9$ mol) over 20-year periods in the Mediterranean sub–basins in the different simulations. Basins are the same as defined in Fig.2 of Adloff et al. (2015), with the eastern basin including the Ionian, Levantine, Adriatic and Aegean subbasins. Values in parenthesis indicate the percentage difference from the 1980–1999 period.

| Simulation | Period | Whole Med. | Western | Eastern | Ionian-Levantine | Adriatic | Aegean | Atlantic buffer zone |
|---|---|---|---|---|---|---|---|---|
| HIS/A2 | 1980–1999 | 13400 | 5520 | 7890 | 7690 | 66.9 | 132 | 8091 |
| | 2030–2049 | 13800 (+3.0) | 5450 (-1.3) | 8350 (+5.8) | 8100 (+5.3) | 88.5 (+32) | 163 (+23) | 8230 (+1.7) |
| | 2080–2099 | 14700 (+9.7) | 5750 (+4.2) | 8920 (+13) | 8650 (+12) | 98.3 (+47) | 164 (+24) | 8510 (+5.2) |
| CTRL | 1980–1999 | 13500 | 5530 | 7970 | 7760 | 66.2 | 144 | 8050 |
| | 2030–2049 | 12900 (-4.4) | 5320 (-3.8) | 7610 (-4.5) | 7420 (-4.4) | 61.7 (-6.8) | 131 (-9.0) | 8050 (0) |
| | 2080–2099 | 12500 (-7.4) | 5170 (-6.5) | 7330 (-8.0) | 7150 (-7.9) | 58.7 (-11) | 123 (-15) | 8050 (0) |
| CTRL_R | 1980–1999 | 13300 | 5470 | 7870 | 7760 | 66.8 | 138 | 8070 |
| | 2030–2049 | 13300 (0) | 5250 (-4.0) | 8020 (+1.9) | 7770 (+0.1) | 88.0 (+32) | 162 (+17) | 8070 (0) |
| | 2080–2099 | 13700 (+3.0) | 5300 (-3.1) | 8440 (+7.2) | 8170 (+5.3) | 94.2 (+41) | 177 (+28) | 8080 (+0.1) |
| CTRL_RG | 1980–1999 | 13300 | 5480 | 7870 | 7760 | 66.8 | 138 | 8090 |
| | 2030–2049 | 13300 (0) | 5270 (-3.8) | 8010 (+1.8) | 7760 (0) | 88.0 (+32) | 162 (+17) | 8080 (-0.1) |
| | 2080–2099 | 13800 (+3.8) | 5390 (-1.6) | 8430 (+7.1) | 8160 (+5.2) | 94.3 (+41) | 177 (+28) | 8080 (-0.1) |

**Table 3.** Simulated integrated nitrate content ($10^9$ mol) over 20-year periods in the Mediterranean sub–basins in the different simulations. Basins are the same as defined in Fig.2 of Adloff et al. (2015), with the eastern basin including the Ionian, Levantine, Adriatic and Aegean subbasins. Values in parenthesis indicate the percentage difference from the 1980–1999 period.

1205

| Simulation | Period | Whole Med. | Western | Eastern | Ionian-Levantine | Adriatic | Aegean | Atlantic buffer zone |
|---|---|---|---|---|---|---|---|---|
| HIS/A2 | 1980–1999 | 25700 | 9680 | 16000 | 13500 | 830 | 1720 | 3210 |
| | 2030–2049 | 23800 (-7.4) | 8980 (-7.2) | 14800 (-7.5) | 12700 (-5.9) | 720 (-13) | 1440 (-16) | 3280 (+2.2) |
| | 2080–2099 | 23400 (-8.9) | 9180 (-5.1) | 14300 (-11) | 12200 (-9.6) | 690 (-17) | 1390 (-19) | 3570 (+11) |
| CTRL | 1980–1999 | 27000 | 10200 | 16700 | 14200 | 880 | 1670 | 3180 |
| | 2030–2049 | 27000 (0) | 10200 (0) | 16900 (+1.2) | 14300 (+0.7) | 890 (+1.1) | 1710 (+2.4) | 3180 (0) |
| | 2080–2099 | 26600 (-1.5) | 9980 (-2.2) | 16600 (-0.1) | 14000 (-1.4) | 880 (0) | 1690 (+1.2) | 3180 (0) |
| CTRL_R | 1980–1999 | 27000 | 10300 | 16700 | 14100 | 875 | 1720 | 3210 |
| | 2030–2049 | 26800 (-0.7) | 10100 (-1.9) | 16700 (0) | 14300 (+1.4) | 780 (-12) | 1610 (-6.4) | 3210 (0) |
| | 2080–2099 | 26400 (-2.2) | 9940 (-3.5) | 16500 (-1.2) | 14100 (0) | 760 (-13) | 1600 (-7.0) | 3220 (0.3) |
| CTRL_RG | 1980–1999 | 27000 | 10300 | 16700 | 14100 | 875 | 1720 | 3230 |
| | 2030–2049 | 26900 (-0.4) | 10200 (-1.0) | 16700 (0) | 14300 (+1.4) | 780 (-12) | 1600 (-7.0) | 3260 (+0.9) |
| | 2080–2099 | 26700 (-1.1) | 10200 (-1.0) | 16500 (-1.2) | 14100 (0) | 750 (-14) | 1600 (-7.0) | 3420 (+5.9) |

**Table 4.** Simulated integrated chl-a ($10^9$ mol) over 20-year periods in the Mediterranean sub–basins in the different simulations. Basins are the same as defined in Fig.2 of Adloff et al. (2015), with the eastern basin including the Ionian, Levantine, Adriatic and Aegean subbasins. Values in parenthesis indicate the percentage difference from the 1980–1999 period.

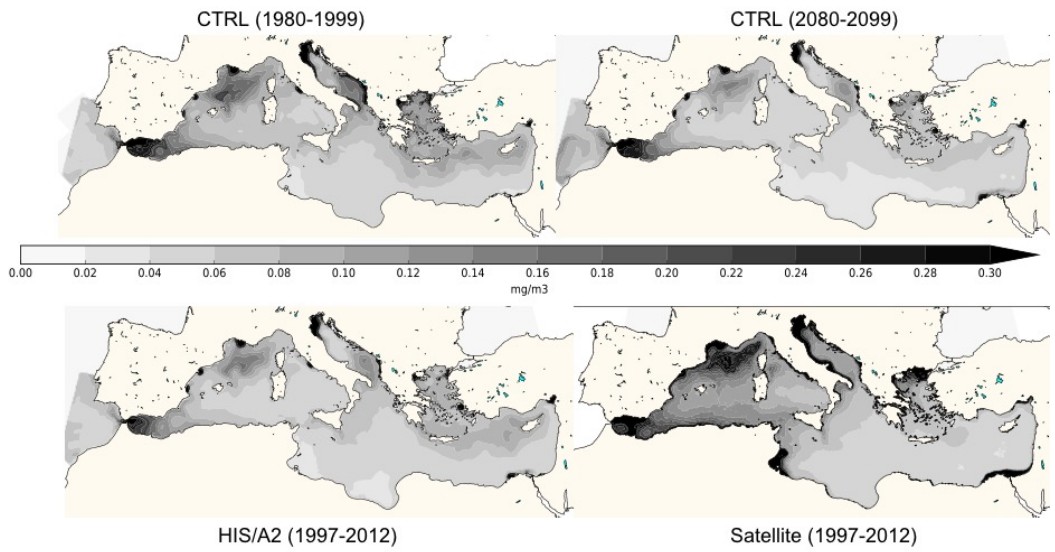

**Figure 1.** Average surface chl-a concentration from the CTRL (top, left: 1980–1999 right: 2080–2099) and HIS/A2 (bottom left) simulations, and from satellite estimations (MyOcean Dataset 1997–2012, bottom right).

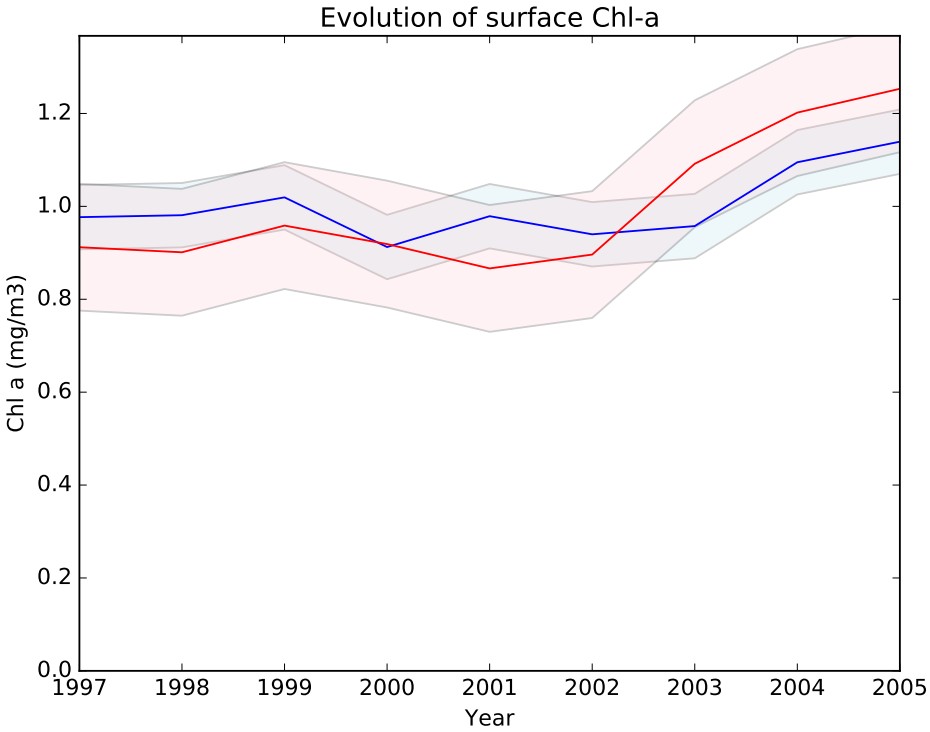

**Figure 2.** Average surface chl-a concentration from the HIS/A2 simulation in blue and from the SeaWiFs satellite data (Bosc et al., 2004) in red over the period 1997–2005. Shaded colors represent the standard deviations. Values are normalized by dividing by the average cholorphyll concentration over the period.

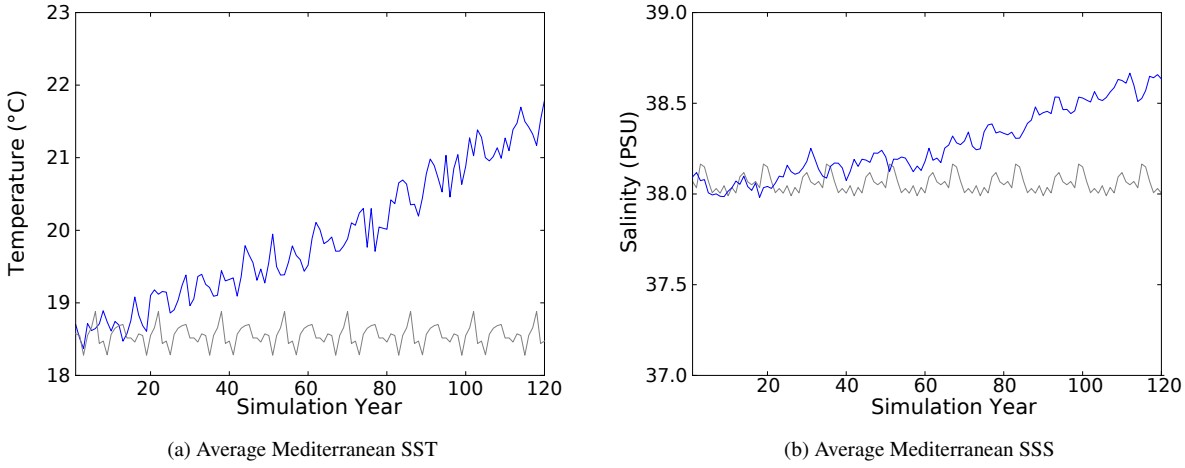

(a) Average Mediterranean SST  (b) Average Mediterranean SSS

**Figure 3.** Evolution of average Mediterranean SST (left) and SSS (right) in the CTRL (grey) and HIS/A2 (blue) simulations.

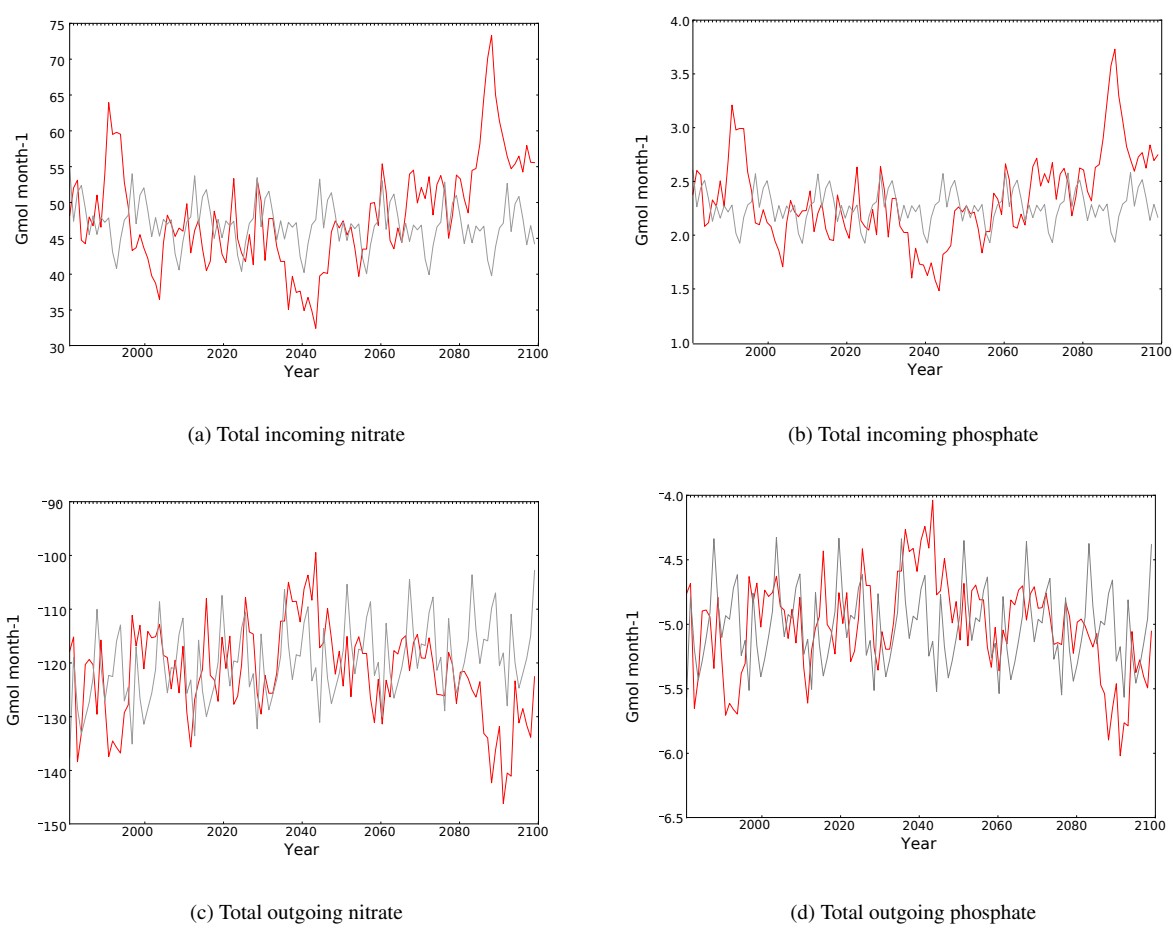

(a) Total incoming nitrate

(b) Total incoming phosphate

(c) Total outgoing nitrate

(d) Total outgoing phosphate

**Figure 4.** Evolution of total incoming (top) and outgoing (bottom) fluxes of nitrate and phosphate ($10^9$ mol month$^{-1}$) through the Strait of Gibraltar in the CTRL (grey) and HIS/A2 (red) simulations. Negative values indicate outgoing fluxes of nutrients.

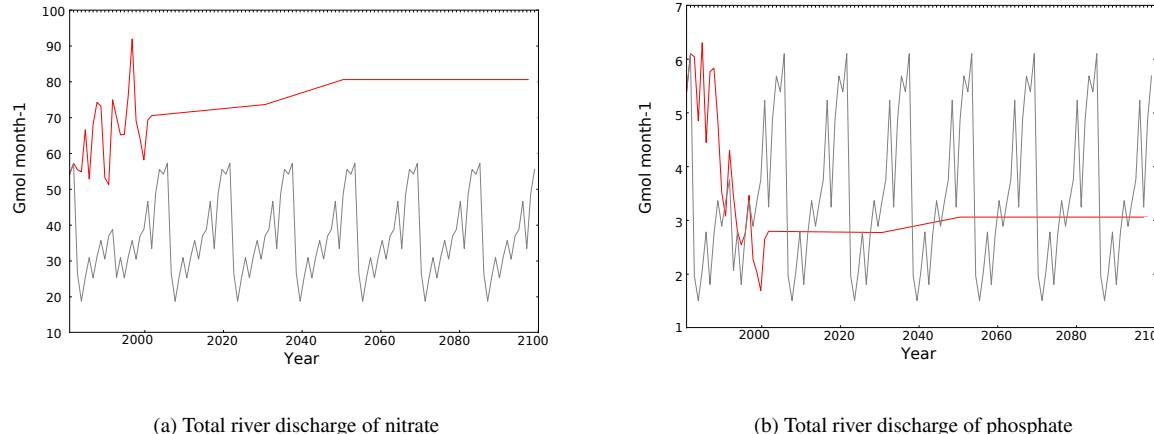

(a) Total river discharge of nitrate

(b) Total river discharge of phosphate

**Figure 5.** Evolution of total river discharge fluxes of nitrate and phosphate ($10^9$mol month$^{-1}$) to the Mediterranean Sea in the CTRL (grey) and HIS/A2 (red) simulations.

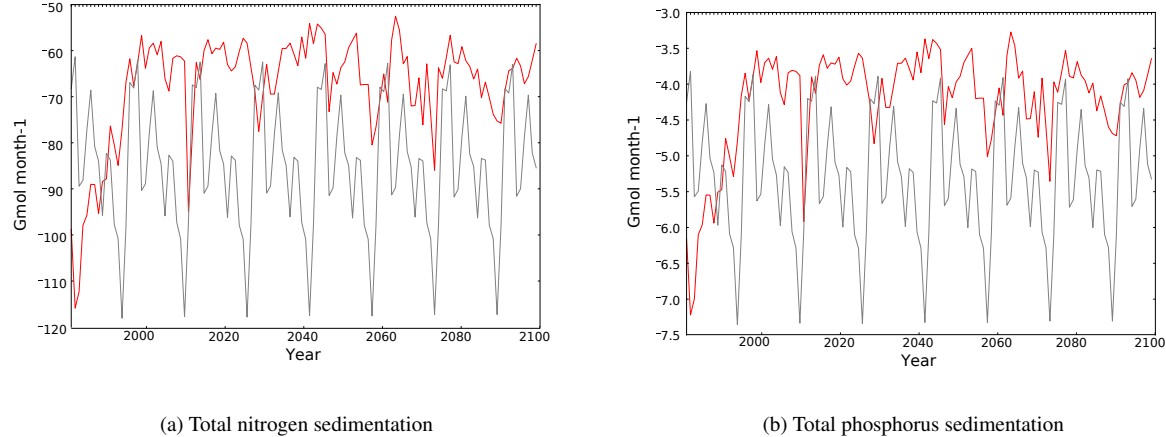

(a) Total nitrogen sedimentation

(b) Total phosphorus sedimentation

**Figure 6.** Evolution of total sedimentation fluxes of N and P ($10^9$mol month$^{-1}$) in the Mediterranean Sea in the CTRL (grey) and HIS/A2 (red) simulations. Negative fluxes indicate that the nutrients are exiting the Mediterranean waters.

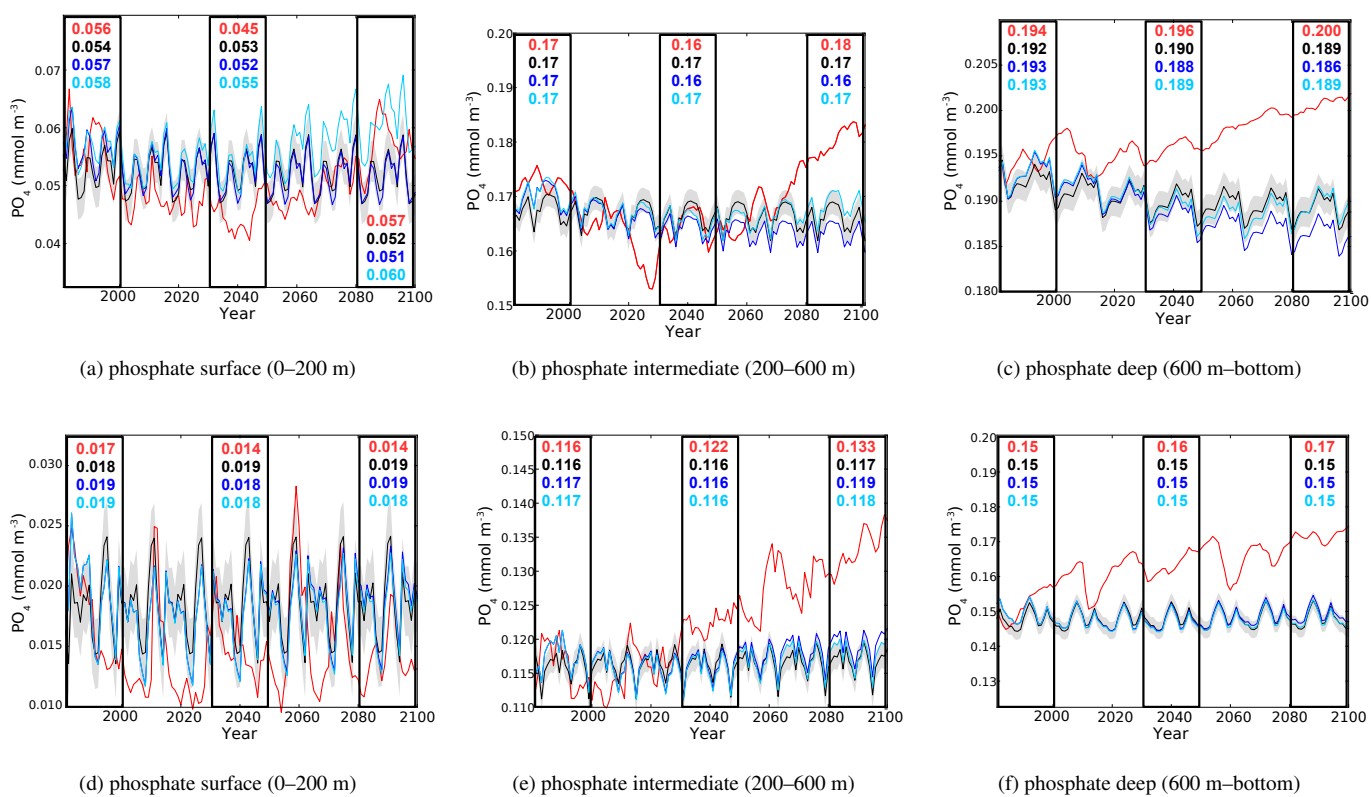

**Figure 7.** Evolution of yearly average phosphate concentration ($10^{-3}$ mol m$^{-3}$) in the surface (left), intermediate (middle) and bottom (right) layers in the western (top) and eastern (bottom) basin. Red lines represent the HIS/A2 simulation, black lines represent the CTRL (with standard deviation), blue and light blue lines represent the CTRL_R and CTRL_RG simulations respectively. Colored numbers in the highlighted areas represent the average concentrations in the corresponding simulations for the highlighted time periods.

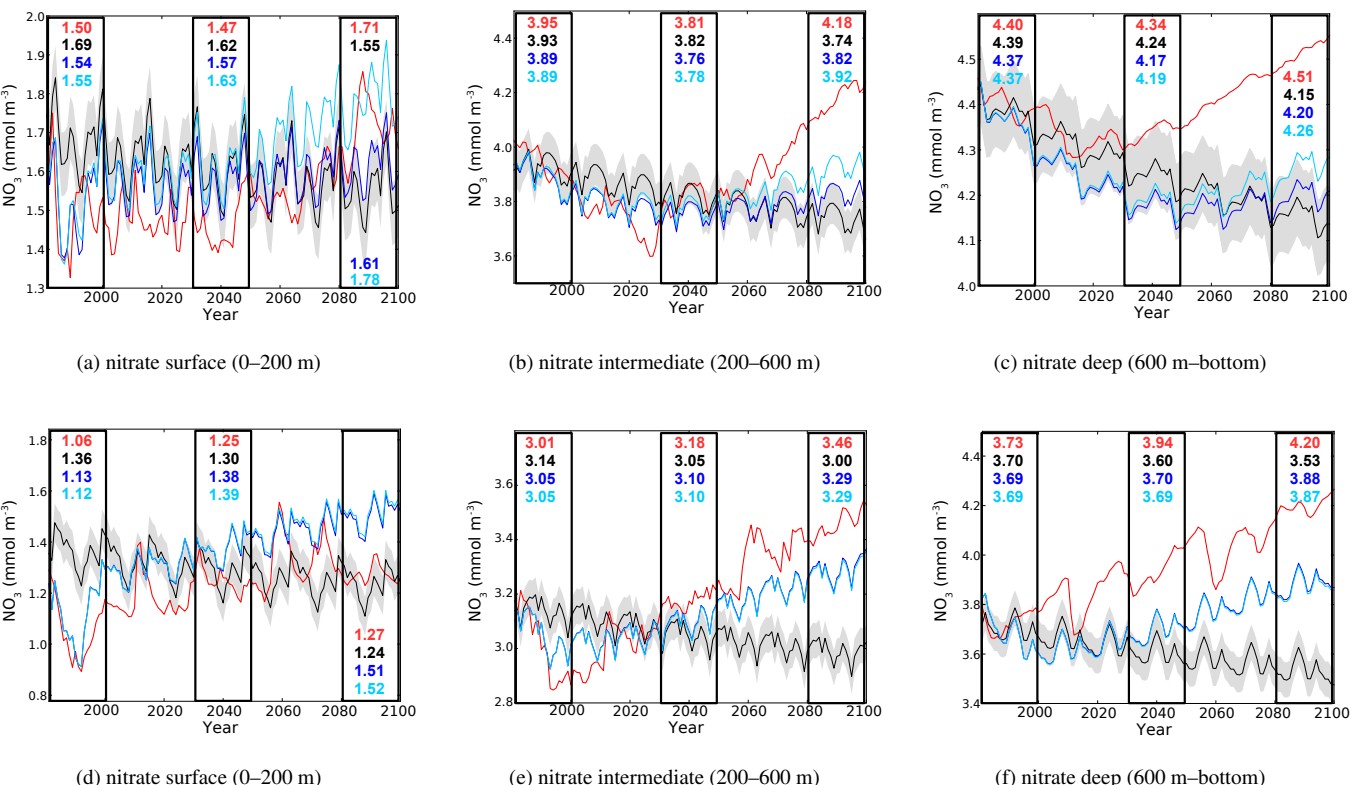

**Figure 8.** Evolution of yearly average nitrate concentration ($10^{-3}$ mol m$^{-3}$) in the surface (left), intermediate (middle) and bottom (right) layers in the western (top) and eastern (bottom) basins. Red lines represent the HIS/A2 simulation, black lines represent the CTRL (with standard deviation), blue and light blue lines represent the CTRL_R and CTRL_RG simulations respectively. Colored numbers in the highlighted areas represent the average concentrations in the corresponding simulations for the highlighted time periods.

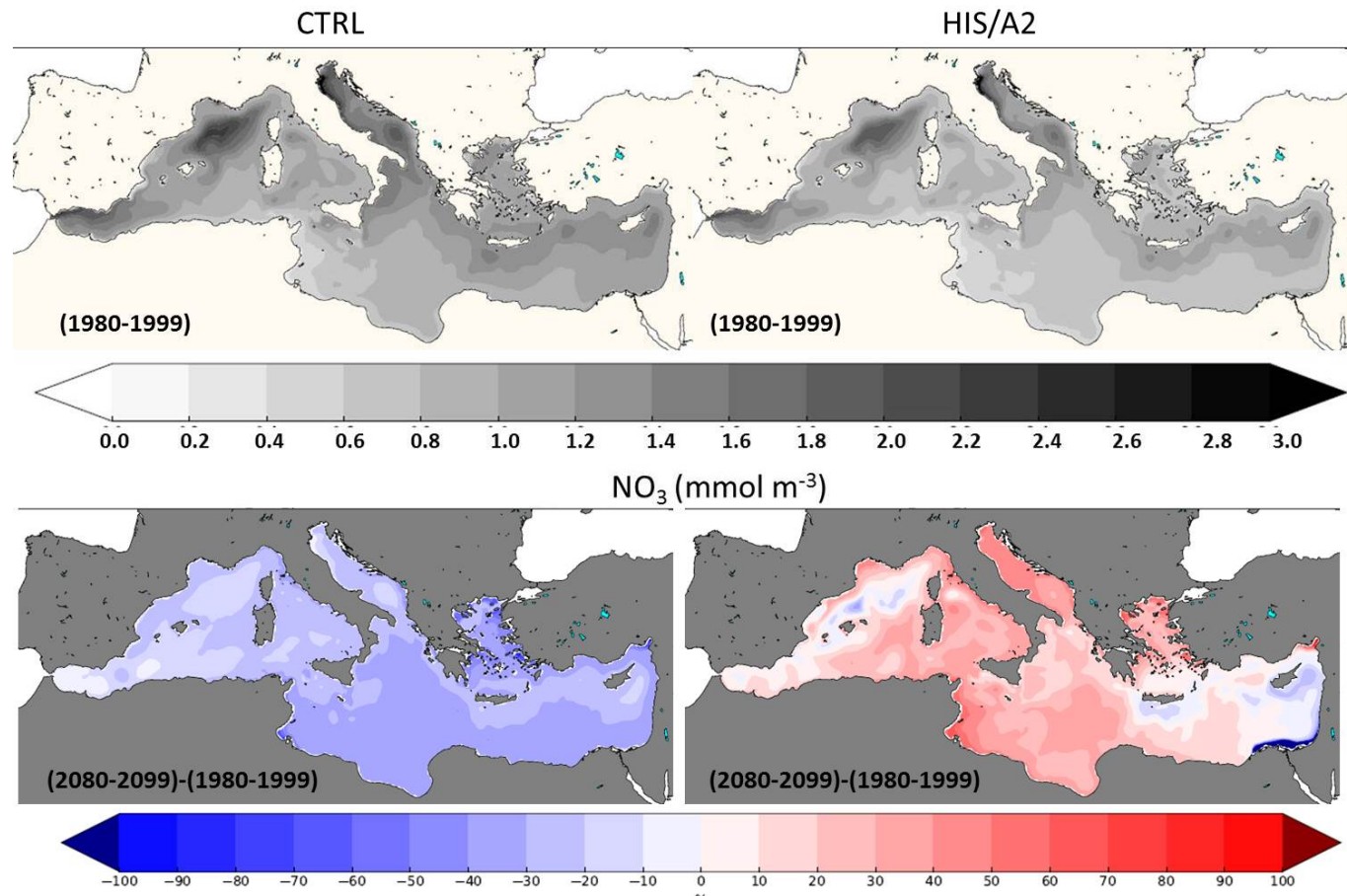

**Figure 9.** Present (1980–1999, top) interannual average surface (0–200 m) concentrations of nitrate ($10^{-3}$mol m$^{-3}$) in the CTRL (left) HIS/A2 (right) simulations. The bottom maps show the percent relative difference between the 2080–2099 and the 1980–1999 periods in the CTRL (left) and HIS/A2 (right) simulations.

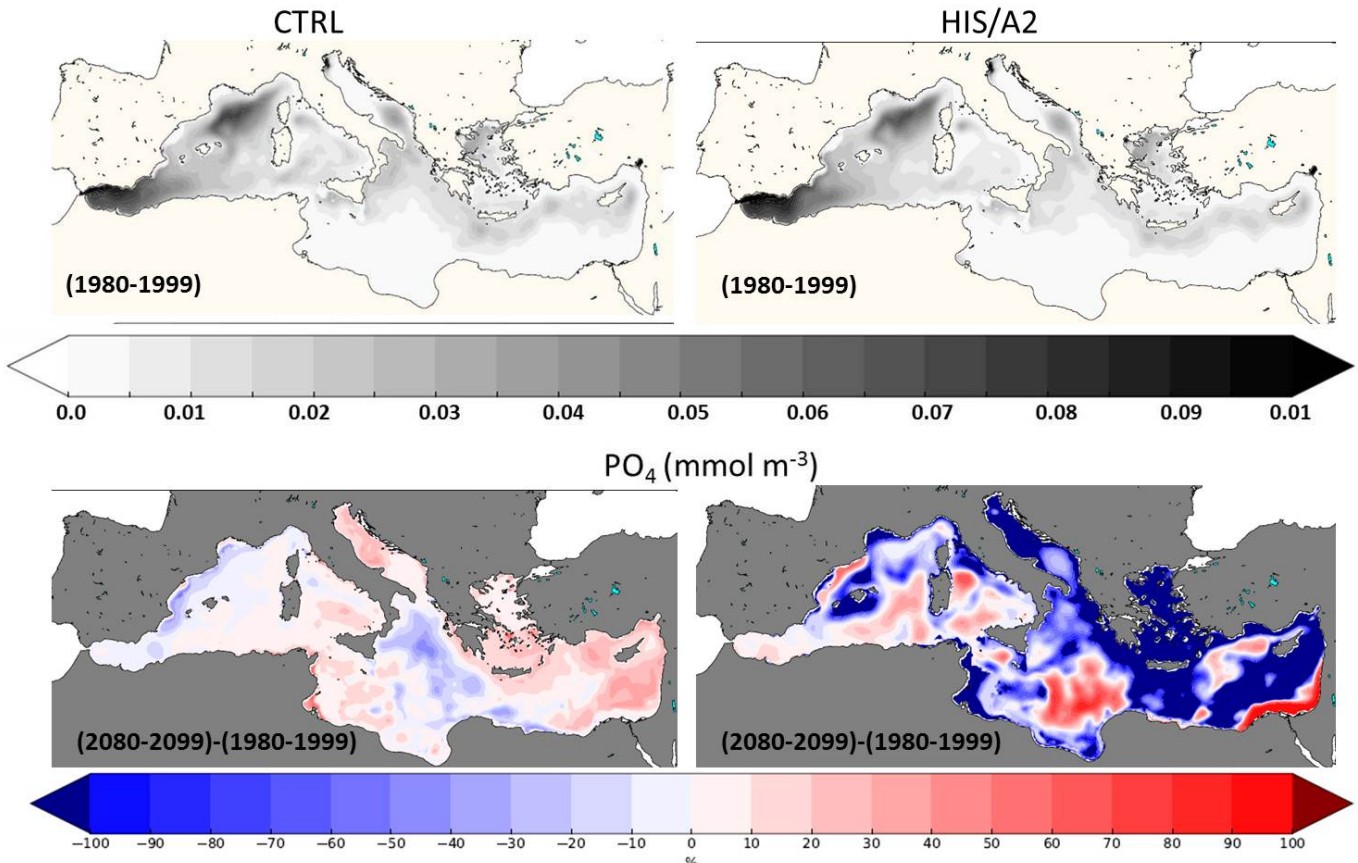

**Figure 10.** Present (1980–1999, top) interannual average surface (0–200m) concentrations of phosphate ($10^{-3}$ mol m$^{-3}$) in the CTRL (left) and HIS/A2 (right) simulations. The bottom maps show the percent relative difference in primary production between the 2080–2099 and the 1980–1999 periods in the CTRL (left) and HIS/A2 (right) simulations.

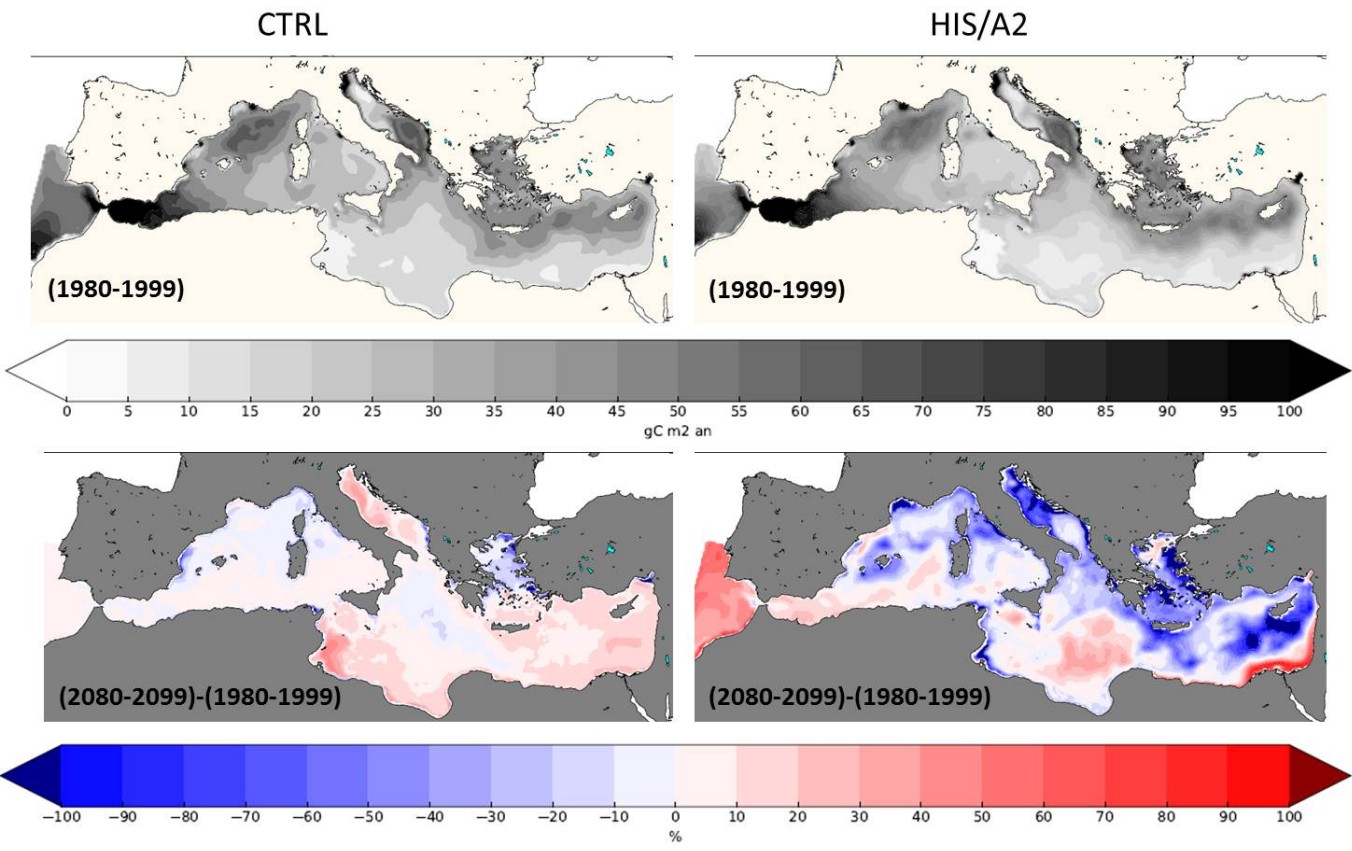

**Figure 11.** Present (1980–1999, top) interannual average surface (0–200 m) integrated primary production (gC m$^{-2}$) in the CTRL (left) and HIS/A2 (right) simulations. The bottom maps show the percent relative difference between the 2080–2099 and the 1980–1999 periods in the CTRL (left) and HIS/A2 (right) simulations.

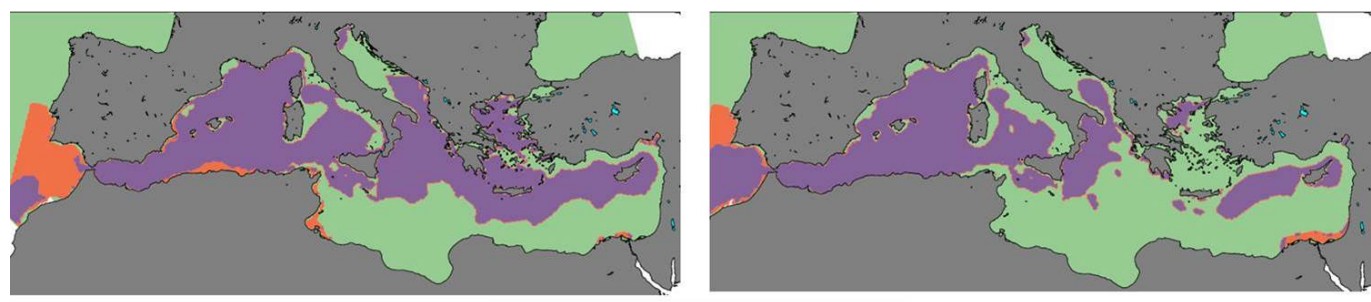

**Figure 12.** Present (1980–1999) and future (2080-2099) interannual average surface (0–200 m) limiting nutrient in the HIS/A2 simulation. N and P colimitation is considered when limitation factors for N and P differ by less than 1 %. Green zones are P–limited, Orange zones are N–limited and purple zones are N and P co–limited.

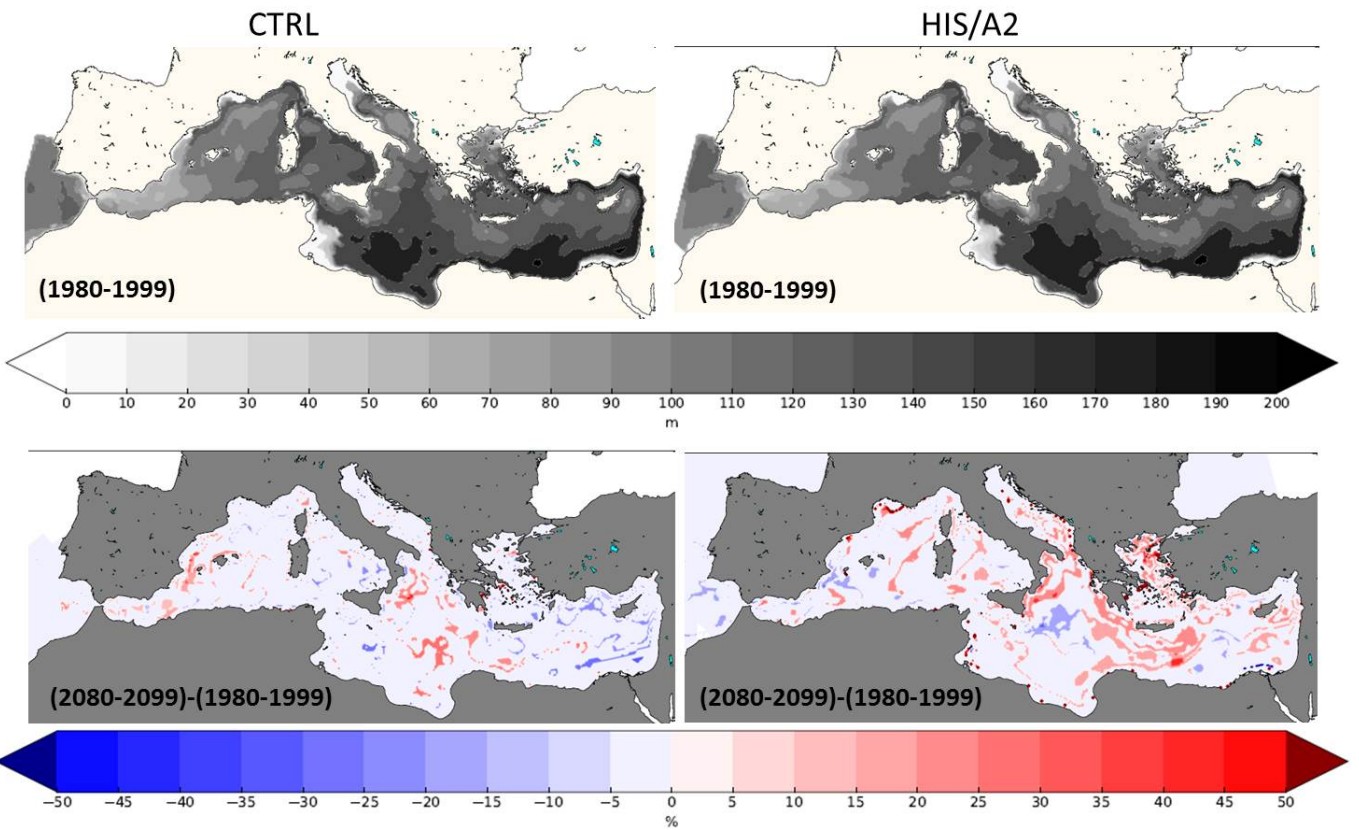

**Figure 13.** Present (1980–1999, top) interannual average DCM (m) in the CTRL (left) and HIS/A2 (right) simulations. The bottom maps show the percent relative difference in DCM between the 2080–2099 and the 1980–1999 periods in the CTRL (left) and HIS/A2 (right) simulations.

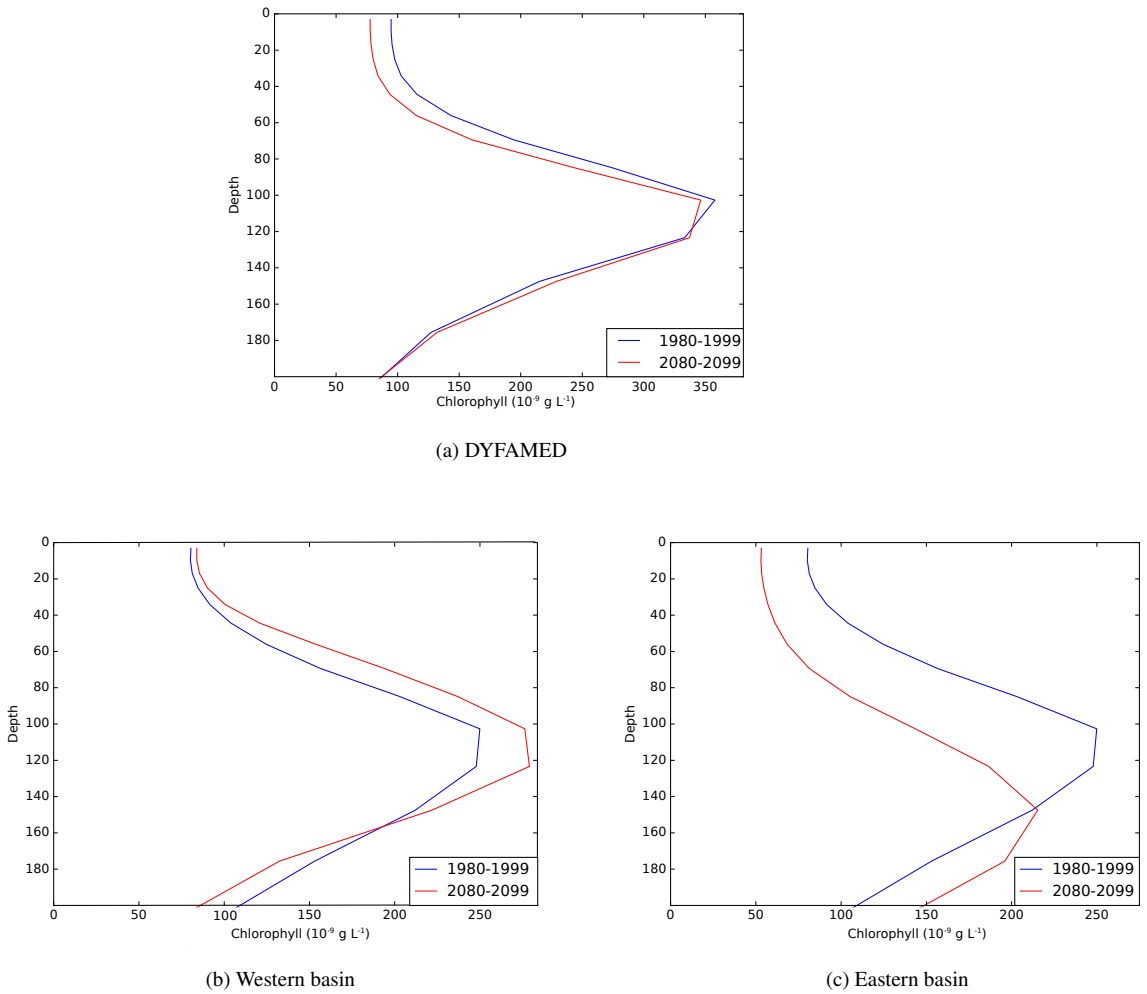

(a) DYFAMED

(b) Western basin

(c) Eastern basin

**Figure 14.** Present (1980–1999) and future (2080–2099) interannual average vertical profiles of total chl-a *a* (ng L$^{-1}$) at the DYFAMED station and averaged profiles over the western and eastern (including Aegean and Adriatic) basins.

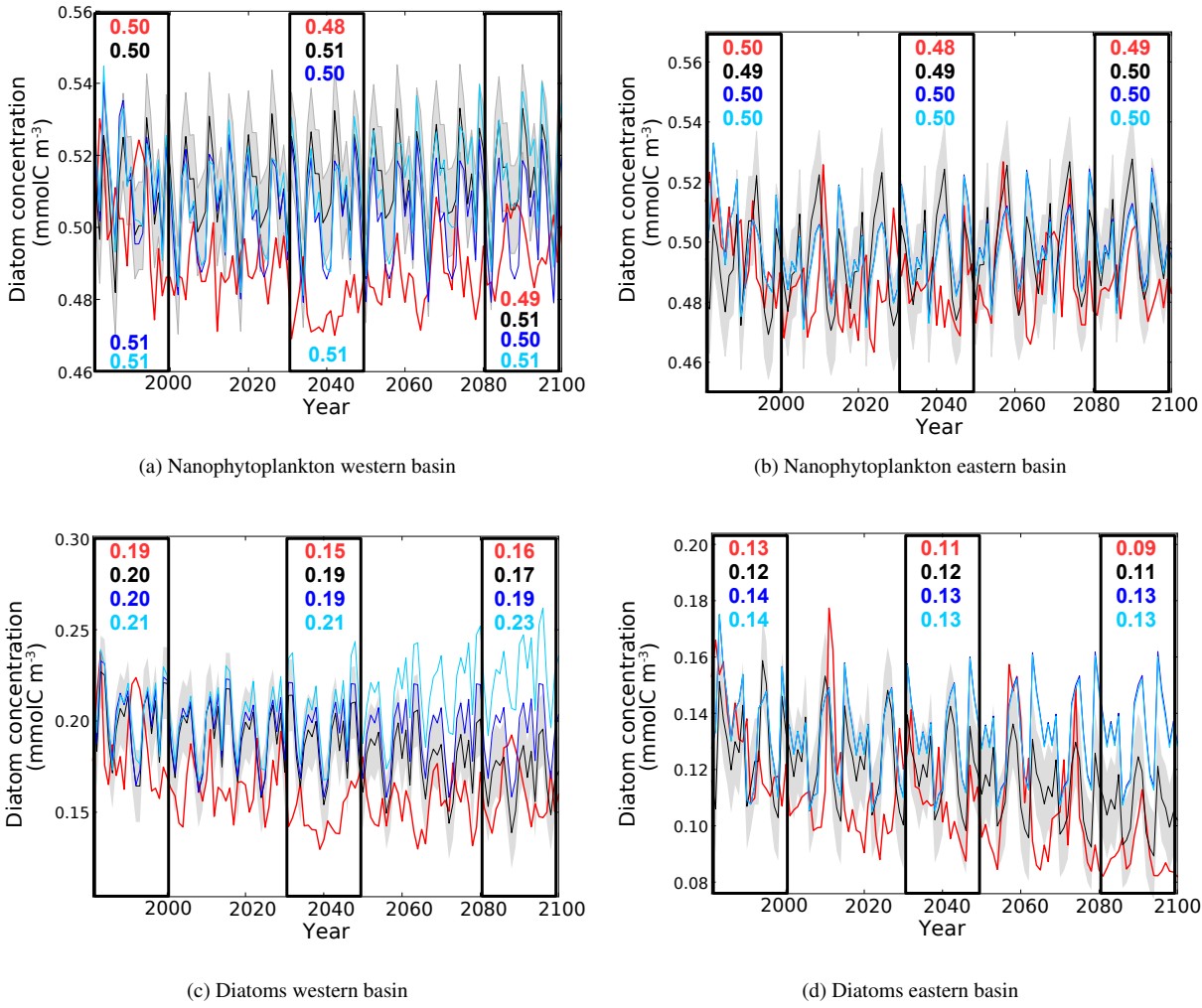

(a) Nanophytoplankton western basin

(b) Nanophytoplankton eastern basin

(c) Diatoms western basin

(d) Diatoms eastern basin

**Figure 15.** Evolution of yearly average nanophytoplankton and diatoms concentration ($10^{-3}$mol m$^{-3}$) in the surface layer of the western and eastern basin. The red line represent the HIS/A2 simulation, the black lines represent the CTRL simulation (with standard deviation), blue and light blue lines represent the CTRL_R and CTRL_RG simulations respectively. Colored numbers in the highlighted areas represent the average concentrations in the corresponding simulations for the highlighted time periods.

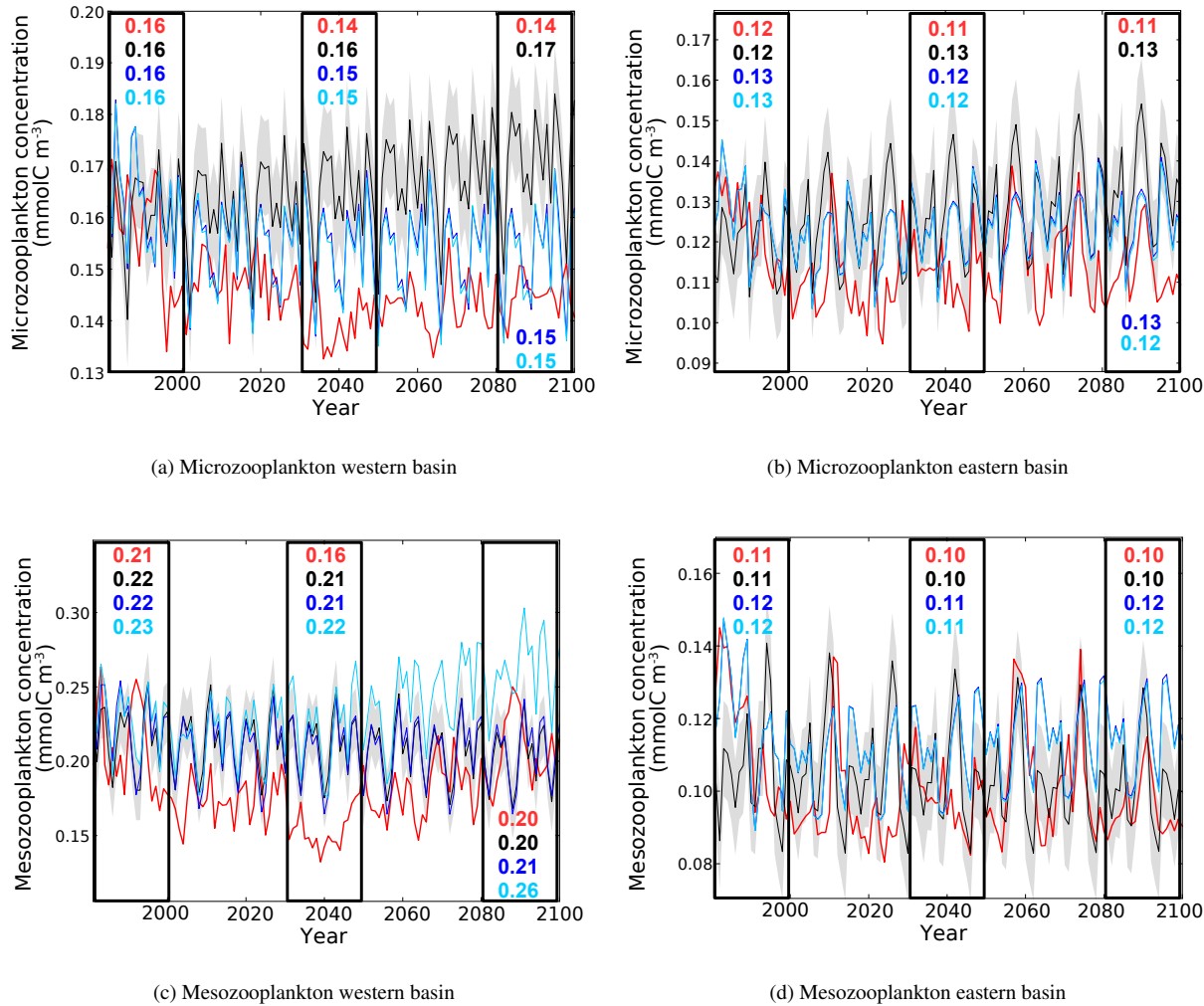

(a) Microzooplankton western basin

(b) Microzooplankton eastern basin

(c) Mesozooplankton western basin

(d) Mesozooplankton eastern basin

**Figure 16.** Evolution of yearly average microzooplankton and mesozooplankton concentrations ($10^{-3}$mol m$^{-3}$) in the surface layer of the western and eastern basins. Red lines represent the HIS/A2 simulation, black lines represent the CTRL simulation (with standard deviation), blue and light blue lines represent the CTRL_R and CTRL_RG simulations respectively. Colored numbers in the highlighted areas represent the average concentrations in the corresponding simulations for the corresponding time periods.

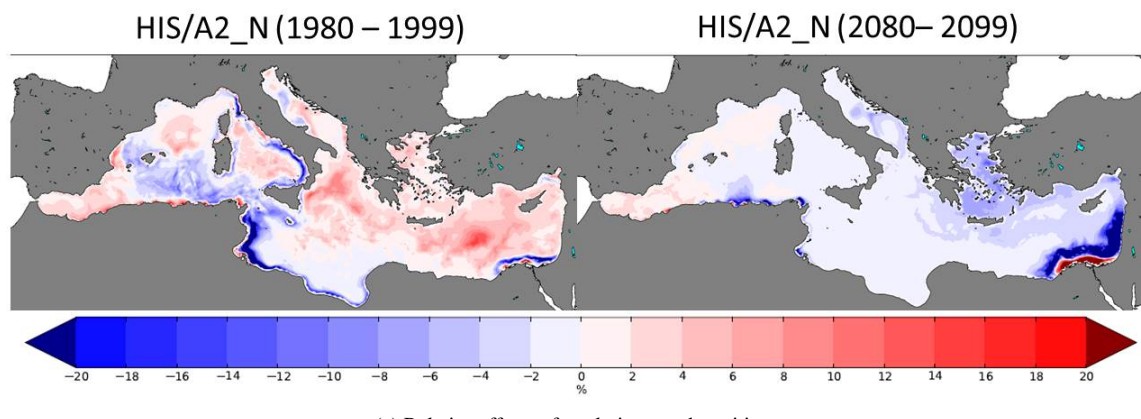

(a) Relative effects of total nitrogen deposition

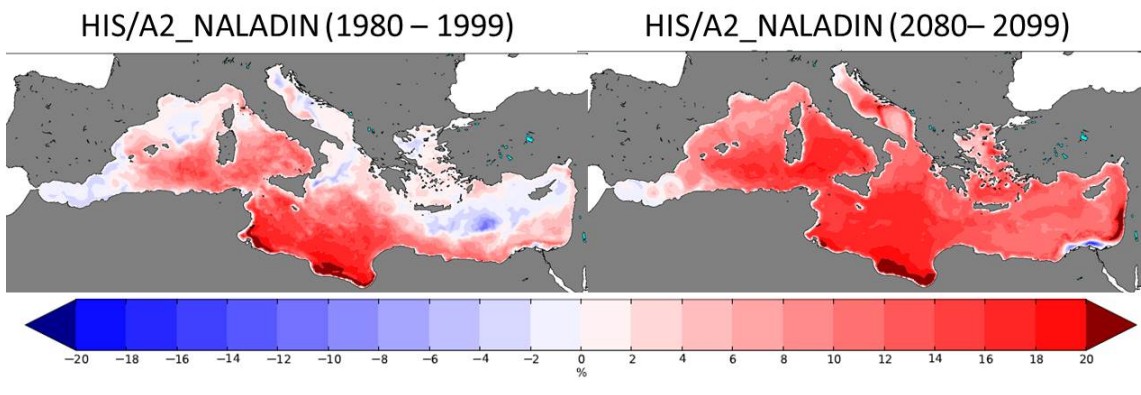

(b) Relative effects of natural dust deposition

**Figure 17.** Present (1980–1999) and future (2080–2099) relative effects of total nitrogen (top) and natural dust (bottom) deposition on surface (0–10 m) total primary production.

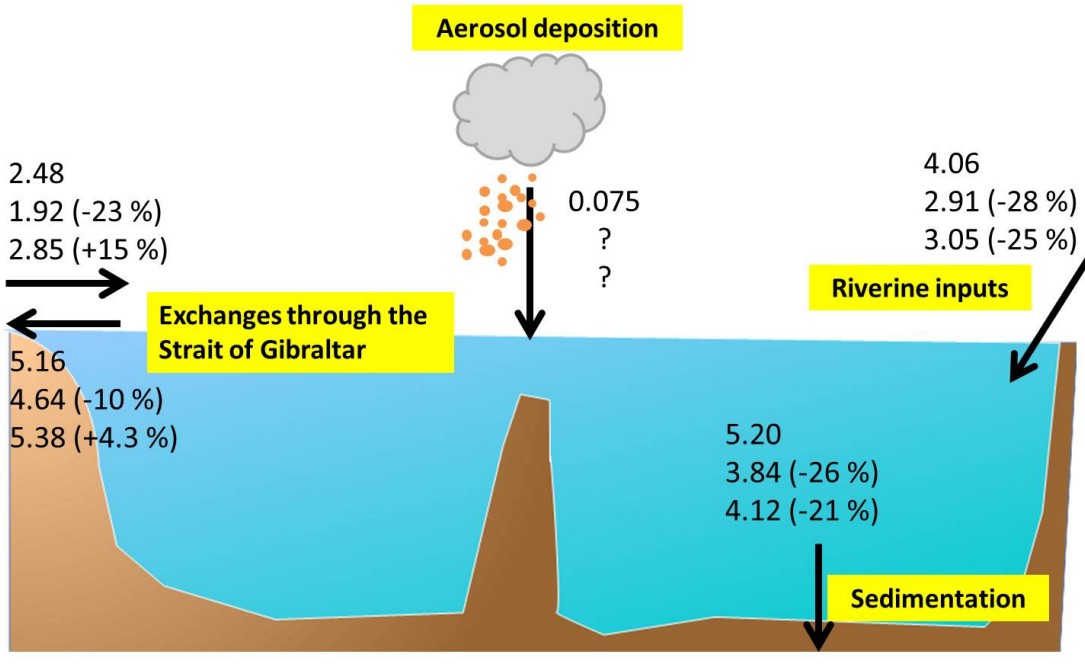

(a) Phosphate fluxes

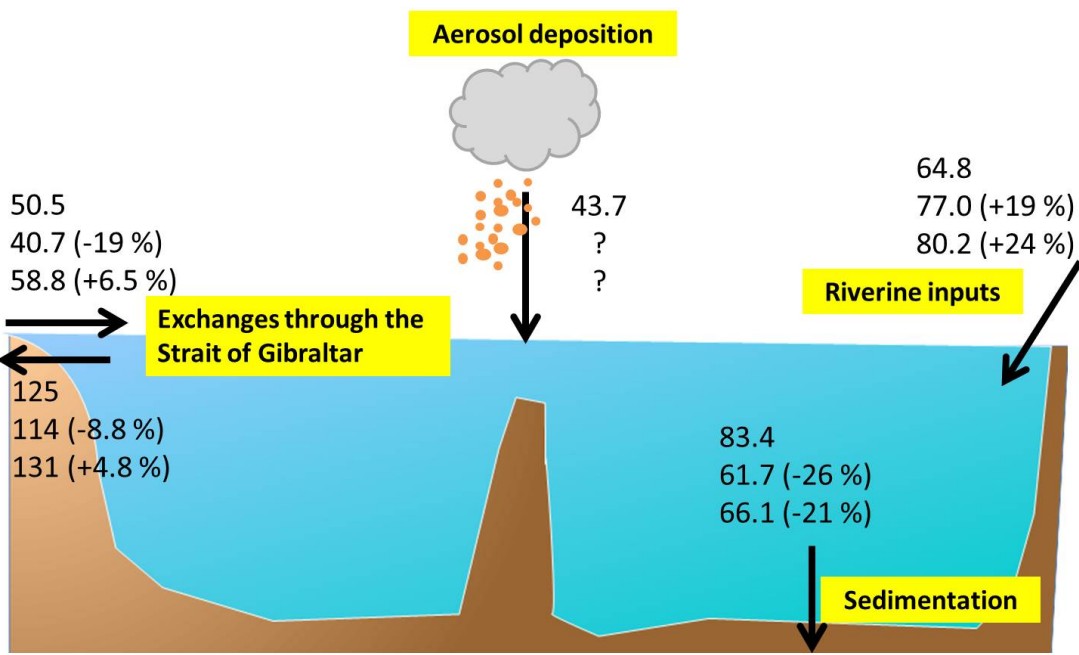

(b) Nitrate fluxes

**Figure 18.** Schematic diagrams illustrating the Mediterranean budgets of phosphate and nitrate. For each component, the 3 lines represent the average fluxes (in Gmol year$^{-1}$) over the periods 1980–1999, 2030-2049 and 2080-2099, numbers in parenthesis indicate the percentage difference from the 1980–1999 values.

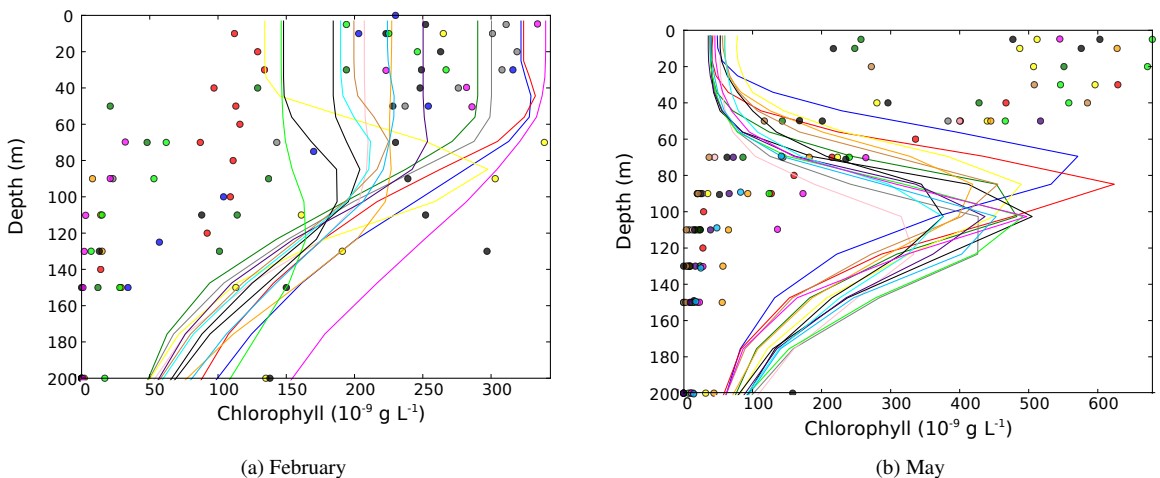

|   |   |
|---|---|
| (a) February | (b) May |

**Figure A1.** Average chl-a–a profiles in February (left) and May (right) for the years 1991 to 2005 at the DY-FAMED station in the Ligurian Sea (43.4277°N, 7.2522°E). Dots represent data points (Marty et al., 2002; Faugeras et al., 2003). Lines represent the HIS/A2 simulation. Colors represent individual years.

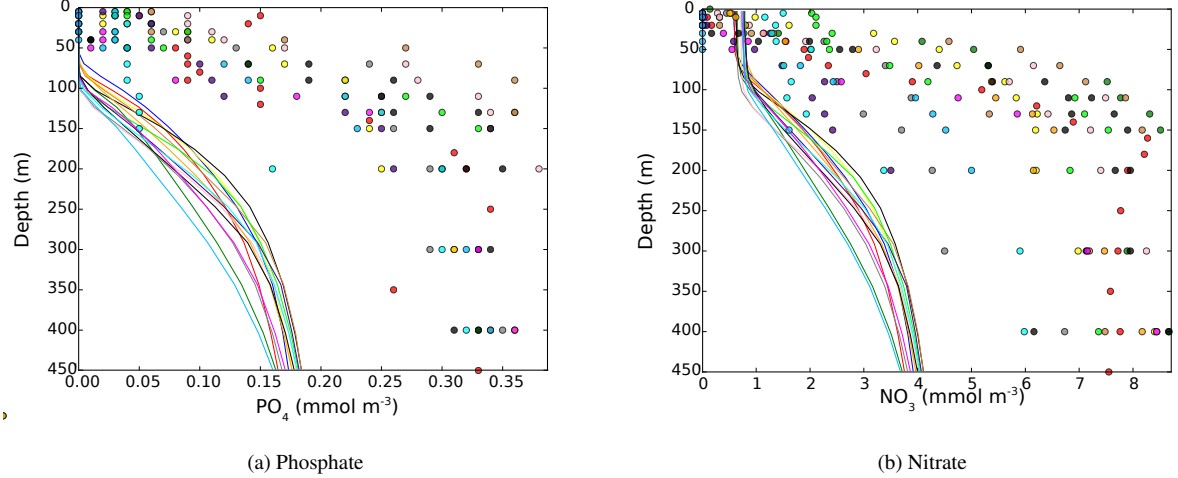

|   |   |
|---|---|
| (a) Phosphate | (b) Nitrate |

**Figure A2.** Average phosphate (left) and nitrate (right) profiles in May for the years 1991 to 2005 at the DY-FAMED station in the Ligurian Sea (43.4277°N, 7.2522°E). Dots represent data points (Marty et al., 2002; Faugeras et al., 2003). Lines represent the HIS/A2 simulation. Colors represent individual years.

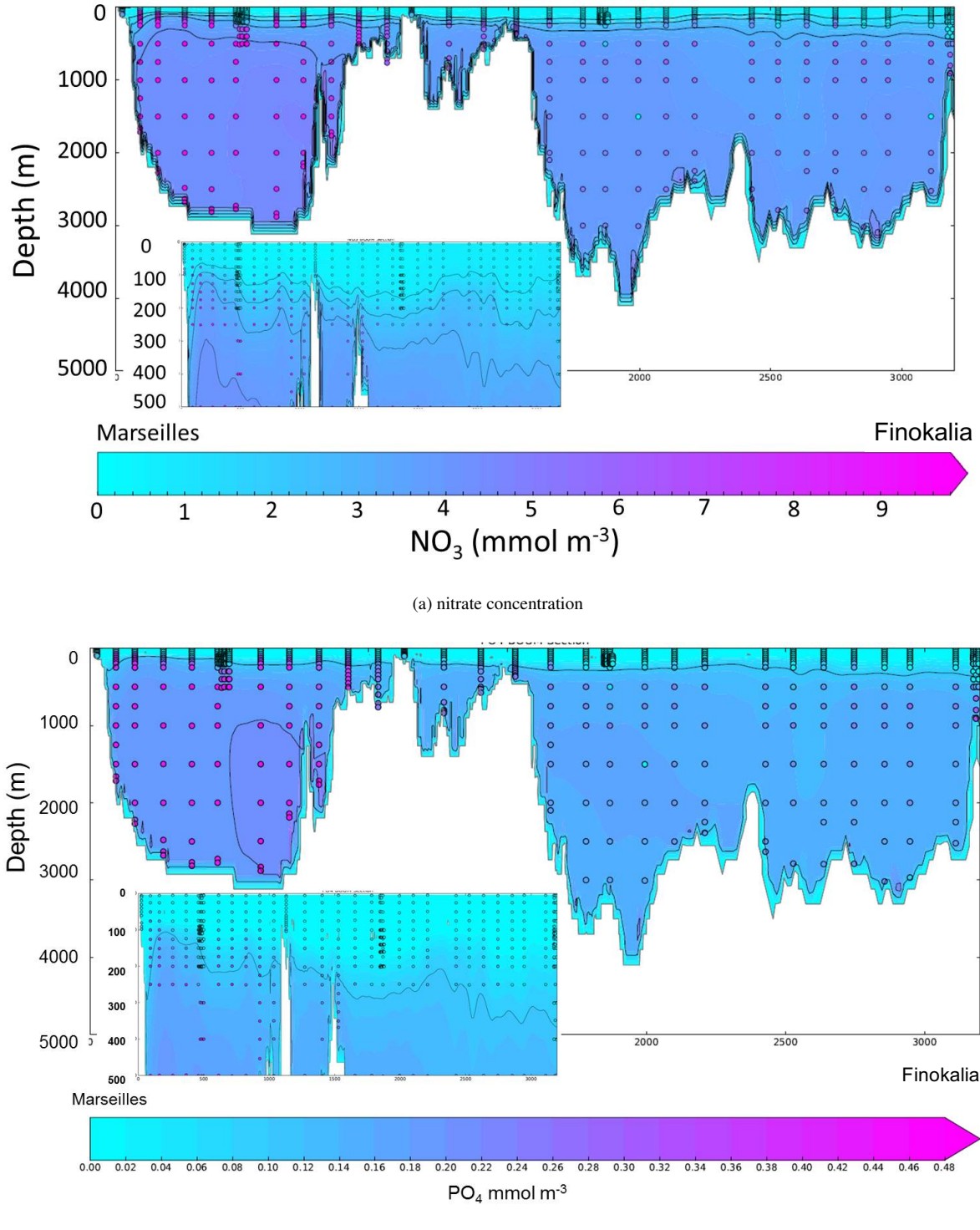

(a) nitrate concentration

(b) PO$_4$ concentration

**Figure A3.** Average concentrations of nitrate (top) and phosphate (bottom) for the 20 first years of the control simulation (CTRL). The dots represent data along a transect from Marseille to Finokalia from the BOUM campaign (distances in km Moutin et al., 2012). The framed areas represent a vertical zoom of the top 500 m along the whole transect.