# Peer review of "Biogeochemical response of the Mediterranean Sea to the transient SRES-A2 climate change scenario"

_Biogeosciences, 2018_

## Referee Comment (RC1) · Anonymous Referee #1 · 15 Jun 2018

Richon et al. analyzed the biogeochemical response of the Mediterranean Sea to the transient SRES–A2 climate change scenario, by using the coupled high resolution model NEMOMED8/PISCES. The main objective is to quantify separately the effects of biogeochemical forcings (external nutrient inputs, i.e., coastal runoff, river discharge and Atlantic inputs through Gibraltar) and climate change (which influences physical processes, e.g., deep convection and thermohaline circulation, vertical mixing and stratification…) on the Mediterranean biogeochemistry. This type of analysis is crucial to predict the response of the Mediterranean Sea to the climate change, because, as mentioned by the authors, this semi–enclosed oligotrophic basin with short residence time of water masses is particularly responsive to climate change and highly sensitive to external nutrient inputs.

Authors point out that by the end of the simulated period (2090s) there is a reduction in the phytoplankton production, mainly in eastern regions. The changes in surface nutrient concentrations are weaker in the western basin because of the strong regulating impact of Gibraltar nutrient exchange, and nutrient runoff and river discharges influence mostly coastal regions of the Mediterranean Sea.

The approach is interesting and can provide significant scientific advances. However, I have concerns on the way the analysis has been conducted and presented in the paper. In addition, the manuscript presented major mistakes and structural issues that made the reading and the review complicated. Therefore, the paper will likely be a significant scientific contribution with major revisions.

The major revisions that are required are summarized below in my major comments, followed by a non-exhaustive list of corrections.

Major comments:

**1)** Line 68 - "Being a semi–enclosed oligotrophic basin, the Mediterranean is highly sensitive to external nutrient inputs. Their origins are mainly from coastal runoff, river discharge (Ludwig et al., 2009), Atlantic inputs through Gibraltar (Gómez, 2003), and atmospheric deposition (Richon et al., 2017, 2018)."

Line 143 - "External nutrient supply for the biogeochemical model include inputs from the Atlantic Ocean and from Mediterranean rivers."

Line 501 – "No atmospheric deposition was considered in this study because there is, to our knowledge, no transient scenario for atmospheric deposition evolution over the Mediterranean Sea."

Line 598 - "Finally, this study accounts for the changes in all external biogeochemical forcings except atmospheric deposition. However, Richon et al. (2017, 2018) showed that atmospheric deposition can account for up to 80 % of phosphate supply in some Mediterranean Sea regions and has significant impacts on surface productivity."

As mentioned by the authors, the atmospheric deposition have a significant impact on the Mediterranean Sea. The first author quoted two of its own papers to support this. Even if no transient scenario for atmospheric deposition exists, did the model contain a present-day atmospheric deposition component? (As for example, the model analysis of Herrmann et al. (2014) and Macias et al. (2015) that used continued present–day discharge of nutrients)

If yes, the influence of the atmospheric depositions should be include in the discussion of the results. If not, I suggest use of continued present–day atmospheric depositions in the model.

**2)** Line 174 - "The control run CTRL is performed with present–day conditions forcing (1966–1981). The scenario simulation is referred to as HIS/A2 as in Adloff et al. (2015). HIS is the name of the historical period (in our case between 1980 and 1999), and A2 is the name of the 2000–2099 scenario simulation."

The present-day period (1966–81) cannot be older than the historical period (1980-99). Therefore, the CTRL simulations does not correspond to the present–day conditions, and because condition forcing between the periods 1966–81 and 1980-99 are different, results from the control simulation (CTRL) differed from those of the scenario simulation (HIS/A2) during the first simulated decades (from 1980 to now). Authors need to justify and clarify this choice.

**3)** The section 3.1 "Evaluation of the NEMOMED8/PISCES mode"

- "The model correctly reproduces the main high–chlorophyll regions such as the Gulf of Lions and"

- "The west–to–east gradient of productivity is also reproduced by the model with values that agree with satellite estimates."

- "In spite of some underestimation of nutrient concentrations that are probably linked with the features of the simulated intermediate and deep waters characteristics"

-"The average chlorophyll concentration observed at the DYFAMED station in the top 200 m is $227 \pm 136 \ 10^{-9} \ \mathrm{g \ L^{-1}}$ (average over the 1991–2005 period), while the model value for the HIS period is $173 \pm 150 \ 10^{-9} \ \mathrm{g \ L^{-1}}$."

-"the PISCES model reproduces the main characteristics of the Mediterranean biogeochemistry, including a salient west–to–east gradient in nutrient concentrations, low surface nutrient concentrations and a deep chlorophyll maximum (DCM)."

In this section, the comparison of the model results with in situ data have been incorrectly conducted. The main issue is that no values to support the comparison between the model and the in situ data are provided (e.g., correlation coefficients, percentage of

differences…). For example, the chlorophyll-a concentration in the Gulf of Lion seems two times lower in the model simulations than the estimates from satellite.

There is no information on the spatial variability of the nutricline depths (i.e., nitracline and phosphacline), and of the DCM.

The figure 1, associated with this section, compares data from the satellite and the model results during two different periods 1980-99 and 1997-2012.

The units are not coherent:

- "1 K colder than observations", temperature in Kelvin?
- Figure A1 – "Chla $10^{-9}$ g m$^{-3}$", in the text: "227 ± 136 $10^{-9}$ g L$^{-1}$", not consistent between them, and in the literature the most common unit is mg/m3 (or microg/L).

Maybe, model values do not agree well with the in situ data, but spatial and temporal variabilities that exist between the different Mediterranean regions have to be simulated by the model. Unfortunately, quantitative information to support this hypothesis are not provided by the authors.

**4)** In the introduction, I understand that the authors want to study the effects of biogeochemical forcings and climate change on,

a- the nitrate and phosphate concentrations in the Mediterranean Sea.
b- the phytoplankton production.
c- the nutrient limitation (nitrate and phosphate).

But, the results are not well presented and discussed.

- Most of the time, there is no quantitative estimates associated with the words "increase", "decrease", "substantial"…

- Figures and tables have to be improve:
  o need to adjust the axes,
  o units are wrong,
  o it is better to represent the anomalies when you compare the model with data (or between two periods) by using a map.

- There are major mistakes, for example: authors provided values for chlorophyll concentration instead of Net Primary Production…

- There is a lack of references in the discussion.

Below, I review sections of the manuscript mainly associated with the first objective of the manuscript, (a) the effects of biogeochemical forcings and climate change on the nitrate and phosphate concentrations in the Mediterranean Sea. I cannot provide a review for

the other two objectives (b and c) because they required a better analysis and discussion of the results associated with this first objective (a).

**Section 3.3.1 Evolution of phosphate and nitrate concentrations**

Figure 3-4 – Adjust the y-axis, it is impossible to evaluate the results.

Line 259 – "A slight accumulation of phosphate is observed in the deep western basin" - For which simulations? Provide values.

Line 268 – "The evolution of nitrate concentration shows a marked accumulation over the century in all regions of the intermediate and deep Mediterranean waters" - For which simulations?

Line 279-289 – Confusing, mixing general results (for both nitrate and phosphate) with results specific to the nitrate that should have been present in previous paragraphs.

This sections need to be clearer. Stay with the same logic when you present your results. Compare CTRL with HIS/A2, western basin with eastern basin, depth by depth…

**3.3.2 Exchange fluxes of nutrients at Gibraltar**

Figure 5 – Keep the same x-axis as in the figures 3 and 4.

Line 293 – "We observe similar trends in phosphate and nitrate fluxes linked to the Redfieldian behavior of the primary production in PISCES." - What do you mean, where can we see this?

Line 295 – "the incoming fluxes decrease" - fluxes of what?

Line 295 - "According to the HIS/A2 simulation, the incoming fluxes decrease slightly until the middle of the century and then increase to reach values higher than the control in the last 25 years of simulations. Outgoing fluxes follow the same trends as incoming fluxes" – For the incoming fluxes, I see, a peak in the 90s, then stable incoming fluxes until a decrease in the 2030s, and then an increase in the last 25 years with a peak in the 2080s. For outgoing fluxes, I see, a slight increase in the first half of the 21st century and a decrease after. Not you?

Line 298 – "We observe a drift in the nitrate outgoing flux in the control." – Provide a value

Line 305 – "Figures 3a and 5b show that the evolution of phosphate concentration in the western basin is linked with Gibraltar inputs (Pearson's correlation coefficient is 0.63, p–value=10−14)" - Correlation between what and what, surface, intermediate or deep concentration of phosphate?

**Section 3.3.3 River fluxes of nutrients**

Figure 6 – Keep the same x-axis as in the figures 3 and 4.

All tables – In the result section you only wrote in percentages. Therefore, provide percentages values in tables.

Line 311 – "River discharge is the main external source of nutrient for the eastern part of the basin." – Need references.

Line 315 – "Nitrate discharge in the HIS/A2 simulation is significantly higher than in CTRL" – How much? Provide a value.

Line 315 – "nitrate total discharge in the Mediterranean has continuously increased from the 1960s (see the CTRL values for the years 1966–1981)." - What was the value in the 1960s? The model simulations start in 1980.

I see that there is no internannual variability in the HIS/A2 simulations. You have to say something about it. Phosphate concentrations mainly decrease between 1980 and 2000. Why? Nitrate and Phosphate concentrations mainly increase between 2030 and 2050. Why?

**Section 4.4 Climate versus biogeochemical forcing effects**

Line 559 – "They found a general decrease in plankton biomass that is lower than in our severe climate change scenario". – Provide a value.

Table 4 – "Simulated integrated chlorophyll production ($10^9$ mol)"

Line 564 – "Results from Herrmann et al. (2014) indicate that chlorophyll production" – Chlorophyll production? Are you sure... I think you want to study Primary Production, or Net Primary Production. It is a major mistake…

Line 567 – "may lead to a decrease in chlorophyll and plankton biomass" - Provide values

Figure 14 & 15 – "nanophytoplankton and diatoms concentration ($10^{-3}$ mol m$^{-3}$)" "mesozooplankton concentrations ($10^{-3}$ mol m$^{-3}$)" – A mol of diatoms? A mol of mesozooplancton? Wrong units…

Line 571 – "In particular, nutrient inputs at Gibraltar have substantial consequences on the western basin." – Provide an estimate.

There are only four references in this crucial section (Lazzari et al., 2014; Herrmann et al., 2014; Macias et al., 2015). It is not enough…

List of corrections:

**1)** Line 7 – "socio-economic", you used both socio-economic and socioeconomic in the text, choose the good one.

**2)** Line 10 – "lead to changes in phytoplankton nutrient limitation factors.", which ones?

**3)** Line 26 – "known as sapropels, have been recorded through the last 10 000 years", It is the most recent sapropel events that apparently lasted for 3000 years, other events occurred before. Please clarify.

**4)** Line 33 – "and had biogeochemical impacts", which ones? Where? Need references.

**5)** Line 35 – "The modification of water transport led to modified nutrient distribution that can alter local productivity.", Need references.

**6)** Line 36 – "short residence time of water." How long? Need references.

**7)** Line 38 – "that changes in these conditions can trigger important circulation changes, ultimately leading to changes in", three times the word "change" in the same sentence.

**8)** Line 40 – "The Mediterranean is connected to the global ocean by the narrow Strait of Gibraltar through which transport contributes substantially to its water and nutrient budgets.", Transport of what? The link between the Strait of Gibraltar and the rest of the paragraph is unclear.

**9)** Line 42 to 46 – "Future climate projections yield […] the western basin for greenhouse gases high-emission scenarios and…" Modify, "Future climate projections with greenhouse gases high-emission scenarios…"

**10)** Line 47 – "In one of these MTHC weakening scenarios, Herrmann et al. (2014) show, in addition, a vertical stratification increase (Adloff et al., 2015)." Herrmann et al., 2014 or Adloff et al., 2015?

**11)** Line 52 – "mixing that bring together available nutrients and phytoplankton", not clear.

**12)** Line 65 – "as a result of density changes", not clear, do you mean less stratify?

**13)** Line 74 – "…chlorophyll–a concentrations, plankton biomass…", Chlorophyll-a concentration is a proxy of phytoplankton biomass, please clarify.

**14)** Line 83 – "In section 3.3, we expose the temporal evolution of the main nutrients, their budgets in present and future conditions and discuss their impact on the biogeochemistry of the Mediterranean Sea.", You should discuss your result in the section 4 discussion.

**15)** Line 117 – "by up to 3 K by", temperature in Kelvin scale?

**16)** Line 121 – "0.5 (practical salinity scale)", not in pratical salinity unit?

**17)** Line 127 – "reduced vertical mixing may also reduce nutrient supply to the surface waters. A reduction in deep convection may also tend to reduce the loss of P and N to the sediment.", Is it not what you want to test? Why do you present this assumption here, in the section "2.2 The SRES–A2 scenario simulation"?

**18)** Line 178 – "the effects of climate and biogeochemical forcings". You used the expressions "climate and biological forcings" and "climate and biological changes", choose one of them.

**19)** Line 201 – "surface average chlorophyll concentrations in the top 10 meters of the CTRL and HIS simulations, and from satellites estimations", it is chlorophyll-a concentration, source of data?

**20)** Line 225 – "analysis reveals much greater variability depending on the region", for which regions? It is important for your results.

**21)** Line 386 – "For instance, the P rich area between Crete and Cyprus is no longer observed in the 2080–2099 period (Figure 9). Moreover,Figure 10 shows that this area matches a productive zone observed in the 1980–1999 period.", It is the only area in the Levantine basin with some phosphate, nitrate and production values different from zero simulated in 2080-99. Are you sure about your observation?

**22)** Line 388 – "The primary production integrated over the euphotic layer (0–200 m) is reduced in our simulation by 10 % on average between 1980–1999 and 2080–2099. However, Figure 10 shows a productivity decrease of more than 50 % in areas such as the Aegean Sea and the Levantine Sea.", Provide time series, as in figure 3.

**23)** Line 397 – "For instance, around Majorca Island, Corsica and Cyprus, changes in local concentrations of nutrients have substantial effects on primary productivity.", Ok for Majorca, but I cannot see something with Corsica and Cyprus. There is also no values provided to evaluate these changes.

**24)** Line 413 – "Sea, the northern Levantine basin and the South Adriatic.", In Fig 11, it is the South of the Levantine Basin and the North of the Adriactic which are P-limited.

**25)** Line 418 – "Figure 12 shows the average depth of the simulated DCM for the period 1980–1999 and for the period 2080–2099.", results for the CTRL not shown, why?

**26)** Line 429 – "At the DYFAMED station, the average DCM depth is unchanged but surface concentration is reduced." A change from $1.10^{-7}$ to $0.75.10^{-7}$ g m$^{-3}$ = $1.10^{-4}$ to $0.75.10^{-4}$ mg m$^{-3}$. Units are certainly wrong…

**27)** Line 432 – "the subsurface maximum in the present and future periods is located at the same depth (100–120 m), but the average productivity is reduced by almost 50 %," Where can we see this? Chlorophyll-a concentration ≠ productivity.

**28)** Line 439 – "Table 4 reports total chlorophyll production in the 1980–1999, 2030–2049 and 2080–2099 periods of all the simulations in all Mediterranean subbasins Adloff et al. (Figure 2 2015)." Why did you quote this reference here?

**29)** Line 496 – "and modification of the physical ocean (vertical mixing, horizontal advection, ...).." Modification of physical processes. Need references.

**30)** Line 497 – "Nutrient fluxes from these sources." Which ones?

**31)** Line 514 – "In these regions, the effects of nutrient runoff changes seem more important than climate change effects (see Table 4)." provide the percentages, and discuss these results.

**32)** Section 4.2 Climate change scenario. I do not see the point of this section. You decide to use the A2 scenario and already justified it in the introduction.

**33)** Line 537 – "Nutrient concentrations in the intermediate and deep layers were shown to be slightly underestimated in comparison to measurements (see appendix)." Provide values.

**34)** Line 543 – "Model values were not corrected to match data, and we are therefore conscious that the uncertainties in the representation of present–day biogeochemistry by the PISCES model may be propagated in the future." This is an important decision that needs to be justify.

**35)** Line 571 – "In particular, nutrient inputs at Gibraltar have substantial consequences" Provide the percentages, and discuss them.

---

## Referee Comment (RC2) · Anonymous Referee #2 · 26 Jun 2018

The paper by Richon et al presents an implementation of the well established state of the art coupled model NEMO-PISCES to the Mediterranean basin for a transient century long simulation under different climate and nutrient input scenarios in order to assess the relative importance of these in present and future nutrient budget of the Mediterranean Sea. Authors also extend the focus from the nutrient to primary productivity and plankton community composition. The work is very comprehensive and including all the aspects is a significant undertaking and authors need to be recognised for that.

Unfortunately authors missed to fully exploit the model to discuss the observed trend

and pattern, often limiting themselves to the description of those or listing a series of potential causes while the model could theoretically support some of these and discard the others. Moreover, sometime the discussion is confused and it seems that consequences are confused with causes (see below for details).

Furthermore, despite the nutrient budget is one of the core topic of the paper, the information are scattered between figures and tables preventing a clear understanding of the changes between the present day and the end of the century. I would strongly recommend authors to summarise all the information either on a table or figure containing all fluxes and stock in the different considered periods and make that the core table/figure from which all discussion unravel. All other figures and tables are very useful to understand the dynamics, but without a single summary point, it is difficult to mentally connect them all to reconstruct the budget.

Method are generally well described, however information on atmospheric deposition is missing and more detail about the initialisation process are welcomed (more details below)

Title and abstracts are clear, language is also generally clear, however some sentence needs revision for clarity. Some of the figures needs improvements, suggestions are given at the end of the detailed review (in the minor comments section).

Areas where further details/discussion is needed or it needs to be clarified:

lines 160-165: Although I fully sympathize with authors regarding the difficulty to have fully consistent source for riverine water and nutrient discharge, and I am not against the choice the authors made, I would suggest authors to briefly discuss the potential impact associated to the incoherency between these, for instance showing how big this incoherence is.

Section 2: there is no mention of atmospheric deposition of nutrient in all the methods section. Only in the discussion (501 and 502) authors state that deposition was not

considered because there are no future estimates of nutrient deposition up to 2100. Is deposition completely neglected or just kept constant at present day value?

Lines 205-210: given the errors at the mouth of the Nile, I suggest to add some more detail on the source of the data for this rivers and the uncertainty associated (e.g. see above).

Lines 210-220: while figure A2 immediately shows area where the model has higher or lower skills, I strongly recommend to add some numerical measures of the ability of the model to capture observed data. Furthermore, authors state that model correctly simulates a DCM, however figure A1 clearly shows how the DCM is much deeper than the data shows (roughly at twice the depth). Authors briefly mention this later in the paper, but I suggest to anticipate this here. It would be also interesting to see a comparison of T, S, nutrient profiles in the same location, to understand the reasons of a deeper DCM

lines 254-256: as authors state, phosphate start increasing in CTRL_RG only at the end of the century, therefore the "strong link" between phosphate and Gibraltar is "proofed" only at the end of the century, for most of the simulation surface phosphate seems to be close to CTRL and CTRL_R despite figure 5 shows quite different P influx around 1990 (positive) and 2040 (negative).

Lines 263-265: the trend highlighted here is apparent only in the A2 forced simulation, raising the doubt that it could be linked to a spin-up issue (see below for other examples on this). Authors explained the initialisation procedure (lines 170-173) although this is not fully clear: all simulations started from the same restart (and in this case initial trend could be due to adjustment to new forcing, particularly in the climate case), or all scenarios have been run for more than 115 year since that initial common restart (and in this case why the trend is only in A2)?

Lines 265-267: the interesting decennial cycle is not evident only in 3e (where is actually weak) and 3f, but also on the surface. Can authors suggest some mechanism

for this? Is this a cycle in intensity of stratification? Is this associated to cycles in the atmospheric patterns?

Line 259 vs 269: authors claims that there is a "slight accumulation of phosphate in the deep Western basin" while "the evolution of nitrate concentration shows marked accumulation in all region". I could be wrong (this is simply a visual calculation), but focussing on A2 trend from 1980 to 2100 figure 3f shows an increase from approx .15 mmol/m3 to 17.5 mmol/m3 (+16%), figures 4e and 4f show a similar relative increase, while 4b and 4c a smaller relative increase. Even if we compare these with the CTRL simulation, the difference between the P accumulation in the deep Western basin and the N accumulation is not that big as the qualification "slight" and "marked" suggest.

Lines 274-277: Can authors explain why riverine discharge seem to have more impact on N than P? How much is simply due to the different evolution of the forcing, and how much is due to internal dynamics?

Lines 283-285: I suggest authors to clarify what they means by "linked with nutrient exhaustion". The link between stringer stratification and lower surface nutrient is well established in literature (due to lower winter mixing), what nutrient exhaustion add to this mechanism and do authors have supporting evidence for this?

Line 301: in the absence of statistical measure of the trends, nutrient fluxes at Gibraltar seems characterised by a high interannual variability until about 2060 rather than a coupled decreasing-increasing trend.

Lines 300-304: authors correctly state that the relative increase in the influx is higher than the relative increasing of the outflux, but what's the difference in absolute term? Is there a change in the net flux?

Line 326: similar to comment on 263-265, could the sudden drop be justified by the adjustment to the new forcing (spin-up)?

Lines 335-370: it is not clear if here authors are still discussing sedimentation fluxes,

or rather the global nutrient budget. If the latter I recommend to separate this part with a different sub title (and develop this around the new suggested figure-table). Also, if the latter is true, I suggest to clarify the sentence "the sum of nitrogen fluxes in the Mediterranean": is this the total net flux in the Mediterranean?

Line 354-356: authors seems to suggest that the accumulation of phosphate could be due by the decreased primary production, however this is in contrast with the fact that P is the more limiting nutrient in most of the domain (an accumulation in P should lead to an increase in primary production in a P limited environment). Authors should clarify the mechanisms behind the observed trends and the interaction among those mechanisms.

Section 3.4: from my understanding of the PISCES model, sedimentation of particulate nutrient is mostly driven by primary (sinking phytoplankton) and secondary (faecal pellet) productivity. If this is true, I would suggest authors to use the patterns observed in the biological productivity to discuss and interpret the sedimentation fluxes (an possibly anticipate 3.4 before sedimentation)

Section 3.4: authors discuss at length nutrient limitation presenting co-limitation as a widespread condition in most area of the Mediterranean. To my knowledge there is quite an extensive literature on P limitation in the Mediterranean, particularly in the Eastern basin (e.g. a long list of publications by Krom and co-authors), and authors do not refer to any of this. I strongly suggest to include these in the discussion, compare the results from the model with those findings and suggest potential reasons for the difference. (By the way, I want to emphasize that I have not contributed to any of those papers)

line 409: the use of "thus" suggests that as a consequence of the increased P-limitation in the eastern basin, the surface P concentration will further decrease? Could authors clarify what is the positive feedback between P limitation and P reduction?

Lines 438-439: authors state that "The changes in DCM we observe combined with

external nutrient input changes result in 17 % reduction in integrated chlorophyll production", however the DCM is a consequence of the chlorophyll production and not a cause. DCM is simply the location of the sub-surface maximum of chlorophyll, therefore, unless authors clarify the meaning of that sentence, is the reduction in the chlorophyll production that leads to the changes in the DCM

line 468-470: as above, could these be a consequence of the A2 model still adjusting to new atmospheric forcing?

Line 512-514: the fact that coastal area (and in particular the Adriatic sea) is largely influenced by coastal nutrient inputs is not a new finding and authors should acknowledge the past literature

Conclusions: the conclusions are too high level and therefore miss to emphasize the importance of the new findings. For instance the fact that the Western Med is more influenced by the Gibraltar influx than the Eastern Med (lines 590-596) is not that surprising. Furthermore, authors cite their work emphasizing that atmospheric deposition can bring up to 80% of phosphate in some region of the Mediterranean sea. As asked earlier, authors do not clarify if atmospheric deposition is completely neglected in this simulation or simply kept constant: is the former, a lot of the results and discussion are heavily biased by the simplifying assumption and authors should careful and rigorously discuss how this simplification affect all the results presented.

Minor comments:

line 46: this sentence "the Mediterranean thermo- haline circulation (MTHC) may significantly change with a consistent weakening in the western basin for greenhouse gases high-emission scenarios..." needs clarification: does authors means that the MTHC weakens in the western basin in climate change scenario characterized by high emission scenario?

Line 76: which daily 3D forcings are needed by the biogeochemical model? Do authors

refer to boundary condition at Gibraltar?

Line 112: "that" needs to be removed

line 145: Although I understand that authors refer to original manuscript for details, I would suggest authors to briefly explain (or show within a map) the extent of the buffer zone

line 155: Authors state the four largest rivers of the Mediterranean and Black Sea are the Rhone, the Po, the Ebro and the Danube. Although I agree that these are 4 important rivers, these are surely not the 4 biggest one. Acording to Ludwig 2009 (cited by the authors) the Danube mean flow is 6573 m3/s, Rhone 1721m3/s, Po is 1569m3/s and Ebro is 416 m3/s. Clearly the Nile is the largest, and the Dniper is also bigger than the Ebro

lines 158-160: this sentence does not flow correctly in English, and is not clear, I suggest authors to revise it.

Line 315: nitrate needs to be capitalised after the full stop

line 540: If the authors want to suggest that the loss of nitrate in the CTRL run can be due by an underestimation of riverine fluxes in the CTRL riverine forcing, I recommend authors to rewrite the beginning of this sentence and explicitly state that, instead of simply referring to the discrepancy with A2 (since the latter can't influence CTRL)

figures 3 and 4: it is very hard to distinguish between grey, blue and green line. Although I recognise that the quality of the picture in the PDF is not at the highest, I strongly suggest the authors to use more contrasting colours.

Figures 8-9-10: I suggest authors to consider to modify the 2080-2099 panels by showing a map of the difference with the 1980-2000 period to better highlight the evolution of N,P and primary production

―――――――――――――――――

---

## Referee Comment (RC3) · Anonymous Referee #3 · 4 Jul 2018

Review for Biogeoscience entitled:

**Biogeochemcial response of the Mediterranean Sea to Transient SRES-A2 climate change scenario by Richon et al.**

This paper addresses the impact that climate change and future riverine nutrient inputs will have on the biogeochemistry of the Mediterranean Sea over the period 1980-2100 using a high resolution coupled NEMO/PISECES model.   This paper is important for the scientific community as it is the first time a transient simulation on the response of the biogeochemistry of the Mediterranean Sea to climate change has been run.  The authors separate the individual effects of the different scenarios to help determine the reasons for the future biogeochemical changes that the model predicts.   In addition they looked at both the impact of each scenario to nutrients and the phytoplankton and zooplankton communities. This study concludes that nitrate concentrations in the Mediterranean are likely to increase in the future while there is no change in phosphate concentrations. They further predict a decrease in net primary productivity.   In general, the use of English is good although there are some areas which need clarification which I have highlighted below.  However I have some concerns regarding the initial conditions of the model and the analysis of model results which need addressing before this can be published.   This review will start with more general comments before detailing more minor changes.

My main concern is regarding the initial nutrient conditions in the model.  The authors state on line 212 that there is "some underestimation of nutrient concentrations" and again on lines 536-537 that "nutrient concentrations are slightly underestimated" but they do not quantify this.  I was surprised when looking at Figure 4 to see deep water nitrate concentrations between 4 and 4.5 µM in the Western Mediterranean Sea.  I was expecting to see nitrate concentrations on the order of 8-9µM and hence I do not call this a slight underestimation.  Although there is a slight W-E gradient in nutrient concentrations within the model it is not anywhere near as strong as observations suggest. Together with the fact that nitrate in the DW is decreasing in the control scenario suggests to me that the model is not be capturing the biogeochemical cycling of nitrate correctly and raises the question of the validity of future model results. How does this underestimation of nitrate concentrations (and to a lesser extent phosphate concentrations) in the Western basin impact the results of changing circulation such as decreased deep water formation and increased stratification? Would this have a major difference on results?  Can the model really predict future changes due to climate change if it can't predict present day conditions correctly?

My next concern is in regards to whether the authors include dissolved organic matter inputs though the Strait of Gibraltar and from rivers into the model? On line 291-293 the authors say that

*"The Mediterranean is a remineralization basin that has net negative fluxes of inorganic nutrients (i.e. organic nutrients enter the basin through the Gibraltar Strait surface waters and inorganic nutrient leave the Mediterranean through the deep waters of the Gibraltar Strait"*

However there is no mention of dissolved organic nutrients anywhere in this paper.  Do the authors include them in the inputs through the Strait of Gibraltar (or in the riverine input)? If yes this needs to be explicitly stated and if no then they are missing a major source of phosphorus and nitrogen in their model (see Powley et al., 2017; 2018). In addition the paper tries to present a nutrient budget based solely on nitrate and phosphate and then use the imbalanced budget to explain the decrease in nitrate in the CTRL model (Line 540).   However dissolved organic matter inputs need to be

included in the budget so that total N and P inputs and therefore a complete budget can be calculated (see Powley et al., 2017; 2018) In addition I suggest creating a Table summarizing the budget as currently it is difficult to interpret from the graphs. Finally Lazzari et al. (2014) conclude that dissolved organic matter is increasing in their model in response to climate change. Do your results agree? (I know this is not a key result but a sentence regarding this could be added to the discussion)

The results section is very qualitative with little quantitative analysis. Phrases such as slightly increase and significantly increase are common with no data to back them up. In addition I feel that section 3 and especially section 3.3 can be condensed as there is a lot of repetition and is hard to follow in places. This would make the main conclusions and outputs of the paper clearer to the reader. I suggest re-organizing section 3.3 to start with the nutrient budget first, analysing the different inputs and outputs from rivers, Straits of Gibraltar and sediment before going on to look at the effect of the different scenarios to nutrient concentrations. This way you can bring in the analyse from the budgets to explain the concentration trends rather than having to repeat yourself analysing and explaining the trends in nutrient concentrations before you have analysed the causes. I would also in general try and keep the same structure within each section in regards to the analysis i.e compare phosphate first, then nitrate, etc.

**General minor comments**

While I appreciate that you are limited by both data and computational power in your model runs I suggest refraining from using '*external inputs*' and instead be specific and use 'fluxes through the Straits of Gibraltar and riverine inputs'. As far as I understand you are not including atmospheric inputs, direct wastewater discharges or submarine groundwater discharges in your model which can all be considered external nutrient inputs.

Use Strait of Gibraltar or Gibraltar Strait throughout the paper rather than Gibraltar as Gibraltar is a body of land!

**Detailed minor comments**

Line 8: Change "coastal nutrients" to riverine nutrients. Coastal nutrient inputs could mean coastal runoff, direct wastewater discharges submarine groundwater discharge.

Line 9: Do you just mean from riverine inputs rather than external sources.

Line 27: I thought the last deposited Sapropel was 10,000 years ago not that they have been deposited for the last 10,000 years

Line 36: Please quantify the short residence time (i.e 100 year timescale) and add reference.

Line 40-41: Please re-phrase. The word transport in this sentence doesn't make sense.

Line 48: I am confused by the Adloff reference at the end. Do they also show this enhanced vertical stratification?

Line 49: remove "lead to"

Line 70: Add Heurtas et al. (2012) to the Gibraltar references. Add more references for atmospheric deposition or say 'and references therein'. There have been a lot of studies on atmospheric deposition in the Mediterranean region. What about direct wastewater discharges (Powley et al., 2016) and submarine groundwater discharges (Rodellas et al. 2015). Note also Powley et al., (2017;2018) have calculated a complete nutrient budget for the Mediterranean and these should be referenced somewhere in this paper.

Line 100: define the SST acronym rather than on line 222

Line 111: define the SSS rather than on line 222

Line 112: remove 'that'

Section 2.3 Please add a bit more detail regarding the biogeochemical model and the compartments so the reader has an idea of what is included without having to go to the references (i.e Are there compartments for bacteria, DOM etc?).

Section 2.3 Include a sentence regarding why you did not use atmospheric deposition, and other external inputs in this section.

Lines 150-155: Please be specific in which MEA scenario you use. None of the four scenarios are called business as usual. Also how did you combine values from the two Ludwig papers together as Ludwig et al. (2010) states that they are not directly compatible with one another.

Line 175: Why are 1966-1981 conditions used when the model results are from 1980 onwards? Please specify in the text.

Line 184: Write minus rather than using the minus sign as it wasn't clear to me what you meant initially.

Line 203 satellite not satellites

Line 212-213: Quantify the error. Compare model results with measurements. (See my main concern)

Line 224: When you say global I assume you mean across the entire Mediterranean. Please clarify

Line 229-235: I suggest moving this section to where you discuss the budget as no results are given and it confuses the reader

Line 230: Add references after "*nutrient budgets are highly dependent on external sources*".

Line 244-246: State this later on when you are talking about limiting nutrients.

Line 252: How much does the phosphate concentration decrease by? From what to what?

Line 253: Only use significantly if it is statistically significant. If yes then the state the statistics.

Line 254: Add "in the surface water in the Western Mediterranean Sea" after phosphate concentrations. The reader shouldn't have to look at the figures to know which water body you are talking about.

Line 254-255:  What about the comparison between CTRL_RG and CTRL R?

Line 263: *"The eastern part of the basin contains approximately 50 % less phosphate than the western part."* Where have you got this data from? Table 3 shows greater phosphate content in Western Mediterranean than Eastern Mediterranean. If you mean concentrations then I would still argue that your model is not showing 50% less.

Line 265-267: Give quantitative values.

Line 268: State which scenario you are talking about.

Line 272. Reference Figure 5 after Atlantic

Line 278-281: Consider consolidating with previous paragraph as there is a lot of repetition (especially concerning impact of rivers of nitrate concentrations).

Line 293: Reference needs brackets around it.

Line 295-298: Put quantitative results.

Line 305-307: What about statistics for nitrate?

Line 308: I suggest adding a figure showing the net fluxes through the Strait of Gibraltar to show imbalance. Again, I also suggest the authors look at total nitrogen fluxes to be able to determine whether there is an imbalance at the Strait of Gibraltar not just nitrate.

Line 315 Capitalise N in nitrate after full stop.

Lines 315-320:  Is there a difference in the evolution of riverine discharge of nutrients between Western and Eastern basins? Combining everything into one flux makes this impossible to see, but if it differs it would have a large impact on results and may explain the differences to the riverine scenario seen in the two basins.

Line 328: Replace "This" with "The"

Line 329-330: put comma after occurs and flux

Lines 328-332: What happens to sedimentation in the CTRL-R and CTRLRG scenario?  This will make the argument stronger whether sedimentation is linked to decrease in vertical fluxes.  Also please be more explicit about the process that would increase phosphate and nitrate in the deep water.  A lower particulate matter flux to the deep water would lead to lower remineralisation fluxes and therefore lower phosphate and nitrate.  Alternatively, higher water temperatures could lead to higher remineralisation and therefore a lower sedimentation flux despite the same flux of particulate matter exiting the surface water. How do the authors know it is not higher temperatures rather than changes in the water flux that alters the sedimentation flux?

Line 334: A new section heading is needed before line 334.

Line 334: Into rather than in?

Line 334-335. Compared to when? The start of the model run? When you say the sum do you mean the balance between inputs through the Gibraltar Strait, riverine inputs and sedimentation? Be specific. Also, as mentioned before, I suggest including dissolved organic matter in your calculations and to produce a table to show the balance of the different inputs and outputs of phosphorus and nitrogen.

Line 350-351: The authors state *"changes in Gibraltar exchange fluxes of phosphate seem to have limited effect on Mediterranean phosphate content"* but on lines 305-306 they also state *"evolution of phosphate concentration in the Western basin is linked with Gibraltar inputs"*. Which one is it?

Line 353-356: What about increase in temperature effecting results? Luna et al., (2012) hypothesise increasing deep water temperatures will increase prokaryotic metabolism, thus potentially increasing nitrate and phosphate concentrations. Lazzari et al., (2014) also predict that increasing temperatures increase metabolic rates.

Line 378: Replace "and shows" with "showing"

Line 380: Delete as this is repeated and explained in the next paragraph.

Line 384" surface phosphate what? Concentrations? Masses?

Line 386-387: The decrease does not look that clear to me and it certainly doesn't become entirely depleted in phosphate.

Line 409: Remove would

Line 410-414: Why would the phosphate become the major limiting nutrient in areas where primary productivity is reduced?

429-430: Reduced by how much?

Line 433: Are we still talking about chlorophyll or primary productivity?

Line 434: What is "it"?

Line 441: Remove Adloff reference?

Line 452? Do you mean riverine nitrate inputs or total nitrate inputs?

Line 460-463: Which scenario are you talking about?

Line 463: Add Western Mediterranean Sea after diatom concentrations.

Line 465: Add diatoms after concentration

Line 478: Add references if other studies have concluded this.

Line 483: "After all" doesn't make sense in English. I suggest you use "Altogether".

Lines 494-497: Add references to this sentence

Line 514 and 515: When talking about "coastal nutrient inputs" and "coastal runoff" are you only talking about riverine inputs? Be specific.

Line 518: 'developing' not 'developping'

Line 540: I don't think you can say there's an imbalance in sources and sinks without looking at the organic nitrogen aswell (unless you count nitrate sourced from DOM). There will always be an imbalance of nitrate in the Mediterranean if you only look at external sources as you mention yourself it is a remineralization basin.

Line 591-592: Please re-phrase the sentence and be specific to what you are talking about. What inputs?

Line 598: I disagree with this statement. There are lots of nutrients inputs that you haven't considered such as direct wastewater discharges, diffuse runoff, submarine groundwater discharge.

Line620: Shown rather than showed

Table 2: It would be more informative to present the numbers per $m^2$ surface area and then comparisons between basins can happen as the Eastern Mediterranean is almost twice the size of the Western Mediterranean. The reader can still calculate the total mass if the surface area is given. Also taking the difference from the control rather than total values might make it easier to spot trends.

Figures 3 and 4: Please create a greater contrast between the CTRL R and CTRL RG. These are difficult to see at present.

Figure 5: it would be nice if the net flux of nitrate and phosphate across the Strait of Gibraltar could be added aswell.

Figures 8,9,10 and 12. I struggled to see the differences between the two time periods which were mentioned in the text. I suggest to produce a figure of the anomaly between 2080-2099 and 1980-1999 rather than having the 2080-2099 figure as it currently is. The changes with time will then be more evident to the reader.

Figure 10. Units in caption do not match units in figure

Figure 12: Why is the HIS/A2 scenario figures not presented aswell?

Figure 13: Units in caption are different than on figure

Figures 14 and15: What are the units? mmol $m^{-3}$ of what? Carbon?

Figure A1: When were the data points collected? Would the HIS/A2 scenario be a better comparison than CTRL as it actually uses 1980-2000 data?

Figure A.2 Change $P_4$ to $PO_4$ in the label.

**References**

Huertas, I. E., A. F. Ríos, J. García-Lafuente, G. Navarro, A. Makaoui, A. Sánchez-Román, S. Rodriguez-Galvez, A. Orbi, J. Ruíz, and F. F. Pérez (2012), Atlantic forcing of the Mediterranean oligotrophy, *Global Biogeochem Cycles*, *26*(2), doi: 10.1029/2011gb004167

Lazzari, P., G. Mattia, C. Solidoro, S. Salon, A. Crise, M. Zavatarelli, P. Oddo, and M. Vichi (2014), The impacts of climate change and environmental management policies on the trophic regimes in the Mediterranean Sea: Scenario analyses, *J Mar Syst*, *135*, 137-149, doi: 10.1016/j.jmarsys.2013.06.005

Luna, G. M., S. Bianchelli, F. Decembrini, E. De Domenico, R. Danovaro, and A. Dell'Anno (2012), The dark portion of the Mediterranean Sea is a bioreactor of organic matter cycling, *Global Biogeochem Cycles*, *26*, doi: 10.1029/2011gb004168

Powley, H.R., Krom, M.D., Van Cappellen, P. (2018). Phosphorus and nitrogen trajectories in the Mediterranean Sea (1950-2030): Understanding basin-wide anthropogenic nutrient enrichment. *Progress in Oceanography,* 162, 257-270

Powley, H.R., Krom, M.D., Van Cappellen, P. (2017). Understanding the unique biogeochemistry of the Mediterranean Sea: Insights from a coupled phosphorus and nitrogen model. *Global Biogeochem. Cycles* **31**, 1010–1031. doi:10.1002/2017GB005648

Powley, H.R., Dürr, H.H., Lima, A.T., Krom, M.D., Van Cappellen, P. (2016). Direct Discharges of Domestic Wastewater are a Major Source of Phosphorus and Nitrogen to the Mediterranean Sea. *Environ. Sci. Technol.* **50**, 8722–8730. doi:10.1021/acs.est.6b01742

Rodellas, V., J. Garcia-Orellana, P. Masque, M. Feldman, and Y. Weinstein (2015), Submarine groundwater discharge as a major source of nutrients to the Mediterranean Sea, *Proc Natl Acad Sci U S A*, *112*(13), 3926-3930, doi: 10.1073/pnas.1419049112

---

## Author Comment (AC1) · 28 Sep 2018

We thank the referee for their comments. The referee expressed concerns about the way the experiments were designed and presented. We tried to respond to all comment and questions. In particular, we made efforts to improve the evaluation of the model results and provided more quantitative analyses of the results. We also reorganised the results section, changed some figures and added new ones in order to make the article easier to read.

All answers to questions and comments are presented in the joint files. We also join a revised version of the manuscript with a track version allowing to follow changes made

since the first submission.

Please also note the supplement to this comment:
https://www.biogeosciences-discuss.net/bg-2018-208/bg-2018-208-AC1-
supplement.zip

---

## Author Comment (AC2) · 28 Sep 2018

We thank the referee for their comments. As suggested, we rearranged the results section in order to include the nutrient budgets first and then discuss the potential causes for their modifications (changes in river inputs, exchanges through the Strait of Gibraltar and sedimentation). We also clarified some methodological points (initial conditions, atmospheric deposition) and present a figure summarising the fluxes of nutrients in the Mediterranean and their evolution throughout the century.

We present all answers to comments and questions in the attached material, along with a revised version the manuscript and a track version allowing to monitor changed

made since the first submission.

Please also note the supplement to this comment:
https://www.biogeosciences-discuss.net/bg-2018-208/bg-2018-208-AC2-supplement.zip

---

## Author Comment (AC3) · 28 Sep 2018

We thank the referee for their comments and questions.

They expressed concerns about initial conditions of the simulations and the lack of dissolved organic nutrients in the budgets presented in the article.

We tried to clarify the methods and results sections in order to clarify how the initial conditions are produced and the influence they may have on our results. In particular, we recognize that our model underestimates nutrient concentrations in comparison to available measurements. This discrepancy between model and measurements raised

the question of the validity of our simulation results in representing the future state of the Mediterranean. We argue that the performances of a model in representing present conditions do not guarantee its accuracy to represent future evolutions. This is due in part to the fact that most climatic and biogeochemical processes are non linear and a consistent mis-estimation of nutrient concentrations in the present does not necessarily hamper the representation of nutrient concentrations trends in the future.

Detailed answers to all comments and questions are attached, along with a revised version of the manuscript and a track version allowing to visualise changes made since the initial submission.

Please also note the supplement to this comment:
https://www.biogeosciences-discuss.net/bg-2018-208/bg-2018-208-AC3-supplement.zip

---

## Author Response (AR1)

*1) As mentioned by the authors, the atmospheric deposition has a significant impact on the Mediterranean Sea. The first author quoted two of its own papers to support this. Even if no transient scenario for atmospheric deposition exists, did the model contain a present-day atmospheric deposition component? (As for example, the model analysis of Herrmann et al. (2014) and Macias et al. (2015) that used continued present–day discharge of nutrients) If yes, the influence of the atmospheric depositions should be included in the discussion of the results. If not, I suggest use of continued present–day atmospheric depositions in the model.*

The simulations presented in the article have no atmospheric deposition. However, we ran simulations containing the continued present-day atmospheric deposition of natural and anthropogenic nitrogen and phosphate from natural desert dust (the aerosol deposition fields used in Richon *et al* 2017). We initially chose to discard the results of these simulations from the article because we wanted to keep as much as possible a coherence in the scenarios for external biogeochemical forcings.

We include in the discussion section (4.1) a paragraph on these simulations (lines 587-601). These simulations show that in the beginning of the simulation period, the effects of nitrogen deposition are important in the northern and eastern part of the Mediterranean (15 to 20 % enhancement in average surface primary production). Phosphate deposition from natural dust has important impacts on the southern part of the Mediterranean (more than 20 % primary production enhancement in the South Ionian basin). These results are coherent with Richon *et al* (2017). By the end of the century, the results from these simulations indicate that the relative effects of nitrogen deposition have declined and the effects of phosphate deposition from dust are observed across the entire basin. These observations may be the result of the general phosphorus limitation in the Mediterranean. As a results of this limitation, the effects of extra nitrogen brought to the surface by deposition are negligible whereas the effects of phosphate are relatively important.

*2) The present-day period (1966–81) cannot be older than the historical period (1980-99). Therefore, the CTRL simulations does not correspond to the present–day conditions, and because condition forcing between the periods 1966–81 and 1980-99 are different, results from the control simulation (CTRL) differed from those of the scenario simulation (HIS/A2) during the first simulated decades (from 1980 to now). Authors need to justify and clarify this choice.*

This is an imprecision from us to call the CTRL period "present-day period". We change the phrasing to "*The control run CTRL is performed with forcing conditions corresponding to the period 1966–1981.*"
This control period was chosen in order to avoid years with too important warming such as the 80s and 90s.

*In this section, the comparison of the model results with in situ data have been incorrectly conducted. The main issue is that no values to support the comparison between the model and the in situ data are provided (e.g., correlation coefficients, percentage of differences...). For example, the chlorophyll-a concentration in the Gulf of Lion seems two times lower in the model simulations than the estimates from satellite.*

*There is no information on the spatial variability of the nutricline depths (i.e., nitracline and phosphacline), and of the DCM.*

*The figure 1, associated with this section, compares data from the satellite and the model results during two different periods 1980-99 and 1997-2012.*

*The units are not coherent:*

   *- "1 K colder than observations", temperature in Kelvin?*

   *- Figure A1 – "Chla $10^{-9}$ g m$^{-3}$", in the text: "227 ± 136 $10^{-9}$ g L$^{-1}$", not consistent between them, and in the literature the most common unit is mg/m3 (or microg/L).*

*Maybe, model values do not agree well with the in situ data, but spatial and temporal variabilities that exist between the different Mediterranean regions have to be simulated by the model. Unfortunately, quantitative information to support this hypothesis are not provided by the authors.*

Reviewer is right, the evaluation of the model performances is mainly a visual comparison between the model outputs and the data.

In order to add some quantitative evaluation, we add a plot showing the average surface chlorophyll-a concentration over the Mediterranean basin from the satellite products of Bosc et al (2004) and from our HIS/A2 simulation for the years 1997 to 2005 (Figure 2). This figure shows the standardized average and standard deviation. We note that the model surface chlorophyll is underestimated (by a factor 2 on average) as shown on the maps. However, the satellites tend to yield overestimations of the surface chlorophyll-a concentrations, especially in the coastal and upwelling regions because of the high particulate matter concentrations in these areas, and also generally in the oligotrophic Mediterranean (see Claustre et al, 2002, D'Ortenzio et al, 2002, Bosc et al, 2004, Morel and Gentili 2009). The values show a good correlation between the model and the data over the 1997-2005 period. Therefore, the comparison of our modelled surface chlorophyll-a concentration with 2 independent data bases (namely SeaWIfs and MyOcean dataset) confirms that our model reproduces satisfyingly the surface chlorophyll-a in the Mediterranean.

We added some precisions on the nutricline variability (lines 233-240): "The vertical distribution of nitrate and phosphate over a section crossing the Mediterranean from East to West as well as chlorophyll and nutrient concentration profiles at the DYFAMED station are shown in appendix (Figures A1 and A3). These figures show that the model produces some seasonal and interannual variability of the nutricline depth and intensity. However, the nutricline depth and DCM depth are consistently overestimated by the model in comparison to the data. The nutricline intensities seem to be underestimated by about 50 % and the depth is overestimated. However, nutricline depth deepens from 100-120 m to 180-200 m between the western and the eastern basins (see Figure A3)."

*- "1 K colder than observations", temperature in Kelvin?*

We changed the temperature difference units to degres C.

*• Figure A1 – "Chla 10-9 g m-3", in the text: "227 ± 136 10−9 g L−1", not consistent between them, and in the literature the most common unit is mg/m3 (or microg/L).*

The units have been checked and corrected to ng/L. Figures A1 and A2 were changed to include the vertical profiles of chlorophyll at the DYFAMED station over several years.

*Section 3.3.1 Evolution of phosphate and nitrate concentrations*
*Figure 3-4 – Adjust the y-axis, it is impossible to evaluate the results.*
Figures 4, 5, 15 and 16 were readjusted. We rearranged the order, they are now figure 7, 8, 15 and 16. We also tried to accentuate the different lines and added average concentrations for the 1980-1999, 2030-2049 and 2080-2099 periods on the plots.

*Line 259 – "A slight accumulation of phosphate is observed in the deep western basin" - For which simulations? Provide values.*
We added the value (0.015 mmol m$^{-3}$) and precised this is for the HIS/A2 simulation.

*Line 268 – "The evolution of nitrate concentration shows a marked accumulation over the century in all regions of the intermediate and deep Mediterranean waters" - For which simulations?*
In the HIS/A2 simulation. We added the sentence "In particular, nitrate concentrations increase of about 0.5 mmol m$^{-3}$ between 1980 and 2099 in the deep eastern basin."

*Line 279-289 – Confusing, mixing general results (for both nitrate and phosphate) with results*

**specific to the nitrate that should have been present in previous paragraphs.**
**This sections need to be clearer. Stay with the same logic when you present your results.**
**Compare CTRL with HIS/A2, western basin with eastern basin, depth by depth…**

We rearranged this paragraph.

*3.3.2 Exchange fluxes of nutrients at Gibraltar*

*Figure 5 – Keep the same x-axis as in the figures 3 and 4.*
Figures 6, 7 and 8 were changed.

*Line 293 – "We observe similar trends in phosphate and nitrate fluxes linked to the*
*Redfieldian behavior of the primary production in PISCES." - What do you mean, where can*
*we see this?*
We modified this section to "Figure 6 shows the evolution of incoming and outgoing nitrate and
phosphate fluxes at Gibraltar in the HIS/A2 and in the CTRL simulations. We observe similar trends
in phosphate and nitrate fluxes in the model. This is linked to the Redfieldian behavior of the primary
production in PISCES."

*Line 295 – "the incoming fluxes decrease" - fluxes of what?*
We added the precision "fluxes of nitrate and phosphate".

*Line 295 - "According to the HIS/A2 simulation, the incoming fluxes decrease slightly until*
*the middle of the century and then increase to reach values higher than the control in the last*
*25 years of simulations. Outgoing fluxes follow the same trends as incoming fluxes" – For*
*the incoming fluxes, I see, a peak in the 90s, then stable incoming fluxes until a decrease in*
*the 2030s, and then an increase in the last 25 years with a peak in the 2080s. For outgoing*
*fluxes, I see, a slight increase in the first half of the 21st century and a decrease after. Not*
*you?*
The outgoing fluxes in figure 6c and 6d are negative, the lower absolute values indicate that the flux
is weaker.
These sentences were changed to "According to the HIS/A2 simulation, the incoming fluxes of nitrate
and phosphate decrease slightly (from 50 the 35 Gmol/month for nitrate and from 2.5 to 1.55
Gmol/month for phosphate) until the middle of the century (with a period of increased incoming fluxes
of both phosphate and nitrate in the 1990s) and then increase to reach values higher than the control
in the last 25 years of simulations (Figure 4). Outgoing fluxes follow the same trends as incoming
fluxes: total outgoing nitrate and phosphate fluxes decrease from 1980 to 2040 (flux values getting
closer to zero) and then increase until the end of the century."

*Line 298 – "We observe a drift in the nitrate outgoing flux in the control." – Provide a value*
We observe a decrease of about 15 % in nitrate outgoing flux between the beginning and the end of
CTRL. (Lines 326-327: "We observe a decreasing trend in the nitrate outgoing flux in the control
(from -129 to -110 Gmol/month representing about 18 %).")

*Line 305 – "Figures 3a and 5b show that the evolution of phosphate concentration in the*
*western basin is linked with Gibraltar inputs (Pearson's correlation coefficient is 0.63, p–*
*value=10−14)" - Correlation between what and what, surface, intermediate or deep concen-*
*tration of phosphate?*

Correlation between surface phosphate concentrations and Gibraltar phosphate inputs. We added
the precision in the sentence and corrected the value that was calculated for the entire water column.

*Section 3.3.3 River fluxes of nutrients*

*Figure 6 – Keep the same x-axis as in the figures 3 and 4.* Figure changed
*All tables – In the result section you only wrote in percentages. Therefore, provide*
*percentages values in tables.*
We added percentage values in the Tables. Also, as suggested by another referee, we provide 2
schematics summarizing the phosphate and nitrate budgets and fluxes. We provide percentages in

these schematics (figure 18).

Sentence changed to "River discharge is the main external source of phosphate for the eastern part of the basin (Krom et al 2004, Christodoulaki et al 2013)".

The difference is between 30 and 60 Gmol/month. Precision added.

The CTRL values are looping on the years 1966 to 1981. Therefore, the CTRL values represent the late 1960s.
Therefore, we can evaluate that nitrate total discharge increase from approximately 20 Gmol/month in 1966 to more than 55 Gmol/month in 1981.

*I see that there is no internannual variability in the HIS/A2 simulations. You have to say something about it. Phosphate concentrations mainly decrease between 1980 and 2000. Why? Nitrate and Phosphate concentrations mainly increase between 2030 and 2050. Why?*

In the Methods section (2.4), we stated "Yearly values are obtained by linear interpolation between 2000 and 2030 and between 2030 and 2050, after which they are held constant until the end of the simulation in 2100." There is no intrisic interannual variability accounted for in the nutrient riverine input scenario and this is the only transient scenario we found available. Moreover, the "Order from Strength" hypothesis appears the most consistent with the A2 climate change scenario.

Phosphate discharge decreased over the Mediterranean between 1980 and 2000 as a result of the European regulations on phosphate content in household products. After the 1980s, phosphate concentration in rivers decreased. No regulation on nitrate lead to the consistent increase in nitrate discharge observed in the forcings.

We added some more precisions in our comparison with Lazzari et al.:
"Lazzari et al (2014) tested the effects of several land-use change scenarios on the A1B SRES climate change scenario over 10-years time slices. They found a general decrease in phytoplankton and zooplankton biomasses (about 5 %) that is lower than in our severe climate change scenario. In our simulations, average phytoplankton biomass decreases by about 2 to 30 % (see Figure 15 and average zooplankton biomass decreases by about 8 and 12 % (see Figure 16). However, our transient simulations revealed non linear trends in plankton biomass evolution. Lazarri et al (2014) also conclude that the river mouth regions are highly sensitive because the Mediterranean Sea is influenced by external nutrient inputs. Our results show the same sensitivity of the Mediterranean to external nutrient inputs."

**Line 564 – "Results from Herrmann et al. (2014) indicate that chlorophyll production" – Chlorophyll production? Are you sure... I think you want to study Primary Production, or Net Primary Production. It is a major mistake…**

Figure 3 and Table 1 from Herrmann et al (2014) show that chlorophyll is increasing by about 8 % between the present and future periods. See section 3.2.2 from their article "The annual total

chlorophyll biomass increases in average by 8 % between the present and future periods (Table 1). This increase is mainly associated with the winter mixing and spring bloom periods (February– May), whereas the total chlorophyll biomass does not change statistically significantly during the stratified summer-fall period (Figure 3). It can be attributed to the convection weakening and surface warming (Figure 2), which favors the photosynthesis in our model"
We changed the term to "Chlorophyll concentration".

**Line 567 – "may lead to a decrease in chlorophyll and plankton biomass" - Provide values**
We changed the sentence to "Our results indicate that the contrasting effects of vertical stratification and biogeochemical changes may lead to a decrease in chlorophyll-*a* concentration, phytoplankton and zooplankton biomass content of up to 50% locally and between 10 and 30 % at the basin scale as indicated by Figures 14, 15 and 16 "

**Figure 14 & 15 – "nanophytoplankton and diatoms concentration (10−3 mol m−3)" "mesozooplankton concentrations (10−3 mol m−3)" – A mol of diatoms? A mol of mesozooplancton? Wrong units…**
The model units are in moles of C. We added this precision in the figures.

**Line 571 – "In particular, nutrient inputs at Gibraltar have substantial consequences on the western basin." – Provide an estimate.**
"Results from Figures 4a and 5a and Table 4 indicate that the increase in nutrient inputs from Gibraltar at the end of the century are responsible for a 2.5 % increase in chlorophyll concentration in the western basin during the 2080-2099 period." lines 622-624

**There are only four references in this crucial section (Lazzari et al., 2014; Herrmann et al., 2014; Macias et al., 2015). It is not enough…**
We have added references to Luna et al 2012, Krom et al 2004, 2010, Pujo-Pay et al 2011 and Chust et al 2014.

List of corrections*:*

**1)** *Line 7 – "socio-economic", you used both socio-economic and socioeconomic in the text, choose the good one.*
We chose to use " socio-economic " in the text.

**2)** *Line 10 – "lead to changes in phytoplankton nutrient limitation factors.", which ones?*
We changed the sentence to " lead an expansion of phosphorus--limited regions across the Mediterranean. "

**3)** *Line 26 – "known as sapropels, have been recorded through the last 10 000 years", It is the most recent sapropel events that apparently lasted for 3000 years, other events occurred before. Please clarify.*
We changed the sentence to "In particular, high stratification events, characterized by the preservation of organic matter in the sediment, known as sapropels, have been recorded through several over geological times, the most recent was recorded 10 000 years ago and lasted about 3 000 years."

**4)** *Line 33 – "and had biogeochemical impacts", which ones? Where? Need references.*
Lascaratos *et al* (1999) showed that the interruption of the water exchanges between the Ionian and the Levantine basins triggered an increase in salinity in the Levantine basin.

**5)** *Line 35 – "The modification of water transport led to modified nutrient distribution that can alter local productivity.", Need references.*

Sentence changed to " Also, changes in the North Ionian Gyre circulation triggered the so-called Bimodal Oscillating System (BiOS) that influences phytoplankton bloom in the Ionian Sea through the modification of water transport that led to modified nutrient distribution and altered local productivity (Civitarese et al, 2010). "

**6)** *Line 36 – "short residence time of water." How long? Need references.*

We added the precision (about 100 years) and refered to Robinson et al 2001.

**7)** *Line 38 – "that changes in these conditions can trigger important circulation changes, ultimately leading to changes in", three times the word "change" in the same sentence.*

We thank the reviewer for this remark. We changed the sentence to " These events show that a semi-enclosed basin with short residence time of water (about 100 years) such as the Mediterranean is highly sensitive to climate conditions and that perturbations of these conditions can modify the circulation, ultimately leading to changes in the biogeochemistry. "

**8)** *Line 40 – "The Mediterranean is connected to the global ocean by the narrow Strait of Gibraltar through which transport contributes substantially to its water and nutrient budgets.", Transport of what? The link between the Strait of Gibraltar and the rest of the paragraph is unclear.*

We modified the sentence and moved it at the beginning of section 3.2.2. " The Mediterranean is connected to the global ocean by the narrow Strait of Gibraltar. Water masses transport through this strait contributes... "

**9)** *Line 42 to 46 – "Future climate projections yield […] the western basin for greenhouse gases high-emission scenarios and…" Modify, "Future climate projections with greenhouse gases high-emission scenarios…"*

Changed

**10)** *Line 47 – "In one of these MTHC weakening scenarios, Herrmann et al. (2014) show, in addition, a vertical stratification increase (Adloff et al., 2015)." Herrmann et al., 2014 or Adloff et al., 2015?*

Adloff et al, sentence changed

**11)** *Line 52 – "mixing that bring together available nutrients and phytoplankton", not clear.*

Phytoplankton cells can't swim against currents and therefore need current to encounter nutrients.

Sentence changed to ""mixing that brings nutrients to phytoplankton"

**12)** *Line 65 – "as a result of density changes", not clear, do you mean less stratify?*
Sentence changed to " density changes (increased stratification isolating the upper layer from the rest of the water column). "

**13)** *Line 74 – "…chlorophyll–a concentrations, plankton biomass…", Chlorophyll-a concentration is a proxy of phytoplankton biomass, please clarify.*
Chlorophyll concentration is linked to phytoplankton biomass through the chlorophyll-to-carbon ratio in the planktonic cells. In this version of PISCES, the Chlorophyll-to-carbon ratio is fixed. Therefore, chlorophyll concentration and phytoplankton biomass follow similar trends in response to nutrient and climate change. We chose to show the evolutions of both chlorophyll and plankton biomass in order to keep the results in this article as easily comparable as possible with other studies.

**14)** *Line 83 – "In section 3.3, we expose the temporal evolution of the main nutrients, their budgets in present and future conditions and discuss their impact on the biogeochemistry of the Mediterranean Sea.", You should discuss your result in the section 4 discussion.*

Sentence changed to "In section 3.3, we expose the temporal evolution of the main nutrients, their budgets in present and future conditions and discuss their impact on the biogeochemistry of the Mediterranean Sea in section 4."

**15) Line 117 – "by up to 3 K by", temperature in Kelvin scale?**
Temperatures in the model are in degrees Celcius. We kept the temperature difference in the international temperature unit: K

**16) Line 121 – "0.5 (practical salinity scale)", not in pratical salinity unit?**

Changed to "practical salinity unit"

**17) Line 127 – "reduced vertical mixing may also reduce nutrient supply to the surface waters. A reduction in deep convection may also tend to reduce the loss of P and N to the sediment.", Is it not what you want to test? Why do you present this assumption here, in the section "2.2 The SRES–A2 scenario simulation"?**

We thank the reviewer for this remark. We removed the sentence.

**18) Line 178 – "the effects of climate and biogeochemical forcings". You used the expressions "climate and biological forcings" and "climate and biological changes", choose one of them.**
We harmonized throughout the text by using " climate and biological changes "

**19) Line 201 – "surface average chlorophyll concentrations in the top 10 meters of the CTRL and HIS simulations, and from satellites estimations", it is chlorophyll-a concentration, source of data?**
It is chlorphyll a concentration in both data and model. The data come from the MyOcean product (http://marine.copernicus.eu)

**20) Line 225 – "analysis reveals much greater variability depending on the region", for which regions? It is important for your results.**
We added the sentence "For instance, the Balearic Sea is more sensitive to warming than the rest of the western basin, and the eastern basin has a more intense warming than the western basin (up to 3 K warming in the eastern basin and in the Balearic Sea). Also, the surface salinity in the Aegean Sea increases more than the other regions. "

**21) Line 386 – "For instance, the P rich area between Crete and Cyprus is no longer observed in the 2080–2099 period (Figure 9). Moreover, Figure 10 shows that this area matches a productive zone observed in the 1980–1999 period.", It is the only area in the Levantine basin with some phosphate, nitrate and production values different from zero simulated in 2080-99. Are you sure about your observation?**
The color scale of the figure makes it difficult to see, but the phosphate concentration in the eastern Mediterranean is very close to 0. We changed the sentence to "around Crete and Cyrpus" to avoid confusion.

**22) Line 388 – "The primary production integrated over the euphotic layer (0–200 m) is reduced in our simulation by 10 % on average between 1980–1999 and 2080–2099. However, Figure 10 shows a productivity decrease of more than 50 % in areas such as the Aegean Sea and the Levantine Sea.", Provide time series, as in figure 3.**
In order to make the argument clearer and since it has been suggested by other reviewers, we decided to include difference maps instead of the 2080-2099 maps.

**23) Line 397 – "For instance, around Majorca Island, Corsica and Cyprus, changes in local concentrations of nutrients have substantial effects on primary productivity.", Ok for Majorca, but I cannot see something with Corsica and Cyprus. There is also no values provided to evaluate these changes.**
Sentence changes to " For instance, around Cyprus, changes in local concentrations of nutrients (decrease of about 50~\% in phosphate concentration) have substantial effects on primary

productivity (decrease from 40-50 gC m-2 year-1 to 20-30 gC m-2 year-1). "

**24) Line 413 – "Sea, the northern Levantine basin and the South Adriatic.", In Fig 11, it is the South of the Levantine Basin and the North of the Adriactic which are P-limited.**
As stated in the sentence before, these regions are N and P colimited (see figure 12).

**25) Line 418 – "Figure 12 shows the average depth of the simulated DCM for the period 1980–1999 and for the period 2080–2099.", results for the CTRL not shown, why?**
We chose not to show the CTRL values because we did not want to overload the article with figures. Plus, the DCM does not vary much between the simulations nor during the 21st century.

**26) Line 429 – "At the DYFAMED station, the average DCM depth is unchanged but surface concentration is reduced." A change from 1.10-7 to 0.75.10-7 g m-3 = 1.10-4 to 0.75.10-4 mg m-3. Units are certainly wrong…**
Units and figures have been corrected

**27) Line 432 – "the subsurface maximum in the present and future periods is located at the same depth (100–120 m), but the average productivity is reduced by almost 50 %," Where can we see this? Chlorophyll-a concentration ≠ productivity.**

We changed the sentence to "chlorophyll concentration is reduced"

**28) Line 439 – "Table 4 reports total chlorophyll production in the 1980–1999, 2030–2049 and 2080–2099 periods of all the simulations in all Mediterranean subbasins Adloff et al. (Figure 2 2015)." Why did you quote this reference here?**

Because the subbasins are descried in the Adlof et al article.

**29) Line 496 – "and modification of the physical ocean (vertical mixing, horizontal advection, ...).." Modification of physical processes. Need references.**
We added references to Ludwig et al 2009, Krom et al 2010 and Santinelli et al 2012.

**30) Line 497 – "Nutrient fluxes from these sources." Which ones?**
Sentence changed to " Nutrient fluxes external sources (rivers, aerosols and Gibraltar) may evolve separately "

**31) Line 514 – "In these regions, the effects of nutrient runoff changes seem more important than climate change effects (see Table 4)." provide the percentages, and discuss these results.**
Tables 2, 3 and 4 were corrected to add the percentage of concentration change for each period in comparison to the 1980-1999 period. In the Adriatic basin, table 3 shows that riverine nitrate discharge is responsible for 41 % increase in nitrate concentration over the simulation period. In the CTRL_RG simulation, nitrate concentrations are similar to the CTRL_R simulation showing no influence of Gibraltar inputs in this region. Finally, nitrate concentrations in the HIS/A2 simulation are close to the CTRL_R values showing that most of the nitrate evolution in this region is linked with riverine discharge.

**32) Section 4.2 Climate change scenario. I do not see the point of this section. You decide to use the A2 scenario and already justified it in the introduction.**
This section is provided as an encouragement to develop climate change scenarios over the Mediterranean in order to assess uncertainties in the current results.

**33) Line 537 – "Nutrient concentrations in the intermediate and deep layers were shown to be slightly underestimated in comparison to measurements (see appendix)." Provide values.**
" Nutrient concentrations can be underestimated by up to 50~\%, in particular in the deep eastern basin. "

**34) Line 543 – "Model values were not corrected to match data, and we are therefore conscious that the uncertainties in the representation of present–day biogeochemistry by the PISCES model may be propagated in the future." This is an important decision that needs to**

***be justify.***

In order to study the effect of climate and biogeochemical changes as perturbations we needed to start from a relatively stable initial state. This initial state has been obtained thanks to a long term spin-up simulation. The downside of this approach is the formation of a bias from the present day biogeochemical state. After quantifying the model bias against available observations, we could have corrected the model values (for instance multiplying nutrient concentrations by a factor to correct for the underestimated values in the deep layers). However, such a correction seems dangerous when modeling future evolution of the Mediterranean because threshold effects, and non linear processes are frequent in marine biogeochemical reactions. Therefore, artificially and arbitrarily correcting values may lead to masking some reactions.

***35) Line 571 – "In particular, nutrient inputs at Gibraltar have substantial consequences" Provide the percentages, and discuss them.***

"Results from Figures 4a and 5a and Table 4 indicate that the increase in nutrient inputs from Gibraltar at the end of the century are responsible for a 2.5 % increase in chlorophyll concentration in the western basin during the 2080-2099 period." lines 622-624

**Responses to reviewer 2.**

We thank the reviewer for their very constructive comments. We added Figures to summarize the nitrate and phosphate budgets and fluxes in the different periods we studied.

*lines 160-165: Although I fully sympathize with authors regarding the difficulty to have fully consistent source for riverine water and nutrient discharge, and I am not against the choice the authors made, I would suggest authors to briefly discuss the potential impact associated to the incoherency between these, for instance showing how big this incoherence is.*

Adloff et al (2015) evaluate the changes in runoff in the HIS/A2 simulation. Their Table 2 shows that the total freshwater runoff to the Mediterranean is lower than the Ludwig et al 2009 estimate (by about 30 %). They found approximately 27 % decrease in total runoff by the end of the 21st century. This trend is consistent with the deceasing trend found by Ludwig et al (2010). However, the 2050 estimates of freshwater runoff from Ludwig et al (2010) are only 13 % lower than the 1970 and 2000 estimates. Therefore, it seems that the freshwater runoff decrease in the physical model is more important than in the nutrient runoff model. This may result in higher nutrient concentrations at the river mouth. However, nutrients from river discharge are consumed rapidly at proximity of the river mouth and we believe this potential higher concentrations don't have a large impact of the results.

We added these details in Section 2.4 (lines 178-187)

*Section 2: there is no mention of atmospheric deposition of nutrient in all the methods section. Only in the discussion (501 and 502) authors state that deposition was not considered because there are no future estimates of nutrient deposition up to 2100. Is deposition completely neglected or just kept constant at present day value?*

In the simulations presented initially in the article, deposition is completely neglected. However, we performed extra simulations that are described in the discussion section. These new simulations include total nitrogen deposition from anthropogenic and natural sources modeled with the LMDz-INCA global atmospheric model (simulation labelled HIS/A2-N). Another simulation (HIS/A2-NALADIN) includes both nitrogen from LMDz-INCA and natural dust deposition modeled with the high resolution regional model ALADIN-Climat. These 2 atmospheric models are respectively available for the 1997-2012 and 1980-2012 periods which we repeat during the entire simulation period. Description of the results are included in the discussion section (lines 593-606).

*Lines 205-210: given the errors at the mouth of the Nile, I suggest to add some more detail on the source of the data for this rivers and the uncertainty associated (e.g. see above).*

The difference between model and satellite estimate of chlorophyll around coastal areas is likely linked with the uncertainty of satellite estimates in such areas. Satellite estimates are based on analysis of the water color. Therefore, in shallow, turbid areas, they often interpret the high concentrations of colored material as chlorophyll (see Morel and Gentili 2009). We added this reference. See lines 234-235: "Moreover, Morel and Gentili (2009) show that satellite estimates have a systematic positive bias in the coastal regions because of the high concentrations of colored dissolved organic matter."

*Lines 210-220: while figure A2 immediately shows area where the model has higher or lower skills, I strongly recommend to add some numerical measures of the ability of the model to capture observed data. Furthermore, authors state that model correctly simulates a DCM, however figure A1 clearly shows how the DCM is much deeper than the data shows (roughly at twice the depth). Authors briefly mention this later in the paper, but I suggest to anticipate this here. It would be also interesting to see a comparison of T, S, nutrient profiles in the same*

*location, to understand the reasons of a deeper DCM*

We added some precisions in section 3.1 (lines 201 to 216). We also added figures in appendix to show that the overestimation of the DCM depth is likely due to the overestimation of the nutricline depth as shown in figures A1 and A2.

*lines 254-256: as authors state, phosphate start increasing in CTRL_RG only at the end of the century, therefore the "strong link" between phosphate and Gibraltar is "proofed" only at the end of the century, for most of the simulation surface phosphate seems to be close to CTRL and CTRL_R despite figure 5 shows quite different P influx around 1990 (positive) and 2040 (negative).*

Reviewer is right. Figure 5 shows that Gibraltar fluxes are decreasing in the first half of the simulation period and increasing after 2040. The same trends are observed in the surface and intermediate western basin for phosphate concentrations in the HIS/A2 simulation but not in CTRL_RG. As shown by Table 2, the overall effects of Gibraltar exchanges of phosphate have limited effects on phosphate concentrations. The trend we observe in Figure 4a and 4b may be the sign that Gibraltar exchange effects are visible in the HIS/A2 simulation only after climate change has affected stratification. This paragraph has been rewritten to clarify.

*Lines 263-265: the trend highlighted here is apparent only in the A2 forced simulation, raising the doubt that it could be linked to a spin-up issue (see below for other examples on this). Authors explained the initialisation procedure (lines 170-173) although this is not fully clear: all simulations started from the same restart (and in this case initial trend could be due to adjustment to new forcing, particularly in the climate case), or all scenarios have been run for more than 115 year since that initial common restart (and in this case why the trend is only in A2)?*

All simulations start from the year 1980 from the same restart file after one long spin-up. Therefore, the trends observed in the A2 forced simulation are linked with climate change and/or biogeochemical changes only.

*Lines 265-267: the interesting decennial cycle is not evident only in 3e (where is actually weak) and 3f, but also on the surface. Can authors suggest some mechanism for this? Is this a cycle in intensity of stratification? Is this associated to cycles in the atmospheric patterns?*

The Mediterranean Sea is prone to large decadal variability due to various internal mecanisms and external forcings (e.g. BiOS, EMT, WMT). We prefer not to call it "cycle" as this variability is probably not regular in time. The Mediterranean Sea model (NEMOMED) used to drive PISCES in this study is able to reproduce at least partly this variability (Beuvier et al. 2010, Herrmann et al. 2010, Somot et al. 2018) and it is expected to find its signature in the biogeochemistry variables. However, we would like to underline that the design of the historical and scenario simulations used here is not adequate to draw robust conclusions on the decadal scale. Indeed, some of the physical forcings (river, Near Atlantic) are only evolving every 10 years and the Ludwig et al. nutrient inputs do not represent interannual variability after 2000. Therefore, we prefer not to comment further on this decadal variability. Other modelling setting (hindcast, reanalysis) may be more suitable but are out of the scope of this study.

See for example:
Somot S., Houpert L., Sevault F., Testor P., Bosse A., Taupier-Letage I., Bouin M.N., Waldman R., Cassou C., Sanchez-Gomez E., Durrieu de Madron X., Adloff F., P. Nabat, Herrmann M. (2018) Characterizing, modelling and understanding the climate variability of the deep water formation in the North-Western Mediterranean Sea. *Climate Dynamics,* 51(3), 1179-1210, doi: 10.1007/s00382-016-3295-0

Herrmann M., Sevault F., Beuvier J., Somot S. (2010) What induced the exceptional 2005 convection event in the Northwestern Mediterranean basin ? Answers from a modeling study.
*J. Geophys. Res. - Ocean,* 115 (C12), doi:10.1029/2010JC006162

Beuvier J., Sevault F., Herrmann M., Kontoyiannis H., Ludwig W., Rixen M., Stanev E., Béranger K.,

Somot S. (2010) Modelling the Mediterranean Sea interannual variability over the last 40 years: focus on the EMT. *J. Geophys. Res. - Ocean*, *115*(C8), 1978-2012, doi:10.1029/2009JC005950.

*Line 259 vs 269: authors claims that there is a "slight accumulation of phosphate in the deep Western basin" while "the evolution of nitrate concentration shows marked accumulation in all region". I could be wrong (this is simply a visual calculation), but focussing on A2 trend from 1980 to 2100 figure 3f shows an increase from approx .15 mmol/m3 to 17.5 mmol/m3 (+16%), figures 4e and 4f show a similar relative increase, while 4b and 4c a smaller relative increase. Even if we compare these with the CTRL*
*simulation, the difference between the P accumulation in the deep Western basin and the N accumulation is not that big as the qualification "slight" and "marked" suggest.*
Reviewer is right, we suppressed the adjectives "slight" and "marked" and gave percentage values.

*Lines 274-277: Can authors explain why riverine discharge seem to have more impact on N than P? How much is simply due to the different evolution of the forcing, and how much is due to internal dynamics?*
Ludwig et al (2010) show that even if the global river discharge of N and P over the Mediterranean increases according to their scenario, the N/P ratio of some of the largest rivers is changing. Overall, nitrate total discharge is increasing faster than phosphate total discharge, leading to an increase in the N/P ratio of river discharge over the 21st century (see Figure 1 below).

Moreover, our simulation shows that the Mediterranean is largely P limited. Therefore, phosphate entering the Mediterranean from external sources tend to be consumed faster than nitrate. The figures and tables included in the article are annual and inter-annual average concentrations. They represent the average biogeochemical state of the Mediterranean after nutrient consumption for biological production. Therefore, phosphate concentrations are always low in the P-limited Mediterranean.

[Figure]

*Figure 1: Average N/P ratio of the total riverine discharges of the Mediterranean. Data from Ludwig et al (2010).*

*Lines 283-285: I suggest authors to clarify what they means by "linked with nutrient exhaustion". The link between stronger stratification and lower surface nutrient is well established in literature (due to lower winter mixing), what nutrient exhaustion add to this mechanism and do authors have supporting evidence for this?*

Reviewer is right, stronger stratification and lower surface nutrients are linked by weakest winter mixing leading to less nutrient renewal in the surface layer. In a nutrient-limited system like the Mediterranean, nutrient consumption in surface is fast and the surface is quickly nutrient-poor during stratification periods because of biological production. We changed the sentence to "vertical stratification leads to a decrease in surface layer nitrate concentrations, probably linked both with lower winter mixing and nutrient exhaustion through consumption by phytoplankton."

*Line 301: in the absence of statistical measure of the trends, nutrient fluxes at Gibraltar seems characterised by a high interannual variability until about 2060 rather than a coupled decreasing-increasing trend.*

We calculated a statistical linear regression between the incoming fluxes through the Strait of Gibraltar and time for the first and the second half of the simulation period (before and after 2040) using the Python tool SciPy Stat (www.scipy.org). Statistical indicators are summarized below. Slope indicates the slope of the linear regression, R is the correlation factor, standard error is the error of the slope estimate.

| Period | Slope | R | P-value | Standard error |
|---|---|---|---|---|
| 1980-2040 | -26 | -0.48 | < 0.001 | 6.3 |
| 2041-2099 | 30 | 0.79 | < 0.001 | 3.2 |
| 2030-2040 | -9 | -0.76 | <0.01 | 2.8 |

*Table 1 Statistical trends of phosphate incoming flux through the Strait of Gibraltar.*

| Period | Slope | R | P-value | Standard error |
|---|---|---|---|---|
| 1980-2040 | 1.43 | -0.48 | < 0.001 | 0.3 |
| 2041-2099 | 1.74 | 0.80 | < 0.001 | 0.2 |
| 2030-2040 | -0.55 | -0.77 | < 0.01 | 0.2 |

*Table 2 Statistical trends of nitrate incoming flux through the Strait of Gibraltar.*

This table shows that both the decreasing and the increasing trends during the 2 simulations periods are statistically significant. However, the R value for the 1980-2040 period is low. Most of the decrease is actually observed during the 2030-2040 period (R=-0.76 and -0.77 for phosphate and nitrate respectively).

We changed this section to: "At the end of the 21st century, incoming fluxes of nutrient have increased in the scenario simulation by about 13 % (difference between the 2080-2099 and 1980-1999 periods). But this significant increase (linear regression reveals a positive slope with correlation coefficient greater than 0.75 and p-value < 0.001 for the second half of the simulation period) follows a decrease of over 20~\% in incoming nutrient fluxes between the 1980-1999 and the 2030-2049 periods. Most of the decrease is observed between 2030 and 2040 (decrease 15 and 1 Gmol/month for nitrate and phosphate respectively during this decade)."

*Lines 300-304: authors correctly state that the relative increase in the influx is higher than the relative increasing of the outflux, but what's the difference in absolute term? Is there a change in the net flux?*

We thank the reviewer for this remark. If the relative changes in incoming and outgoing fluxes seem to indicate an increase in the net incoming flux, the absolute values seem to show a rather steady net flux between the beginning and the end of the century. Net flux at the beginning of the century is around -83 Gmol/month for nitrate and -3 Gmol/month for phosphate. At the end of the century, the fluxes are about -80 Gmol/month and -2.5 Gmol/month for nitrate and phosphate respectively. We added these details in the text. Net water, salt and heat fluxes through the Strait of Gibraltar are shown by Adloff et al (2015) to increase at the end of the 21[st] century in the A2 scenario simulation.

*Line 326: similar to comment on 263-265, could the sudden drop be justified by the adjustment to the new forcing (spin-up)?*

This is a good question. We do not believe that the trends observed are linked with adjustment to the new forcings because the trends observed in Figures 4 and 5 are longer than the 1980-1999 period. For instance in Figure 5c, the decrease in nitrate concentration lasts from 1980 to the 2020s. Similarly, the abrupt change in trend in Figure 5e happens before 1999. The changes in physical forcings in the A2 scenario are continuous over the simulation period. No abrupt change in the forcings is applied between the HIS and the A2 periods. Therefore, we do not believe that "spin-up like" drifts are observed in our results.

*Lines 335-370: it is not clear if here authors are still discussing sedimentation fluxes, or rather the global nutrient budget. If the latter I recommend to separate this part with a different sub title (and develop this around the new suggested figure-table). Also, if the latter is true, I suggest to clarify the sentence "the sum of nitrogen fluxes in the Mediterranean": is this the total net flux in the Mediterranean?*
Reviewer is right, this is a new section. We separated this section under the title: "Summary of the phosphate and nitrate evolution under climate and biogeochemical changes in the Mediterranean". We changed the first sentence to "In general, the sum of nitrogen net fluxes in the Mediterranean basin (Riverine, Gibraltar and sedimentary sources and sinks) increases by "

*Line 354-356: authors seems to suggest that the accumulation of phosphate could be due by the decreased primary production, however this is in contrast with the fact that P is the more limiting nutrient in most of the domain (an accumulation in P should lead to an increase in primary production in a P limited environment). Authors should clarify the mechanisms behind the observed trends and the interaction among those mechanisms.*
The P accumulation referred to in this paragraph is the one observed below the mixed layer in the least productive parts of the water column. We added the precision.

*Section 3.4: from my understanding of the PISCES model, sedimentation of particulate nutrient is mostly driven by primary (sinking phytoplankton) and secondary (faecal pellet) productivity. If this is true, I would suggest authors to use the patterns observed in the biological productivity to discuss and interpret the sedimentation fluxes (an possibly anticipate 3.4 before sedimentation)*
Reviewer is right, sedimentation in PISCES is largely influenced by biological productivity. We added a sentence on this matter in section 3.4. However, sedimentary fluxes are also influenced by vertical circulation. In Figure 8 of the article, sedimentary fluxes are highly variable. This may be linked with the sensitivity of the Mediterranean to extreme circulation events such as EMTs that disrupt the vertical circulation.

*Section 3.4: authors discuss at length nutrient limitation presenting co-limitation as a widespread condition in most area of the Mediterranean. To my knowledge there is quite an extensive literature on P limitation in the Mediterranean, particularly in the Eastern basin (e.g. a long list of publications by Krom and co-authors), and authors do not refer to any of this. I strongly suggest to include these in the discussion, compare the results from the model with those findings and suggest potential reasons for the difference. (By the way, I want to emphasize that I have not contributed to any of those papers)*
Reviewer is right. Although we are aware of this extensive literature, we failed to mention it in this section. We include the references in the section. Our study finds that most of the Mediterranean is N/P co-limited because of the way we calculated the limitations and co-limitations. Considering how low nutrient concentrations are in the Mediterranean and how low the nutrient limitation factors are, we consider that the small difference between the limitation factors indicate that the Mediterranean is both limited in P and N. The authors who previously studied the Mediterranean N/P ratio and nutrient limitations derived the limiting nutrient using the N/P ratio of the water (see Krom et al 2010). Therefore they identify the most limiting nutrient. We added the following lines in discussion section 4.3 (lines 695-704) "The modifications of chlorophyll production and plankton biomass are linked to changes in nutrient limitation (Figure 12). Our finding that most of the Mediterranean basin is N and P co-limited seems in contrast with previous literature on the matter (see Krom et al 2005, 2010 ; Pujo-Pay et al 2011 and references therein). These authors found from analyses of the N:P ratio of the waters a clear phosphorus limitation in the major part of the Mediterranean. The discrepancy between our results and literature estimates comes from the way we calculate nutrient limitations. Considering how low nutrient concentrations are in the Mediterranean and how low the nutrient limitation factors are, the small difference between the limitation factors indicate that the Mediterranean is both limited in P and N. Finding no clear definition of nutrient co-limitation, we propose that N and P are co-limiting when the difference in limitation factors is less than 1 %. This definition of nutrient co-limitation applies well to the Mediterranean case because of the very low nutrient concentrations."

*line 409: the use of "thus" suggests that as a consequence of the increased P-limitation in the eastern basin, the surface P concentration will further decrease? Could authors clarify what is the positive feedback between P limitation and P reduction?*
We thank the reviewer for this remark and removed this confusing sentence.

*Lines 438-439: authors state that "The changes in DCM we observe combined with external nutrient input changes result in 17 % reduction in integrated chlorophyll production", however the DCM is a consequence of the chlorophyll production and not a cause. DCM is simply the location of the sub-surface maximum of chlorophyll, therefore, unless authors clarify the meaning of that sentence, is the reduction in the chlorophyll production that leads to the changes in the DCM*

We changed this sentence to "changes in circulation combined with external nutrient input changes..."

*line 468-470: as above, could these be a consequence of the A2 model still adjusting to new atmospheric forcing?*

The apparent stabilization of plankton concentrations appears before 2000 (around 1998). This shows that this is not linked to the A2 forcings adjustment but rather to a response to nutrient conditions.

*Line 512-514: the fact that coastal area (and in particular the Adriatic sea) is largely influenced by coastal nutrient inputs is not a new finding and authors should acknowledge the past literature*

We added a reference to Spillman et al 2007.

*Conclusions: the conclusions are too high level and therefore miss to emphasize the importance of the new findings. For instance the fact that the Western Med is more influenced by the Gibraltar influx than the Eastern Med (lines 590-596) is not that surprising. Furthermore, authors cite their work emphasizing that atmospheric deposition can bring up to 80% of phosphate in some region of the Mediterranean sea. As asked earlier, authors do not clarify if atmospheric deposition is completely neglected in this simulation or simply kept constant: is the former, a lot of the results and discussion are heavily biased by the simplifying assumption and authors should careful and rigorously discuss how this simplification affect all the results presented.*

We modified the conclusion section and referred to the HIS/A2_N and HIS/A2_NALADIN simulations we mentioned in discussion.

*Minor comments:*
*line 46: this sentence "the Mediterranean thermo- haline circulation (MTHC) may significantly change with a consistent weakening in the western basin for greenhouse gases high-emission scenarios..." needs clarification: does authors means that the MTHC weakens in the western basin in climate change scenario characterized by high emission scenario?*

We clarified the sentence "As a result of these changes, the Mediterranean thermohaline circulation (MTHC) may significantly change with a consistent weakening in the western basin and a less certain response in the eastern basin **in climate change scenarios characterized by high greenhouse gases emission**"

*Line 76: which daily 3D forcings are needed by the biogeochemical model? Do authors refer to boundary condition at Gibraltar?*

These forcings are the physical drivers of the circulation such as wind, pressure and precipitation. We added the precision.

*Line 112: "that" needs to be removed*
Done

*line 145: Although I understand that authors refer to original manuscript for details, I would suggest authors to briefly explain (or show within a map) the extent of the buffer zone*

The dimensions of the buffer zone are explained line 94: "The Atlantic boundary is closed at 11°W and tracers are introduced in a buffer zone between 11°W and 6°W."

*line 155: Authors state the four largest rivers of the Mediterranean and Black Sea are the Rhone, the Po, the Ebro and the Danube. Although I agree that these are 4 important rivers, these are surely not the 4 biggest one. Acording to Ludwig 2009 (cited by the authors) the Danube mean flow is 6573 m3/s, Rhone 1721m3/s, Po is 1569m³/s and Ebro is 416 m3/s. Clearly the Nile is the largest, and the Dniper is also bigger than the Ebro*

We thank the reviewer for pointing out this mistake. We corrected the sentence to "4 of the largest".

*lines 158-160: this sentence does not flow correctly in English, and is not clear, I suggest authors to revise it.*

We changed the sentence to : "Ludwig et al. (2010) point out some substantial changes in the nutrient and water budget in specific regions. However, total riverine nutrient input is not drastically changed at the basin scale."

*Line 315: nitrate needs to be capitalised after the full stop*
Changed

*line 540: If the authors want to suggest that the loss of nitrate in the CTRL run can be due by an underestimation of riverine fluxes in the CTRL riverine forcing, I recommend authors to rewrite the beginning of this sentence and explicitly state that, instead of simply referring to the discrepancy with A2 (since the latter can't influence CTRL)*
We changed the sentence to "the low riverine discharge"

*figures 3 and 4: it is very hard to distinguish between grey, blue and green line. Al- though I recognise that the quality of the picture in the PDF is not at the highest, I strongly suggest the authors to use more contrasting colours.*

As suggested by other referees, we modified these figures. The scale was changed, colors and line width was improved and we included average concentration values for the different periods considered in the article. We hope this makes the figures more easily readable.

*Figures 8-9-10: I suggest authors to consider to modify the 2080-2099 panels by show- ing a map of the difference with the 1980-2000 period to better highlight the evolution of N,P and primary production*

We changed these figures to include the relative differences.

*Reviewer 3.*

*This paper addresses the impact that climate change and future riverine nutrient inputs will have on the biogeochemistry of the Mediterranean Sea over the period 1980-2100 using a high resolution coupled NEMO/PISCES model. This paper is important for the scientific community as it is the first time a transient simulation on the response of the biogeochemistry of the Mediterranean Sea to climate change has been run. The authors separate the individual effects of the different scenarios to help determine the reasons for the future biogeochemical changes that the model predicts. In addition they looked at both the impact of each scenario to nutrients and the phytoplankton and zooplankton communities. This study concludes that nitrate concentrations in the Mediterranean are likely to increase in the future while there is no change in phosphate concentrations. They further predict a decrease in net primary productivity. In general, the use of English is good although there are some areas which need clarification which I have highlighted below. However I have some concerns regarding the initial conditions of the model and the analysis of model results which need addressing before this can be published. This review will start with more general comments before detailing more minor changes.*

*My main concern is regarding the initial nutrient conditions in the model. The authors state on line 212 that there is "some underestimation of nutrient concentrations" and again on lines 536-537 that "nutrient concentrations are slightly underestimated" but they do not quantify this. I was surprised when looking at Figure 4 to see deep water nitrate concentrations between 4 and 4.5 μM in the Western Mediterranean Sea. I was expecting to see nitrate concentrations on the order of 8-9μM and hence I do not call this a slight underestimation. Although there is a slight W-E gradient in nutrient concentrations within the model it is not anywhere near as strong as observations suggest. Together with the fact that nitrate in the DW is decreasing in the control scenario suggests to me that the model is not be capturing the biogeochemical cycling of nitrate correctly and raises the question of the validity of future model results. How does this underestimation of nitrate concentrations (and to a lesser extent phosphate concentrations) in the Western basin impact the results of changing circulation such as decreased deep water formation and increased stratification? Would this have a major difference on results? Can the model really predict future changes due to climate change if it can't predict present day conditions correctly?*

This point was also outlined by another reviewer. We added in the appendix section vertical profiles of phosphate and nitrate at the DYFAMED station and along the BOUM section. These figures show that nutrient concentrations in the deep Mediterranean are underestimated by the model. However, the nutricline is represented for both nitrate and phosphate, although is it too weak. The model is indeed not representing the correct nutrient concentrations in the deep Mediterranean. However, the main biogeochemical characteristics are represented such as the position of the nutricline, a deep chlorophyll maximum and the west-to-east gradient of productivity. Moreover, we added in figure A1 of the article the chlorophyll profiles for different years at the DYFAMED station in February and in May. This figure shows that the model is able to reproduce a seasonal and an interannual variability in the biogeochemistry of the Mediterranean that generate a surface chlorophyll distribution that is supported by observations (figure 1 and 2).

The decrease of nitrate deep water concentrations in the CTRL may be linked with the unbalanced sources and sinks of nitrate in the Mediterranean. There may be sources of nitrate that we underestimate or fail to include due to a lack of observations (submarine groundwater discharge or sediment remobilization for instance). The same modelling approach, conducted with a higher resolution model (1/12°, Richon et al, 2017), lead also to underestimation of deep nutrients concentration, but with less amplitude. It suggests that changes obtained by increasing the dynamical model resolution improve the dynamics of the model (convections, eddy activities …), but also in consequences biogeochemical variables. Unfortunately, long climate change simulation performed for this study can not be conducted yet with higher resolution.

We believe that the discrepancy between modeled and measured nutrient concentrations are not

impacting qualitatively our conclusions. The results of this study should be looked at qualitatively and the trends should be remembered rather than the absolute numbers. The trends we observe (accumulation of nitrate, decrease of surface productivity) are the result of multiple factors and have the potential to modify nutrient limitation and surface primary and secondary production. Stratification is virtually isolating the surface layer from regenerated nutrients from the deep and the increase in riverine nitrate discharge (impacting the surface layer) will consequently increase surface nitrate concentration. The trends we observe are explained by the changes in sources and sinks of nutrients and by the circulation changes. Therefore, we believe the trends are robust, but we agree that the absolute values of present and future concentrations may be wrong.

Moreover, it is difficult to assess the robustness of future response of climate models. Models with important bias in the present conditions may not perform worse in future simulations than models with good performances in the present. This is mainly because physical mechanisms driving the response to climate change are not the same than the ones driving model bias in the present. Again, the model and scenario we use in this study are the only ones directly available to us at the moment.

*My next concern is in regards to whether the authors include dissolved organic matter inputs though the Strait of Gibraltar and from rivers into the model? On line 291-293 the authors say that*
*"The Mediterranean is a remineralization basin that has net negative fluxes of inorganic nutrients (i.e. organic nutrients enter the basin through the Gibraltar Strait surface waters and inorganic nutrient leave the Mediterranean through the deep waters of the Gibraltar Strait"*

*However there is no mention of dissolved organic nutrients anywhere in this paper. Do the authors include them in the inputs through the Strait of Gibraltar (or in the riverine input)? If yes this needs to be explicitly stated and if no then they are missing a major source of phosphorus and nitrogen in their model (see Powley et al., 2017; 2018). In addition the paper tries to present a nutrient budget based solely on nitrate and phosphate and then use the imbalanced budget to explain the decrease in nitrate in the CTRL model (Line 540). However dissolved organic matter inputs need to be included in the budget so that total N and P inputs and therefore a complete budget can be calculated (see Powley et al., 2017; 2018) In addition I suggest creating a Table summarizing the budget as currently it is difficult to interpret from the graphs. Finally Lazzari et al. (2014) conclude that dissolved organic matter is increasing in their model in response to climate change. Do your results agree? (I know this is not a key result but a sentence regarding this could be added to the discussion)*

Organic forms of phosphorus and nitrate are not included in the version of PISCES we used. They can only be calculated by multiplying the DOC by the Redfiled ratio. However, organic forms of nutrients are not available to phytoplankton. The only forms of organic matter are dissolved organic carbon (DOC) and particulate organic carbon. DOC from river and Gibraltar inputs is included. It can be remineralised to inorganic nutrients and therefore acts as an indirect source of nutrients. However, it is impossible to quantify this remineralization in our simulations. In the HIS/A2 simulation, the DOC riverine inputs are kept constant to the 2000 value over the 21$^{st}$ century. The evolution of DOC incoming and outgoing at Gibraltar is shown below. The three graphs represent respectively total incoming, outgoing and net DOC fluxes through the Strait of Gibraltar in the HIS/A2 simulation (Red) and in CTRL (blue). Numbers in the top right boxes represent respectively the total fluxes for the periods 1980-1999, 2080-2099 and 2040-2059. These figures show that the net DOC flux is increasing throughout the 21$^{st}$ century.

[Figure]

*Figure 1 Influx of DOC through the Strait of Gibraltar*

[Figure]

*Figure 2 Outflux of DOC through the Strait of Gibraltar*

[Figure]

*Figure 3 Net flux of DOC through the Strait of Gibraltar*

In spite of this increase in DOC flux, the concentrations seems to be decreasing in the basin in our simulations (see the following figures). DOC concentrations in the surface layer are increasing in HIS/A2, but the important decrease in the intermediate and deep layers is probably linked with a decrease in vertical water flux.

[Figure]

*Figure 4 Interannual water column concentration of DOC in the eastern basin for all simulations (HIS/A2 in red, CTRL in black, CTRL_R in blue and CTRL_RG in green)*

[Figure]

*Figure 5 Interannual water column concentration of DOC in the western basin for all simulations (HIS/A2 in red, CTRL in black, CTRL_R in blue and CTRL_RG in green)*

Reviewer is right to point out that in the absence of organic nutrient forms, our N and P budget is unbalanced. We add this point in the discussion (section 4.3).

***The results section is very qualitative with little quantitative analysis. Phrases such as slightly increase and significantly increase are common with no data to back them up. In addition I feel that section 3 and especially section 3.3 can be condensed as there is a lot of repetition and is hard to follow in places. This would make the main conclusions and outputs of the paper clearer to the reader. I suggest re-organizing section 3.3 to start with the nutrient budget first, analysing the different inputs and outputs from rivers, Straits of Gibraltar and sediment before going on to look at the effect of the different scenarios to nutrient concentrations. This way you can bring in the analyse from the budgets to explain the concentration trends rather than having to repeat yourself analysing and explaining the trends in nutrient concentrations before you have analysed the causes. I would also in general try and keep the same structure within each section in regards to the analysis i.e compare phosphate first, then nitrate, etc.***
We thank the reviewer for this suggestion. We followed the advice and reorganised the section. Also, as suggested by another reviewer as well, we tried to provide more quantitative analyses of the results.

***General minor comments***
***While I appreciate that you are limited by both data and computational power in your model runs I suggest refraining from using 'external inputs' and instead be specific and use 'fluxes through the Straits of Gibraltar and riverine inputs'. As far as I understand you are not including atmospheric inputs, direct wastewater discharges or submarine groundwater discharges in your model which can all be considered external nutrient inputs.***

***Use Strait of Gibraltar or Gibraltar Strait throughout the paper rather than Gibraltar as Gibraltar is a body of land!***

We thank the reviewer for these remarks and tried to correct the sentences wherever possible

***Detailed minor comments***

*Line 8: Change "coastal nutrients" to riverine nutrients. Coastal nutrient inputs could mean coastal runoff, direct wastewater discharges submarine groundwater discharge.*
Changed

*Line 9: Do you just mean from riverine inputs rather than external sources.*
Sentence changed to "These contrasted variations result from an unbalanced nitrogen--to--phosphorus input from fluxes through the Strait of Gibraltar and riverine discharge and lead an expansion of phosphorus--limited regions across the Mediterranean. "

*Line 27: I thought the last deposited Sapropel was 10,000 years ago not that they have been deposited for the last 10,000 years*
We added precisions in this sentence :"In particular, high stratification events, characterized by the preservation of organic matter in the sediment, known as sapropels, have been recorded through several over geological times, the most recent was recorded 10~000 years ago and lasted about 3~000 years."

*Line 36: Please quantify the short residence time (i.e 100 year timescale) and add reference.*
Precision added. The residence time is around 100 years (Robinson et al. 2001).
Robinson, A. R., Leslie, W. G., Theocharis, A., & Lascaratos, A. (2001). Mediterranean sea circulation. *Ocean currents: a derivative of the Encyclopedia of Ocean Sciences*, 1689-1705

*Line 40-41: Please re-phrase. The word transport in this sentence doesn't make sense.*
We removed this sentence that was out of the scope of the paragraph

*Line 48: I am confused by the Adloff reference at the end. Do they also show this enhanced vertical stratification?*
We rephrased : "In all A2 runs, Adloff et al (2015) show an increase in the stratification index at the end of the 21st century."

*Line 49: remove "lead to"*
Done

*Line 70: Add Heurtas et al. (2012) to the Gibraltar references. Add more references for atmospheric deposition or say 'and references therein'. There have been a lot of studies on atmospheric deposition in the Mediterranean region. What about direct wastewater discharges (Powley et al., 2016) and submarine groundwater discharges (Rodellas et al. 2015). Note also Powley et al., (2017;2018) have calculated a complete nutrient budget for the Mediterranean and these should be referenced somewhere in this paper.*
We thank the reviewer for these suggestions and added more references in this part.

*Line 100: define the SST acronym rather than on line 222*
Done

*Line 111: define the SSS rather than on line 222* Done
*Line 112: remove 'that'*
Done

*Section 2.3 Please add a bit more detail regarding the biogeochemical model and the compartments so the reader has an idea of what is included without having to go to the references (i.e Are there compartments for bacteria, DOM etc?).*
There is no explicit bacterial compartment but bacterial biomass is calculated using zooplankton biomass (see Aumont et al (2006) for details). Organic matter is divided in 2 forms: dissolved organic carbon (DOC) and particulate organic carbon. Other organic nutrients such as phosphorus and nitrogen are not explicitly represented in this version of PISCES but can be derived from the Redfield ratio.

These precisions have been added to the section.

**Section 2.3 Include a sentence regarding why you did not use atmospheric deposition, and other external inputs in this section.**
We added the following sentence in section 2.4:
"We did not include atmospheric deposition as there is currently no scenario for its future evolution. Similarly, we did not include submarine groundwater discharge and direct wastewater discharge as there is to date no climatology for these sources." (lines 152-154).

**Lines 150-155: Please be specific in which MEA scenario you use. None of the four scenarios are called business as usual. Also how did you combine values from the two Ludwig papers together as Ludwig et al. (2010) states that they are not directly compatible with one another.**

The scenario we use is the Order from Strength (OS) from Ludwig et al 2010.  PO4 and NO3 flux are from Ludwig et al. 2010 for both HIS and A2 only DIC and Si are based on Ludwig et al. 2009. There is no incompatibility issue.

**Line 175: Why are 1966-1981 conditions used when the model results are from 1980 onwards? Please specify in the text.**
The 1966-1981 period was chosen to avoid years with too much warming, which are observed as of 1980. During the CTRL, these years are looped over the simulation time (120 years). We present the results on the 1980-2099 time scale. Precisions are added in lines 187-190.

**Line 184: Write minus rather than using the minus sign as it wasn't clear to me what you meant initially.**
Changed

**Line 203 satellite not satellites**
Changed. Note that we added new figures and data in this section to evaluate the model performances.

**Line 212-213: Quantify the error. Compare model results with measurements. (See my main concern)**
We added a value of the underestimation (approximately 50%)

**Line 224: When you say global I assume you mean across the entire Mediterranean. Please clarify**
We changed the word "global" to "basin-wide" for more clarity.

**Line 229-235: I suggest moving this section to where you discuss the budget as no results are given and it confuses the reader**
This paragraph is intended as an introduction to the long section of results following. In these line we present the vocabulary we use afterwards. This is why we consider this paragraph important at this stage.

**Line 230: Add references after "nutrient budgets are highly dependent on external sources".**
References added to Ludwig et al 2009, 2010, Christodoulaki et al 2013, Huertas et al 2012

**Line 244-246: State this later on when you are talking about limiting nutrients.**
We removed the words 'limiting nutrients' from this sentence.

**Line 252: How much does the phosphate concentration decrease by? From what to what?**
Phosphate decreases by about 0.015 mmol/m3 in the surface layer and 0.017 mmol/m3 in the intermediate layer in the first half of the simulation period. It increases in the second half to reach concentrations close to the 1980 value in the surface layer and higher concentrations in the

intermediate layer (by about 0.01 mmol/m3). These precisions were added.

**Line 253: Only use significantly if it is statistically significant. If yes then the state the statistics.**
We removed the word "significantly" from the sentence.

**Line 254: Add "in the surface water in the Western Mediterranean Sea" after phosphate concentrations. The reader shouldn't have to look at the figures to know which water body you are talking about.**
Added

**Line 254-255: What about the comparison between CTRL_RG and CTRL R?**
We added the following sentences: "Figure 7a shows that the difference in phosphate concentrations in the surface layer of the western Mediterranean in the CTRL_RG and CTRL_R simulations is important only at the end of the 21st century (approximately from 2070). Therefore, the similar evolutions of phosphate concentration in HIS/A2 and of incoming fluxes of phosphate through the Strait of Gibraltar throughout the simulation period must be linked with changes in physical conditions. In this very dynamic part of the Mediterranean, changes in physical conditions linked with climate change are preconditioning the western basin to become more sensitive to nutrient fluxes thought the Strait of Gibraltar."

**Line 263: "The eastern part of the basin contains approximately 50 % less phosphate than the western part." Where have you got this data from? Table 3 shows greater phosphate content in Western Mediterranean than Eastern Mediterranean. If you mean concentrations then I would still argue that your model is not showing 50% less.**
Reviewer is right, this sentence is confusing. The eastern basin naturally contains more nutrients as its volume is greater. We mean that the concentrations are approximately 50% less in the eastern basin. We changed the sentence to "Nutrient concentrations in the eastern part of the basin are lower than in the western part (50 % lower phosphate concentration in the surface layer, about 30 % lower concentration in the intermediate layer and about 15 to 20 % lower concentration in the deep layer)."

**Line 265-267: Give quantitative values.**
We modified this paragraph to add quantitative values in the text:"In the surface layer, phosphate concentration decreases in the beginning of the simulation and remains low during the 21st century (from 0.022 mmol/m3 in 1980 to less that 0.015mmol/m3 in 2000, Figure 4d). There is, however, a large annual variability in surface phosphate concentration with peaks up to 0.025mmol/m3 in 2060. But the HIS/A2 simulation values are consistently below the CTRL concentrations showing an important effect of climate change on surface phosphate reduction. We observe in Figures 4e and 4f an accumulation of phosphate in the intermediate and deep layers (17 and 13 % respectively), with large decennial variability of phosphate concentration in the deep eastern basin. In both of these layers, HIS/A2 concentrations are higher than the CTRL concentrations. "

**Line 268: State which scenario you are talking about.**
We modified this paragraph to add precisions and quantitative values.

**Line 272. Reference Figure 5 after Atlantic**
The paragraph was modified and references to figures added

**Line 278-281: Consider consolidating with previous paragraph as there is a lot of repetition (especially concerning impact of rivers of nitrate concentrations).**
We thank the reviewer for the suggestion and merged the paragraph with the previous one.

**Line 293: Reference needs brackets around it.** Changed, reference to Gomez et al 2003 included as well.

**Line 295-298: Put quantitative results.**
This paragraph was modified to include more quantitative values.

**Line 305-307: What about statistics for nitrate?**
The Pearson correlation coefficient is 0.80 (p-value < 1%). We also recalculated the statistic for phosphate because we noticed the previous result was for the entire water column.

**Line 308: I suggest adding a figure showing the net fluxes through the Strait of Gibraltar to show imbalance. Again, I also suggest the authors look at total nitrogen fluxes to be able to determine whether there is an imbalance at the Strait of Gibraltar not just nitrate.**
In order to keep the number of figures, we did not include the net fluxes in the article. However, the Figures are shown below and we added discussion about the net fluxes in the paragraph (see lines 334 to 339).

[Figure]

*Figure 1: Net flux of phosphate throught the Strait of Gibraltar in HIS/A2 (red) and in CTRL (grey)*

[Figure]

*Figure 2:*
*Net flux of nitrate through the Strait of Gibraltar in HIS/A2 (red) and CTRL (grey)*

Done

**Lines 315-320: Is there a difference in the evolution of riverine discharge of nutrients between Western and Eastern basins? Combining everything into one flux makes this impossible to see, but if it differs it would have a large impact on results and may explain the differences to the riverine scenario seen in the two basins.**

We do not have a figure for the riverine discharge of the western and eastern basins separated. However, the figures below show nitrate and phosphate flux from 3 of the major rivers of the Mediterranean: the Po, the Nile and the Rhone. The nutrient outflow from these rivers have important impact on the local productivity.

Figure 3 below shows that the nutrient discharge from these important rivers evolves differently. In particular, phosphate flux from the Nile increases abruptly between 2000 and 2050 while nitrate flux remains low. This important phosphate source in the P-limited Levantine basin explains the high productivity observed at the end of the century in our HIS/A2 simulation.

[Figure]

*Figure 3: Phosphate (top) and nitrate (bottom) fluxes from the Nile (left), the Po (middle) and the Rhone (right) rivers during the simulation period. From Ludwig et al 2010.*

This was corrected, but the sentence was moved to section 3.3.5, line 335.

Done, this sentence was also moved to section 3.3.5.

**Lines 328-332: What happens to sedimentation in the CTRL-R and CTRLRG scenario? This**

*will make the argument stronger whether sedimentation is linked to decrease in vertical fluxes. Also please be more explicit about the process that would increase phosphate and nitrate in the deep water. A lower particulate matter flux to the deep water would lead to lower remineralisation fluxes and therefore lower phosphate and nitrate. Alternatively, higher water temperatures could lead to higher remineralisation and therefore a lower sedimentation flux despite the same flux of particulate matter exiting the surface water. How do the authors know it is not higher temperatures rather than changes in the water flux that alters the sedimentation flux?*

We did not calculate the sedimentation fluxes in CTRL_R and CTRL_RG.
Temperature does not affect remineralization rates in the PISCES model. Therefore, the changes in export fluxes between CTRL and CTRL_R or CTRL and CTRL_RG are only linked with the changes in surface nutrient concentrations and vertical water fluxes.
We looked at the POC export fluxes at 1000m in the different simulations in order to explore the changes in sinking material. Our results show that POC export at 1000m in the HIS/A2 simulation is reduced more than twofold in the 2080-2099 period in comparison to the 1980-1999 period. In the CTRL_R and CTRL_RG simulations, the change in POC export is lower (up to -30%) and in the same order of magnitude than in the CTRL simulation. These observations are in favour of the hypothesis of lower vertical water fluxes explaining the decrease in sedimentation rates. Lower sedimentation coupled with a constant remineralization rate (because remineralization is invariant with temperature in PISCES) leads to the accumulation of nitrate and phosphate in the deep Mediterranean.
*Line 334: A new section heading Is needed before line 334.*
We moved this paragraph in the discussion section 4.4

*Line 334: Into rather than in?*
Changed

*Line 334-335. Compared to when? The start of the model run? When you say the sum do you mean the balance between inputs through the Gibraltar Strait, riverine inputs and sedimentation? Be specific. Also, as mentioned before, I suggest including dissolved organic matter in your calculations and to produce a table to show the balance of the different inputs and outputs of phosphorus and nitrogen.*
We mean in comparison to the beginning of the simulation (1980). "Sum" means indeed balance between inputs and outputs, we modified the sentence to be more specific.
We do not have organic nutrients explicitly represent in the model. However, we added Figure 18 in discussion section 4.4 to summarize the fluxes of nutrients to the basin and the evolution of nutrient budgets.

*Line 350-351: The authors state "changes in Gibraltar exchange fluxes of phosphate seem to have limited effect on Mediterranean phosphate content" but on lines 305-306 they also state "evolution of phosphate concentration in the Western basin is linked with Gibraltar inputs". Which one is it?*
This sentence is confusing. We mean that the fluxes through the Strait of Gibraltar are having an important impact on the western basin, but do not seem to have large impact on the nutrient budget of the entire Mediterranean. We added "global" in the sentence to make it clearer.

*Line 353-356: What about increase in temperature effecting results? Luna et al., (2012) hypothesise increasing deep water temperatures will increase prokaryotic metabolism, thus potentially increasing nitrate and phosphate concentrations. Lazzari et al., (2014) also predict that increasing temperatures increase metabolic rates.*
We thank the reviewer for the reference to Luna et al that we were not aware of. Prokaryotes are not explicitly modeled in PISCES, and nutrient recycling is not a function of temperature in the model. Therefore, temperature increase in our simulations may have effects on planktonic production, but not on remineralization.

We added the following lines in the discussion : "Moreover, Luna et al 2012 hypothesise that the

warm temperature of the deep Mediterranean may be a cause for important nutrient recycling via prokaryotic metabolism. In the version of PISCES used in this study, nutrient recycling is dependant on oxygen, depth, plankton biomass and bacterial activity. Therefore, we could not observe the effects of temperature on nutrient recycling."

**Line 378: Replace "and shows" with "showing"**
Done

**Line 380: Delete as this is repeated and explained in the next paragraph.**
Done

**Line 384" surface phosphate what? Concentrations? Masses?**
Concentration. This was added in the sentence

**Line 386-387: The decrease does not look that clear to me and it certainly doesn't become entirely depleted in phosphate.**
We removed the word 'largely', but phosphate concentration is reduced in all the eastern basin.

**Line 409: Remove would**
Done

**Line 410-414: Why would the phosphate become the major limiting nutrient in areas where primary productivity is reduced?**
Based on the results shown in Figure 11 and 12, the P-limited areas at the end of the simulation period match areas where the most important primary productivity decrease is observed. Our hypothesis is therefore that the imbalance in nitrate and phosphate budgets leading to decrease in the surface phosphate budget drives the surface eastern Mediterranean to a P-limitation. This hypothesis is confirmed by the nutrient budgets in Tables 2 and 3, even though we do not have the budgets of organic nutrients.

**429-430: Reduced by how much?**
We corrected the figure that was using the wrong units and find that the surface concentration in the 2080-2099 period is actually increased by about 25 ng/L in comparison to 1980-1999. This shows that local variability in circulation and biogeochemitry is an important feature of the Mediterranean biogeochemistry.

**Line 433: Are we still talking about chlorophyll or primary productivity?**
Chlorophyll, we changed this confusing term.

**Line 434: What is "it"?**
We replaced by "subsurface chlorophyll concentration"

**Line 441: Remove Adloff reference?**
This reference was formatted wrongly. **It is "Figure 2 from Adloff et al"**

**Line 452? Do you mean riverine nitrate inputs or total nitrate inputs?**
Yes, we added the precision in the sentence.

**Line 460-463: Which scenario are you talking about?**
We rephrased this sentence: "In HIS/A2, we observe lower biomass for both phytoplankton classes across the Mediterranean Sea at the end of the century than at the beginning. "

**Line 463: Add Western Mediterranean Sea after diatom concentrations.**
Done

*Line 465: Add diatoms after concentration*
Done

*Line 478: Add references if other studies have concluded this.*
We are not aware of any studies focusing in zooplankton using NEMO/PISCES. This sentence is an hypothesis based on the authors' experience with the model.

*Line 483: "After all" doesn't make sense in English. I suggest you use "Altogether".*
We thank the reviewer for this suggestion and modified the sentence accordingly.

*Lines 494-497: Add references to this sentence*
References have been added to Ludwig et al (2009), Krom et al (2010) and Santinelli et al (2012).
*Line 514 and 515: When talking about "coastal nutrient inputs" and "coastal runoff" are you only talking about riverine inputs? Be specific.*
We refer to riverine inputs. We tried to harmonize this paragraph to make it more specific.

*Line 518: 'developing' not 'developping'* Corrected

*Line 540: I don't think you can say there's an imbalance in sources and sinks without looking at the organic nitrogen aswell (unless you count nitrate sourced from DOM). There will always be an imbalance of nitrate in the Mediterranean if you only look at external sources as you mention yourself it is a remineralization basin.*
We do not have organic nitrogen in the model. However, nitrate can be recycled from DOC in PISCES and is therefore included in the budget.
We added a sentence on this matter: "Organic forms of nutrients are not included in this version of PISCES. Powley et al (2017) show that organic forms of nutrient are an important part of the Mediterranean elemental budgets. Therefore, we may be missing a part of the N and P budgets in our calculations."

*Line 591-592: Please re-phrase the sentence and be specific to what you are talking about. What inputs?*
We rephrased: "Our results also illustrate how climate change and nutrient inputs from riverine sources and fluxes through the Strait of Gibraltar have contrasted influences on the Mediterranean Sea productivity."

*Line 598: I disagree with this statement. There are lots of nutrients inputs that you haven't considered such as direct wastewater discharges, diffuse runoff, submarine groundwater discharge.*
We changed this part to: "Finally, this study accounts for the changes in fluxes through the Strait of Gibraltar and riverine inputs, but some potentially important sources are missing such as direct wastewater discharge, submarine groundwater and atmospheric deposition. Measurements and models are still missing in order to include comprehensive datasets for past and future evolution of these nutrient sources."

*Line620: Shown rather than showed*
Corrected

*Table 2: It would be more informative to present the numbers per m2 surface area and then comparisons between basins can happen as the Eastern Mediterranean is almost twice the size of the Western Mediterranean. The reader can still calculate the total mass if the surface area is given. Also taking the difference from the control rather than total values might make it easier to spot trends.*
We added in the tables 2, 3 and 4 percentages. These correspond to the relative difference between each period and the 1980-1999 period and help see the trends. We also added on Figures 7,8, 14

and 15 the average concentrations of tracer for each simulation for the periods 1980-1999, 2030-2049 and 2080-2099.

*Figures 3 and 4: Please create a greater contrast between the CTRL R and CTRL RG. These are difficult to see at present.*
These figures were changed

*Figure 5: it would be nice if the net flux of nitrate and phosphate across the Strait of Gibraltar could be added aswell.*
For the sake of the number of figures, we chose not to include the net flux in the figures. However, we discuss it in the text.

*Figures 8,9,10 and 12. I struggled to see the differences between the two time periods which were mentioned in the text. I suggest to produce a figure of the anomaly between 2080-2099 and 1980-1999 rather than having the 2080-2099 figure as it currently is. The changes with time will then be more evident to the reader.*
We produced the anomaly maps and changed the figures

*Figure 10. Units in caption do not match units in figure* Changed
*Figure 12: Why is the HIS/A2 scenario figures not presented aswell?*
The CTRL is not presented. We chose not to show it as we are not discussing it and it is almost the same partition as in the HIS/A2 during the 1980-1999 period.

*Figure 13: Units in caption are different than on figure* Changed
*Figures 14 and15: What are the units? mmol m-3 of what? Carbon?* Yes, corrected
*Figure A1: When were the data points collected? Would the HIS/A2 scenario be a better comparison than CTRL as it actually uses 1980-2000 data?*
We changed the figure to include the comparison of HIS/A2 for the corresponding years
*Figure A.2 Change P4 to PO4 in the label.* Done

[revised manuscript text omitted]

(a) phosphate surface (0–200 m)  (b) phosphate intermediate (200–600 m)  (c) phosphate deep (600 m–bottom)

(d) phosphate surface (0–200 m)  (e) phosphate intermediate (200–600 m)  (f) phosphate deep (600 m–bottom)

**Figure 7.**  yearly average phosphate concentration ($10^{-3}$ mol m$^{-3}$) in the surface (left), intermediate (middle) and bottom (right) layers in the western (top) and eastern (bottom) basin.  Red lines represent the HIS/A2 simulation,  black lines represent the CTRL (with standard deviation),  blue and  light blue lines represent the CTRL_R and CTRL_RG simulations respectively. Colored numbers in the highlighted areas represent the average concentrations in the corresponding simulations for the highlighted time periods.

[Figure]

(a) nitrate surface (0–200 m)  (b) nitrate intermediate (200–600 m)  (c) nitrate deep (600 m–bottom)

(d) nitrate surface (0–200 m)  (e) nitrate intermediate (200–600 m)  (f) nitrate deep (600 m–bottom)

**Figure 8.** Evolution of yearly average nitrate concentration ($10^{-3}$mol m$^{-3}$) in the surface (left), intermediate (middle) and bottom (right) layers in the western (top) and eastern (bottom) basins.  Red lines represent the HIS/A2 simulation,  black lines represent the CTRL (with standard deviation),  blue and  light blue lines represent the CTRL_R and CTRL_RG simulations respectively. Colored numbers in the highlighted areas represent the average concentrations in the corresponding simulations for the highlighted time periods.

[Figure]

Evolution of total river discharge fluxes of nitrate and phosphate (10⁹ mol month⁻¹) to  the  relative difference (in %) between the 2080–2099 and

Evolution of total sedimentation fluxes of N and P (10⁹ mol month⁻¹) in  1980–1999 periods in CTRL (left) and HIS/A2 (right).

Evolution of total river discharge fluxes of nitrate and phosphate (10⁹ mol month⁻¹) to  the  relative difference (in %) between the 2080–2099 and

Evolution of total sedimentation fluxes of N and P (10⁹ mol month⁻¹) in  1980–1999 periods in CTRL (left) and HIS/A2 (right).

**Figure 9.**  Present (1980–1999, top)  interannual average surface (0–200 m)  concentrations of nitrate  (10⁹⁻³ mol m⁻³) in the CTRL (left)  HIS/A2 (right) simulations.  Evolution of total river discharge fluxes of nitrate and phosphate (10⁹ mol month⁻¹) to  the  relative difference (in %) between the 2080–2099 and

Evolution of total sedimentation fluxes of N and P (10⁹ mol month⁻¹) in  1980–1999 periods in CTRL (left) and HIS/A2 (right).

[revised manuscript text omitted]

---

## Referee Report (RR1)

Authors sufficiently addressed all of my comments, and the quality of the manuscript is significantly improved. The contribution of this modeling study in the effort to understand the impacts of climate change on the Mediterranean Sea biogeochemistry is clearer than before. However, some points in the present version of the manuscript have to be restructured. This mainly concerned the uncertainties and limitations associated to this study, and one part of the results (see my major comments). In addition, several minor corrections are necessary (see my minor comments). Therefore, I believe that this paper will likely be a significant contribution and reach the quality standards for publication in the Biogeosciences journal after minor revisions of the manuscript.

Major comments:

1) Sections "4.2 Climate Change Scenario" and "4.3 Uncertainties from the PISCES model": these sub-sections need to be regrouped, completed, and rearranged at the end (or at the beginning) of the discussion section. In order to have a better insight into all uncertainties and limitations of the study (most of them already mentioned by the authors), to consider their influences on the results and processes discussed in the other sub-sections of the discussion (i.e., 4.1 and 4.4), and maybe to discuss some perspectives. For this, this sub-section should have a more general title, for example, "Limitations of the Study", and divided into three parts (at least):

    a.  Climate change scenario: mostly section 4.2
    b.  PISCES model: mostly section 4.3
    c.  External sources of nutrients used (riverine and Atlantic inputs) and missing (atmospheric deposition): scattered in the manuscript

For example, I was expecting to have information in the discussion section about the influence of the following uncertainties and/or limitations on the biogeochemical response of the Mediterranean Sea,

- How the Atlantic condition and Atlantic nutrient concentrations used in the model could influence the biogeochemical results obtained, mainly for the Western Basin? For example, the authors wrote,

> Line 646 - "…the choice of atmospheric and Atlantic conditions has a strong influence on the MTHC."

with no information or discussion about possible consequences on the biogeochemical results obtained in their study.

- How the atmospheric deposition (not represented in the model, line 157 - "We did not include atmospheric deposition…") could influence the fact that the Mediterranean Sea is becoming more P-limited at the end of the century in your study?

2) In the conclusion – between lines 755 and 763 – authors highlighted the differences between the western and eastern basins in their biogeochemical responses to the climate change. This part of the results, which, I believe, is one of the major contribution of the manuscript, was absent in the discussion. There were even some contradictory statements, between the results, discussion and conclusion sections (see below), which make difficult to determine the author's opinion on some key aspects:

> Line 449 – "But the strong difference between CTRL_R and HIS/A2 at the end of the century indicates that vertical stratification leads to a decrease in surface layer nitrate concentrations, probably linked both with lower winter mixing and nutrient consumption by phytoplankton."

> Line 739 – "Stratification may lead to increased productivity in the surface because of the nutrient concentration increase (see also Macias et al., 2015)…"

> Line 751 – "increased stratification, and changes in Atlantic and river inputs, can lead to a significant accumulation of nitrate and a decrease in biological productivity in the surface…"

> Line 760 – "the eastern basin is more sensitive to vertical mixing and river inputs than the western basin […] and the stratification observed in the future leads to a reduction in surface productivity…"

Please, clarify your interpretation of the results in the discussion, in order to support all your statements in the conclusion.

Minor comments:

1) Line 68 – "Macias et al. (2015) simulated […]. They found that […] primary productivity over the eastern Mediterranean basin may increase as a result of density changes (increased stratification isolating the upper layer from the rest of the water column)."

Need to be corrected, because in the abstract of Macias et al. (2015),

> "In the eastern basin, on the contrary, all model runs simulate an increase in surface production linked to a density increase (less stratification) because of the increasing evaporation rate."

2) Line 81 – "This study aims at understanding the biogeochemical response of the Mediterranean to a "business–as–usual" climate change scenario throughout the 21st century."

The objective of the study should be more detailed. Please, highlight, for example, that the study mainly focus on the influence of the external sources of nutrients (rivers, Atlantic).

3) Lines 127 - 139 – "Temperature and salinity increase strongly, leading to a decrease in surface density and an overall increase in vertical stratification. Average sea surface temperature of the Mediterranean rises by up to 3°C by the end of the century. However, the temperature rise is not homogeneous in the basin, regions such as the Balearic, Aegean, Levantine and North Ionian undergo a more intense warming (over 3.4 °C) probably due to the addition of the atmosphere-originated quasi-homogeneous warming with the local effect of surface current changes. The salinity increases by 0.5 (practical salinity units) on average across the basin. In the A2 simulation, the entire Mediterranean basin is projected to become more stratified by 2100 and deep water formation is generally reduced. These variations in hydrological characteristics of the water masses generate important changes in the circulation and in particular in the vertical mixing intensity. The strong reduction in vertical mixing observed in all deep water formation areas of the basin is linked with the changes in salinity and temperature of the water masses."

This paragraph needs to be re-write because there are some repetitions, for example, about the stratification and vertical mixing.

4) Line 166 – "Nutrients input from rivers are derived from Ludwig et al. (2010) before 2000, Dissolved inorganic carbon (DIC) and Si are derived from Ludwig et al. (2009)."

What are included in "Nutrients"? Replace "Dissolved inorganic carbon" by "Dissolved Inorganic Carbon". "Si" means silicates? DIC and Si inputs are derived from Ludwig et al. 2009?

5) Line 190 – "This may result in higher nutrient concentrations at the river mouth. […] However, nutrients from river discharge are consumed rapidly at proximity of the river mouth and we believe these potential higher concentrations don't have a large impact of the results."

If all nutrients from river discharge are consumed near the river mouths, why did you study the influence of the river inputs at the scale of the Mediterranean basin?

6) Line 205 – "This period was chosen in order to avoid including in the CTRL years with too important warming such as the 1980s and 1990s." Need to be corrected.

7) Line 212 – "present–day conditions […] present–day conditions". Need to remove these expressions, or to define them.

8) Line 235 – "from satellite estimations from MyOcean Dataset (http://marine.copernicus.eu)". Which product of MyOcean, quote the full link to access this dataset.

9) Line 236 – "All chlorophyll values in the article and the data are chlorophyll–a." I do not understand. Do you mean that the word "chlorophyll" stands for "chlorophyll-a" throughout the manuscript? If so, it needs to be define when you use the word chlorophyll for the first time.

10) Line 240 – "with values that", are you talking about the difference in magnitude?

11) Line 243 – "Moreover, several studies (see e.g. Claustre et al., 2002; Morel and Gentili, 2009) show that satellite estimates have a systematic positive bias in the coastal regions because of the high concentrations of colored dissolved organic matter and the presence of dust particles in seawater back scattering light."

Need to be corrected. The bias in the coastal regions is due to the presence of sediment: turbid water (case-2 water, higher concentration of inorganic particles). The "general bias" over the Mediterranean basin is due to the presence of colored dissolved organic matter in seawater and other components (not well known yet) that modify the optical properties of the seawater.

12) Line 247 – "the average chlorophyll surface concentration". That has been normalized?

13) Line 252 – "2 independent datasets" Replace "2" by "two". These datasets are not independent, the original data are from the same satellite sensors.

14) Line 267 – "200 m is $233 \pm 146 \cdot 10^{-9}$ g $L^{-1}$ (average over the 1991–2005 period), while the model value for the HIS/A2 simulation over the same period is $159 \pm 87 \cdot 10^{-9}$ g $L^{-1}$." The unit should be modified into: µg $L^{-1}$ or mg $m^{-3}$, and this sentence should be in the previous paragraph.

15) Maybe the paragraphs in the method (Section 2.2, lines 122 – 139) should be located into the Section 3.2.

16) Lines 290 – 295, already mentioned lines 210 – 219.

17) Line 301 – "Phosphate content in the entire Mediterranean has increased in our simulation by 6 % over the 21$^{st}$ century…"

Line 310 – "However, climate change effects lead to a global enhancement of 10 % in phosphate content in 2080–2099 in comparison to 1980–1999."

Is it 6 % or 10 %?

18) Line 384 – "P and N", you wrote phosphate and nitrate before, and now P and N. Please, stay consistent throughout the manuscript.

19) Line 417 – "Nutrient concentrations in the eastern part…" Replace nutrient by phosphate.

20) Line 422 – "a large annual variability" Replace by "a large interannual variability".

21) Line 427 – In the previous paragraph, about the western basin, some suggestions/interpretations were included. However, in this paragraph about the eastern basin, there were no suggestions to explain your results, why?

22) Line 459 – "In the Mediterranean Sea, primary productivity is mainly limited by these 2 nutrients and their evolution in the future may impact the productivity of the basin." Not necessary, the sentence can be removed.

23) Line 461 – "showing an accumulation of nitrate in large zones of the basin," add "by the end of the century".

24) Line 463 – "...and a small area in the southeastern Levantine" Also in the Gulf of Lion and south of Crete.

25) Line 465 – "...except near the mouth of the Nile and in the Alboran Sea." Also in the Ionian Sea, in the Tyrrhenian Sea, Algerian basin and between Crete and Cyprus.

26) Line 466 – "the N:P ratio (i.e. increase in P discharge and decrease in N discharge) in this river in our scenario." Replace by "the N:P ratio in this river in our scenario (i.e. increase in P discharge and decrease in N discharge).".

27) Line 471 – "All the most productive zones of the beginning of the century are reduced in size and intensity by the end of the century." This statement does not convince me... Too broad and unclear...

28) Line 481 – You already mentioned this area around Cyprus in the paragraph before.

29) Line 499 – "South Adriatic", North Adriatic?

30) Line 501 – "One specificity of Mediterranean biology is that most planktonic productivity occurs below the surface at a depth called the deep chlorophyll maximum (DCM). Hence, most of the chlorophyll concentration is not visible by satellites (Moutin et al., 2012)." I do not think that these sentences are necessary, and I do not think that this satellite limitation is discussed in Moutin et al., 2012.

31) Line 508 – "as the South Ionian and the Tyrrhenian basin." Not visible in these areas, mainly visible in the south of Crete.

32) Line 511-524 – statements in this paragraph are not supported by the results. For example,

- "but surface concentration is enhanced by about $25\ 10^{-9}$ g $L^{-1}$", that means 0.025 µg $L^{-1}$, which represents almost nothing.
- "This shows that local variability in the Mediterranean circulation and biogeochemistry is important." A general statement not supported by results.
- "the average chlorophyll concentration is reduced by almost 50 %", looking at figure 14a, there is not a 50 % decrease in the chlorophyll concentration.

33) Lines 527, 532, 533 – all percentage quoted in the text are different in Table 4. Please, double-check the values.

34) Line 546 and 556 – lack of quantitative estimates.

35) Section 4.1 Biogeochemical forcings – Not a good title. This section mainly discussed the influences of external sources of nutrients

36) Line 617 – "these regions" replace by "this region".

37) Line 624 – "2" replace by "two"

38) Line 701 – close parenthesis.

39) Figure A3a – It is impossible to read the colorbar.

---

## Referee Report (RR2)

This revised manuscript is improved from the previous version and the authors have taken on board the comments by the reviewers. There is now quantitative arguments throughout the manuscript and I accept what the authors say regarding the limitations of the model and that this should not stop the manuscript from being published as it is the best model and model inputs that they have available. They have clarified what is in the model and now acknowledge that organic P and N may also affect the budget.  In addition they have created a nice figure summarising the inputs and outputs of phosphate and nitrate to the basin.  However, I have still found this manuscript relatively difficult to read in places with the keys points lost within the text. There are now places with extremely long paragraphs (i.e lines 166-195, 331-362, 589-620, 696-731) and in these paragraphs it becomes unclear which key point the author wanted the reader to get from the it. I think both the results and discussion sections can still be improved and there should be increased emphasis on the impact of the results in the discussion.  I feel that the results the authors present are important and should be published but at present their impact does not come across strongly enough in the discussion.

**Lines 166-195:** There is now a very long paragraph in the method section discussing the riverine input. The authors have addressed my concerns in this paragraph but have additionally added further sentences which in my opinion become too detailed in regards to differences between the different inputs (Ludwig vs Adloff etc ) in different models which I find confusing.  In addition, there is now a lot of repetition between this section and the first paragraph of the discussion. Although the authors obviously do need to acknowledge the potential errors in their results I think these sections can be reduced.

**Section 3.3**: I still find this section hard to follow. The authors have now added quantitative metrics in this section but they are generally put in brackets rather than integrated into the text which is making it awkward to read.  In addition although  I appreciate that the authors have tried to change this section, the authors still explain trends such as decrease in phosphorus content by decrease in riverine inputs before presenting the river inputs and therefore are having to repeat things. I suggest putting the results on P and N content after discussing the other terms in the budget.

**Discussion:** The discussion feels dominated by statements about the limitations/uncertainties of the model with weak statements on the impact and interpretation of results. I think it would improve the manuscript if there was a better integration of the literature with the authors own arguments and conclusions from this study within the discussion.  In this revised manuscript the authors have added additional comparisons with other literature which is important and I think was needed, but it currently reads as a list **(Lines 696-731)**. The authors are trying to justify why their results are different than what is in the literature rather than using the literature to put their results into context and strengthen their arguments.  For example, in lines **727-731** rather than explaining why you can not observe the effects of temperature on nutrient recycling within this model, you can maybe say how an increase in nutrient cycling due to warmer temperatures may strengthen or weaken your conclusions.

Along these lines the discussion on N and P limitation (**Lines 713-726**) could be a paragraph/section to itself. In this section the authors state the their results are *"in contrast with previous literature on the matter"* (**Lines 715-716** ).  However there is evidence within the literature for P and N co-limitation in the Mediterranean.  Whilst generally the spring phytoplankton bloom is P limited, N and P co-limitation has been observed, especially during the stratification period (Thingstad et al., 2005; Tanaka et al., 2011) and there is some evidence of the spring phytoplankton bloom being N and P co-limited in the Western

Mediterranean (Pasqueron de Fommmervault et al., 2015) or even N limited (Marty et al. 2002). In addition N limitation has been predicted in the Alboran Gyre aswell (Ramirez et al., 2005; Lazzari et al., 2016). What time period do you calculate the N and P limitation for (i.e annual mean, spring bloom etc)? This may also affect what you are predicting compared to the literature

Finally, at other reviewers suggestion the authors have now included a scenario on atmospheric deposition but only present this within the discussion **(Lines 621-638)**. I feel it should be fully integrated into the text (i.e in the methods and results section) rather than tagged onto the discussion. It does provide some important insite on the effect of climate change despite only considering a climatology of atmospheric inputs rather than potential future ones. The authors could further hypothesise what potential future changes in atmospheric deposition may have on the results in the discussion based upon regional projections of atmospheric inputs into the future (i.e Lambarque et al., 2013). Whilst I appreciate they can't actually run a scenario they may be able to comment on whether it is likely to enhance/dampen the trend they see.

**Figure 18:** I suggest reversing the input and output arrows through the Strait of Gibraltar so that they are the same as the actual water flow. Currently it is suggesting an estuarine flow rather than anti-estuarine.

**References**

Lamarque, J. F., F. Dentener, J. McConnell, C. U. Ro, M. Shaw, R. Vet, D. Bergmann, P. Cameron-Smith, S. Dalsoren, R. Doherty, G. Faluvegi, S. J. Ghan, B. Josse, Y. H. Lee, I. A. MacKenzie, D. Plummer, D. T. Shindell, R. B. Skeie, D. S. Stevenson, S. Strode, G. Zeng, M. Curran, D. Dahl-Jensen, S. Das, D. Fritzsche, and M. Nolan (2013), Multi-model mean nitrogen and sulfur deposition from the Atmospheric Chemistry and Climate Model Intercomparison Project (ACCMIP): evaluation of historical and projected future changes, *Atmos Chem Phys*, *13*(16), 7997-8018, doi: 10.5194/acp-13-7997-2013

Lazzari, P., C. Solidoro, S. Salon, and G. Bolzon (2016), Spatial variability of phosphate and nitrate in the Mediterranean Sea: A modeling approach, *Deep Sea Res Part I Oceanogr Res Pap*, *108*, 39-52, doi: 10.1016/j.dsr.2015.12.006

Marty, J. C., J. Chiaverini, M. D. Pizay, and B. Avril (2002), Seasonal and interannual dynamics of nutrients and phytoplankton pigments in the western Mediterranean Sea at the DYFAMED time-series station (1991-1999), *Deep-Sea Res Pt II*, *49*(11), 1965-1985, doi: 10.1016/s0967-0645(02)00022-x

Pasqueron de Fommervault, O., C. Migon, F. D'Ortenzio, M. Ribera d'Alcalà, and L. Coppola (2015), Temporal variability of nutrient concentrations in the northwestern Mediterranean sea (DYFAMED time-series station), *Deep Sea Res Part I Oceanogr Res Pap*, *100*, 1-12, doi: http://dx.doi.org/10.1016/j.dsr.2015.02.006

Ramirez, T., D. Cortes, J. M. Mercado, M. Vargas-Yanez, M. Sebastian, and E. Liger (2005), Seasonal dynamics of inorganic nutrients and phytoplankton biomass in the NW Alboran Sea, *Estuarine Coastal Shelf Sci*, *65*(4), 654-670, doi: 10.1016/j.ecss.2005.07.012

Tanaka, T., T. F. Thingstad, U. Christaki, J. Colombet, V. Cornet-Barthaux, C. Courties, J. D. Grattepanche, A. Lagaria, J. Nedoma, L. Oriol, S. Psarra, M. Pujo-Pay, and F. Van Wambeke (2011), Lack of P-limitation of phytoplankton and heterotrophic prokaryotes in surface waters of three anticyclonic eddies in the stratified Mediterranean Sea, *Biogeosciences*, *8*(2), 525-538, doi: 10.5194/bg-8-525-2011

Thingstad, T. F., M. D. Krom, R. F. C. Mantoura, G. A. F. Flaten, S. Groom, B. Herut, N. Kress, C. S. Law, A. Pasternak, P. Pitta, S. Psarra, F. Rassoulzadegan, T. Tanaka, A. Tselepides, P. Wassmann, E. M. S. Woodward, C. W. Riser, G. Zodiatis, and T. Zohary (2005b), Nature of phosphorus limitation in the ultraoligotrophic eastern Mediterranean, *Science*, *309*(5737), 1068-1071, doi: 10.1126/science.1112632

---

## Editor Decision (ED1)

Authors sufficiently addressed all of my comments, and the quality of the manuscript is significantly improved. The contribution of this modeling study in the effort to understand the impacts of climate change on the Mediterranean Sea biogeochemistry is clearer than before. However, some points in the present version of the manuscript have to be restructured. This mainly concerned the uncertainties and limitations associated to this study, and one part of the results (see my major comments). In addition, several minor corrections are necessary (see my minor comments). Therefore, I believe that this paper will likely be a significant contribution and reach the quality standards for publication in the Biogeosciences journal after minor revisions of the manuscript.

Major comments:

1) Sections "4.2 Climate Change Scenario" and "4.3 Uncertainties from the PISCES model": these sub-sections need to be regrouped, completed, and rearranged at the end (or at the beginning) of the discussion section. In order to have a better insight into all uncertainties and limitations of the study (most of them already mentioned by the authors), to consider their influences on the results and processes discussed in the other sub-sections of the discussion (i.e., 4.1 and 4.4), and maybe to discuss some perspectives. For this, this sub-section should have a more general title, for example, "Limitations of the Study", and divided into three parts (at least):

   a. Climate change scenario: mostly section 4.2
   b. PISCES model: mostly section 4.3
   c. External sources of nutrients used (riverine and Atlantic inputs) and missing (atmospheric deposition): scattered in the manuscript

For example, I was expecting to have information in the discussion section about the influence of the following uncertainties and/or limitations on the biogeochemical response of the Mediterranean Sea,

- How the Atlantic condition and Atlantic nutrient concentrations used in the model could influence the biogeochemical results obtained, mainly for the Western Basin? For example, the authors wrote,

> Line 646 - "…the choice of atmospheric and Atlantic conditions has a strong influence on the MTHC."

with no information or discussion about possible consequences on the biogeochemical results obtained in their study.

- How the atmospheric deposition (not represented in the model, line 157 - "We did not include atmospheric deposition…") could influence the fact that the Mediterranean Sea is becoming more P-limited at the end of the century in your study?

2) In the conclusion – between lines 755 and 763 – authors highlighted the differences between the western and eastern basins in their biogeochemical responses to the climate change. This part of the results, which, I believe, is one of the major contribution of the manuscript, was absent in the discussion. There were even some contradictory statements, between the results, discussion and conclusion sections (see below), which make difficult to determine the author's opinion on some key aspects:

> Line 449 – "But the strong difference between CTRL_R and HIS/A2 at the end of the century indicates that vertical stratification leads to a decrease in surface layer nitrate concentrations, probably linked both with lower winter mixing and nutrient consumption by phytoplankton."

> Line 739 – "Stratification may lead to increased productivity in the surface because of the nutrient concentration increase (see also Macias et al., 2015)…"

> Line 751 – "increased stratification, and changes in Atlantic and river inputs, can lead to a significant accumulation of nitrate and a decrease in biological productivity in the surface…"

> Line 760 – "the eastern basin is more sensitive to vertical mixing and river inputs than the western basin […] and the stratification observed in the future leads to a reduction in surface productivity…"

Please, clarify your interpretation of the results in the discussion, in order to support all your statements in the conclusion.

Minor comments:

1) Line 68 – "Macias et al. (2015) simulated […]. They found that […] primary productivity over the eastern Mediterranean basin may increase as a result of density changes (increased stratification isolating the upper layer from the rest of the water column)."

Need to be corrected, because in the abstract of Macias et al. (2015),

> "In the eastern basin, on the contrary, all model runs simulate an increase in surface production linked to a density increase (less stratification) because of the increasing evaporation rate."

2) Line 81 – "This study aims at understanding the biogeochemical response of the Mediterranean to a "business–as–usual" climate change scenario throughout the 21st century."

The objective of the study should be more detailed. Please, highlight, for example, that the study mainly focus on the influence of the external sources of nutrients (rivers, Atlantic).

3) Lines 127 - 139 – "Temperature and salinity increase strongly, leading to a decrease in surface density and an overall increase in underline{vertical stratification}. Average sea surface temperature of the Mediterranean rises by up to 3°C by the end of the century. However, the temperature rise is not homogeneous in the basin, regions such as the Balearic, Aegean, Levantine and North Ionian undergo a more intense warming (over 3.4 °C) probably due to the addition of the atmosphere-originated quasi-homogeneous warming with the local effect of surface current changes. The salinity increases by 0.5 (practical salinity units) on average across the basin. In the A2 simulation, the entire Mediterranean basin is projected to become more stratified by 2100 and deep water formation is generally reduced. These variations in hydrological characteristics of the water masses generate important changes in the circulation and in particular in the vertical mixing intensity. The strong reduction in vertical mixing observed in all deep water formation areas of the basin is linked with the changes in salinity and temperature of the water masses."

This paragraph needs to be re-write because there are some repetitions, for example, about the stratification and vertical mixing.

4) Line 166 – "Nutrients input from rivers are derived from Ludwig et al. (2010) before 2000, Dissolved inorganic carbon (DIC) and Si are derived from Ludwig et al. (2009)."

What are included in "Nutrients"? Replace "Dissolved inorganic carbon" by "Dissolved Inorganic Carbon". "Si" means silicates? DIC and Si inputs are derived from Ludwig et al. 2009?

5) Line 190 – "This may result in higher nutrient concentrations at the river mouth. […] However, nutrients from river discharge are consumed rapidly at proximity of the river mouth and we believe these potential higher concentrations don't have a large impact of the results."

If all nutrients from river discharge are consumed near the river mouths, why did you study the influence of the river inputs at the scale of the Mediterranean basin?

6) Line 205 – "This period was chosen in order to avoid including in the CTRL years with too important warming such as the 1980s and 1990s." Need to be corrected.

7) Line 212 – "present–day conditions […] present–day conditions". Need to remove these expressions, or to define them.

8) Line 235 – "from satellite estimations from MyOcean Dataset ([http://marine.copernicus.eu)](http://marine.copernicus.eu)". Which product of MyOcean, quote the full link to access this dataset.

9) Line 236 – "All chlorophyll values in the article and the data are chlorophyll–a." I do not understand. Do you mean that the word "chlorophyll" stands for "chlorophyll-a" throughout the manuscript? If so, it needs to be define when you use the word chlorophyll for the first time.

10) Line 240 – "with values that", are you talking about the difference in magnitude?

11) Line 243 – "Moreover, several studies (see e.g. Claustre et al., 2002; Morel and Gentili, 2009) show that satellite estimates have a systematic positive bias in the coastal regions because of the high concentrations of colored dissolved organic matter and the presence of dust particles in seawater back scattering light."

Need to be corrected. The bias in the coastal regions is due to the presence of sediment: turbid water (case-2 water, higher concentration of inorganic particles). The "general bias" over the Mediterranean basin is due to the presence of colored dissolved organic matter in seawater and other components (not well known yet) that modify the optical properties of the seawater.

12) Line 247 – "the average chlorophyll surface concentration". That has been normalized?

13) Line 252 – "2 independent datasets" Replace "2" by "two". These datasets are not independent, the original data are from the same satellite sensors.

14) Line 267 – "200 m is $233 \pm 146 \ 10^{-9}$ g $L^{-1}$ (average over the 1991–2005 period), while the model value for the HIS/A2 simulation over the same period is $159 \pm 87 \ 10^{-9}$ g $L^{-1}$." The unit should be modified into: $\mu g \ L^{-1}$ or mg $m^{-3}$, and this sentence should be in the previous paragraph.

15) Maybe the paragraphs in the method (Section 2.2, lines 122 – 139) should be located into the Section 3.2.

 16) Lines 290 – 295, already mentioned lines 210 – 219.

17) Line 301 – "Phosphate content in the entire Mediterranean has increased in our simulation by 6 % over the 21$^{st}$ century…"

Line 310 – "However, climate change effects lead to a global enhancement of 10 % in phosphate content in 2080–2099 in comparison to 1980–1999."

Is it 6 % or 10 %?

18) Line 384 – "P and N", you wrote phosphate and nitrate before, and now P and N. Please, stay consistent throughout the manuscript.

19) Line 417 – "Nutrient concentrations in the eastern part…" Replace nutrient by phosphate.

20) Line 422 – "a large annual variability" Replace by "a large interannual variability".

21) Line 427 – In the previous paragraph, about the western basin, some suggestions/interpretations were included. However, in this paragraph about the eastern basin, there were no suggestions to explain your results, why?

22) Line 459 – "In the Mediterranean Sea, primary productivity is mainly limited by these 2 nutrients and their evolution in the future may impact the productivity of the basin." Not necessary, the sentence can be removed.

23) Line 461 – "showing an accumulation of nitrate in large zones of the basin," add "by the end of the century".

24) Line 463 – "…and a small area in the southeastern Levantine" Also in the Gulf of Lion and south of Crete.

25) Line 465 – "…except near the mouth of the Nile and in the Alboran Sea." Also in the Ionian Sea, in the Tyrrhenian Sea, Algerian basin and between Crete and Cyprus.

26) Line 466 – "the N:P ratio (i.e. increase in P discharge and decrease in N discharge) in this river in our scenario." Replace by "the N:P ratio in this river in our scenario (i.e. increase in P discharge and decrease in N discharge).".

27) Line 471 – "All the most productive zones of the beginning of the century are reduced in size and intensity by the end of the century." This statement does not convince me... Too broad and unclear...

28) Line 481 – You already mentioned this area around Cyprus in the paragraph before.

29) Line 499 – "South Adriatic", North Adriatic?

30) Line 501 – "One specificity of Mediterranean biology is that most planktonic productivity occurs below the surface at a depth called the deep chlorophyll maximum (DCM). Hence, most of the chlorophyll concentration is not visible by satellites (Moutin et al., 2012)." I do not think that these sentences are necessary, and I do not think that this satellite limitation is discussed in Moutin et al., 2012.

31) Line 508 – "as the South Ionian and the Tyrrhenian basin." Not visible in these areas, mainly visible in the south of Crete.

32) Line 511-524 – statements in this paragraph are not supported by the results. For example,

- "but surface concentration is enhanced by about 25 $10^{-9}$ g $L^{-1}$", that means 0.025 µg $L^{-1}$, which represents almost nothing.
- "This shows that local variability in the Mediterranean circulation and biogeochemistry is important." A general statement not supported by results.
- "the average chlorophyll concentration is reduced by almost 50 %", looking at figure 14a, there is not a 50 % decrease in the chlorophyll concentration.

33) Lines 527, 532, 533 – all percentage quoted in the text are different in Table 4. Please, double-check the values.

34) Line 546 and 556 – lack of quantitative estimates.

35) Section 4.1 Biogeochemical forcings – Not a good title. This section mainly discussed the influences of external sources of nutrients

36) Line 617 – "these regions" replace by "this region".

37) Line 624 – "2" replace by "two"

38) Line 701 – close parenthesis.

39) Figure A3a – It is impossible to read the colorbar.

This revised manuscript is improved from the previous version and the authors have taken on board the comments by the reviewers. There is now quantitative arguments throughout the manuscript and I accept what the authors say regarding the limitations of the model and that this should not stop the manuscript from being published as it is the best model and model inputs that they have available. They have clarified what is in the model and now acknowledge that organic P and N may also affect the budget. In addition they have created a nice figure summarising the inputs and outputs of phosphate and nitrate to the basin. However, I have still found this manuscript relatively difficult to read in places with the keys points lost within the text. There are now places with extremely long paragraphs (i.e lines 166-195, 331-362, 589-620, 696-731) and in these paragraphs it becomes unclear which key point the author wanted the reader to get from the it. I think both the results and discussion sections can still be improved and there should be increased emphasis on the impact of the results in the discussion. I feel that the results the authors present are important and should be published but at present their impact does not come across strongly enough in the discussion.

**Lines 166-195:** There is now a very long paragraph in the method section discussing the riverine input. The authors have addressed my concerns in this paragraph but have additionally added further sentences which in my opinion become too detailed in regards to differences between the different inputs (Ludwig vs Adloff etc ) in different models which I find confusing. In addition, there is now a lot of repetition between this section and the first paragraph of the discussion. Although the authors obviously do need to acknowledge the potential errors in their results I think these sections can be reduced.

**Section 3.3**: I still find this section hard to follow. The authors have now added quantitative metrics in this section but they are generally put in brackets rather than integrated into the text which is making it awkward to read. In addition although I appreciate that the authors have tried to change this section, the authors still explain trends such as decrease in phosphorus content by decrease in riverine inputs before presenting the river inputs and therefore are having to repeat things. I suggest putting the results on P and N content after discussing the other terms in the budget.

**Discussion:** The discussion feels dominated by statements about the limitations/uncertainties of the model with weak statements on the impact and interpretation of results. I think it would improve the manuscript if there was a better integration of the literature with the authors own arguments and conclusions from this study within the discussion. In this revised manuscript the authors have added additional comparisons with other literature which is important and I think was needed, but it currently reads as a list **(Lines 696-731)**. The authors are trying to justify why their results are different than what is in the literature rather than using the literature to put their results into context and strengthen their arguments. For example, in lines **727-731** rather than explaining why you can not observe the effects of temperature on nutrient recycling within this model, you can maybe say how an increase in nutrient cycling due to warmer temperatures may strengthen or weaken your conclusions.

Along these lines the discussion on N and P limitation (**Lines 713-726**) could be a paragraph/section to itself. In this section the authors state the their results are *"in contrast with previous literature on the matter"* (**Lines 715-716** ). However there is evidence within the literature for P and N co-limitation in the Mediterranean. Whilst generally the spring phytoplankton bloom is P limited, N and P co-limitation has been observed, especially during the stratification period (Thingstad et al., 2005; Tanaka et al., 2011) and there is some evidence of the spring phytoplankton bloom being N and P co-limited in the Western

Mediterranean (Pasqueron de Fommmervault et al., 2015) or even N limited (Marty et al. 2002). In addition N limitation has been predicted in the Alboran Gyre aswell (Ramirez et al., 2005; Lazzari et al., 2016). What time period do you calculate the N and P limitation for (i.e annual mean, spring bloom etc)? This may also affect what you are predicting compared to the literature

Finally, at other reviewers suggestion the authors have now included a scenario on atmospheric deposition but only present this within the discussion **(Lines 621-638)**. I feel it should be fully integrated into the text (i.e in the methods and results section) rather than tagged onto the discussion. It does provide some important insite on the effect of climate change despite only considering a climatology of atmospheric inputs rather than potential future ones. The authors could further hypothesise what potential future changes in atmospheric deposition may have on the results in the discussion based upon regional projections of atmospheric inputs into the future (i.e Lambarque et al., 2013). Whilst I appreciate they can't actually run a scenario they may be able to comment on whether it is likely to enhance/dampen the trend they see.

**Figure 18:** I suggest reversing the input and output arrows through the Strait of Gibraltar so that they are the same as the actual water flow. Currently it is suggesting an estuarine flow rather than anti-estuarine.

**References**

Lamarque, J. F., F. Dentener, J. McConnell, C. U. Ro, M. Shaw, R. Vet, D. Bergmann, P. Cameron-Smith, S. Dalsoren, R. Doherty, G. Faluvegi, S. J. Ghan, B. Josse, Y. H. Lee, I. A. MacKenzie, D. Plummer, D. T. Shindell, R. B. Skeie, D. S. Stevenson, S. Strode, G. Zeng, M. Curran, D. Dahl-Jensen, S. Das, D. Fritzsche, and M. Nolan (2013), Multi-model mean nitrogen and sulfur deposition from the Atmospheric Chemistry and Climate Model Intercomparison Project (ACCMIP): evaluation of historical and projected future changes, *Atmos Chem Phys*, *13*(16), 7997-8018, doi: 10.5194/acp-13-7997-2013

Lazzari, P., C. Solidoro, S. Salon, and G. Bolzon (2016), Spatial variability of phosphate and nitrate in the Mediterranean Sea: A modeling approach, *Deep Sea Res Part I Oceanogr Res Pap*, *108*, 39-52, doi: 10.1016/j.dsr.2015.12.006

Marty, J. C., J. Chiaverini, M. D. Pizay, and B. Avril (2002), Seasonal and interannual dynamics of nutrients and phytoplankton pigments in the western Mediterranean Sea at the DYFAMED time-series station (1991-1999), *Deep-Sea Res Pt II*, *49*(11), 1965-1985, doi: 10.1016/s0967-0645(02)00022-x

Pasqueron de Fommervault, O., C. Migon, F. D'Ortenzio, M. Ribera d'Alcalà, and L. Coppola (2015), Temporal variability of nutrient concentrations in the northwestern Mediterranean sea (DYFAMED time-series station), *Deep Sea Res Part I Oceanogr Res Pap*, *100*, 1-12, doi: http://dx.doi.org/10.1016/j.dsr.2015.02.006

Ramirez, T., D. Cortes, J. M. Mercado, M. Vargas-Yanez, M. Sebastian, and E. Liger (2005), Seasonal dynamics of inorganic nutrients and phytoplankton biomass in the NW Alboran Sea, *Estuarine Coastal Shelf Sci*, *65*(4), 654-670, doi: 10.1016/j.ecss.2005.07.012

Tanaka, T., T. F. Thingstad, U. Christaki, J. Colombet, V. Cornet-Barthaux, C. Courties, J. D. Grattepanche, A. Lagaria, J. Nedoma, L. Oriol, S. Psarra, M. Pujo-Pay, and F. Van Wambeke (2011), Lack of P-limitation of phytoplankton and heterotrophic prokaryotes in surface waters of three anticyclonic eddies in the stratified Mediterranean Sea, *Biogeosciences*, *8*(2), 525-538, doi: 10.5194/bg-8-525-2011

Thingstad, T. F., M. D. Krom, R. F. C. Mantoura, G. A. F. Flaten, S. Groom, B. Herut, N. Kress, C. S. Law, A. Pasternak, P. Pitta, S. Psarra, F. Rassoulzadegan, T. Tanaka, A. Tselepides, P. Wassmann, E. M. S. Woodward, C. W. Riser, G. Zodiatis, and T. Zohary (2005b), Nature of phosphorus limitation in the ultraoligotrophic eastern Mediterranean, *Science*, *309*(5737), 1068-1071, doi: 10.1126/science.1112632

---

## Author Response (AR2)

Dear Editor,

Please find below the authors' responses to the reviewers' comments.

We have improved the manuscript by incorporating changes suggested by the reviewers. In particular, we have reorganized the results section and tried to express more clearly our findings in section 3.3.5. We also have reorganized the discussion by emphasizing all limitations of the study into section 4.1, and summarizing and discussing the impacts of our results in section 4.2.

As suggested by reviewer 2, we incorporated the results concerning atmospheric deposition effects in the methods and results section.

The authors would like to emphasize that the English has been corrected throughout the manuscript in an effort to make it easier to read.

You can find a track version of the manuscript highlighting all changes attached to these answers.

Best regards,

The authors

*Authors sufficiently addressed all of my comments, and the quality of the manuscript is significantly improved. The contribution of this modeling study in the effort to understand the impacts of climate change on the Mediterranean Sea biogeochemistry is clearer than before. However, some points in the present version of the manuscript have to be restructured. This mainly concerned the uncertainties and limitations associated to this study, and one part of the results (see my major comments). In addition, several minor corrections are necessary (see my minor comments). Therefore, I believe that this paper will likely be a significant contribution and reach the quality standards for publication in the Biogeosciences journal after minor revisions of the manuscript.*

We thank the reviewer for their comments, which we have answered fully, while reorganizing and clarifying the manuscript accordingly.

*Major comments:*

*1) Sections "4.2 Climate Change Scenario" and "4.3 Uncertainties from the PISCES model": these sub-sections need to be regrouped, completed, and rearranged at the end (or at the beginning) of the discussion section. In order to have a better insight into all uncertainties and limitations of the study (most of them already mentioned by the authors), to consider their influences on the results and processes discussed in the other sub-sections of the discussion (i.e., 4.1 and 4.4), and maybe to discuss some perspectives. For this, this sub-section should have a more general title, for example, "Limitations of the Study", and divided into three parts (at least):*

*a. Climate change scenario: mostly section 4.2*

*b. PISCES model: mostly section 4.3*

*c. External sources of nutrients used (riverine and Atlantic inputs) and missing (atmospheric deposition): scattered in the manuscript*

*For example, I was expecting to have information in the discussion section about the influence of the following uncertainties and/or limitations on the biogeochemical response of the Mediterranean Sea,*

*- How the Atlantic condition and Atlantic nutrient concentrations used in the model could influence the biogeochemical results obtained, mainly for the Western Basin? For example, the authors wrote,*

*Line 646 - "…the choice of atmospheric and Atlantic conditions has a strong influence on the MTHC."*

*with no information or discussion about possible consequences on the biogeochemical results obtained in their study.*

*- How the atmospheric deposition (not represented in the model, line 157 - "We did not include atmospheric deposition…") could influence the fact that the Mediterranean Sea is becoming more P-limited at the end of the century in your study?*

We have rearranged and modified the Discussion to address these comments. We regrouped sections 4.2 and 4.3 in one section named "Sources of uncertainties" (lines 585 to 685). This section is divided into 3 subsections covering:

- 1: Uncertainties linked with the climate change scenario (lines 593 to 623)

In this subsection, we discuss the potential impacts of different climate change scenarios on the Mediterranean Sea. Our HIS/A2 simulation increased stratification in Mediterranean leads to lower surface nutrient concentrations and hence lower primary productivity (Figures 7a,d and 8a,d). Conversely, Macias et al (2015) found that the RCP4.5 and RCP8.5 scenarios reduced the simulated vertical stratification , which led to slightly increased surface primary productivity. This large sensitivity to the choice of climate change scenario, calls for the use of a greater number of scenarios when modeling related changes in the Mediterranean Sea.

See lines 599 to 611: *"Overall, the increase in stratification in our A2 climate change scenario leads to different evolutions of nutrient concentrations between the surface and the intermediate and deep waters with surface waters becoming more sensitive to external nutrient sources (Figures 7 and 8). On the other hand, Macias et al. (2015) found that primary productivity slightly increased as a result of decreased stratification in the climate change scenarios RCP 8.5 and RCP 4.5. The A2 scenario that we used was the only one available with 3–D daily forcings, as necessary for coupling with the PISCES biogeochemical model. However, Adloff et al. (2015) showed that other SRES scenarios such as the A1B or B1 may lead to a future decline in the vertical stratification with probably different consequences on the Mediterranean Sea biogeochemistry. Our study is thus only a first step for transient modeling of the Mediterranean Sea biogeochemistry. It should be complemented by new simulations that explore the various sources of uncertainty (model choice, internal variability, scenario choice) once appropriate forcings become available for multiple models as expected from the Med–CORDEX initiative (Ruti et al., 2016)."*

- 2: Uncertainties from the PISCES model (lines 625 to 646)

- 3: Uncertainties linked with external nutrient sources (lines 648 to 685).

In this subsection, we discuss the importance of accurately representing the external nutrient sources in future Mediterranean biogeochemical simulations. In an effort to facilitate reading, we divided this section in 3 paragraphs:

Fluxes through the Strait of GIbraltar

The Atlantic conditions and nutrient concentrations were shown with the CTRL_RG simulation to have a strong influence on the western Mediterranean biogeochemistry. Hence it is important to have realistic scenarios for the evolution of the Atlantic boundary conditions to simulate the Mediterranean. In our

case, the Atlantic conditions are simulated with the same model (NEMO/PISCES) and the same climate change scenario (A2). Therefore, they are entirely compatible with our scenario.

Lines 675 to 680: "*Results from our CTRL_RG simulation show that the increase in nitrate and phosphate incoming fluxes through the Strait of Gibraltar leads to higher surface concentrations in the western Mediterranean. The Atlantic nutrient concentrations are derived from a global version of the same model used our simulations (NEMO/PISCES) and forced under the same A2 climate change scenario). Therefore, there is no incompatibility issue between for the forcing and model.*"

Riverine nutrient fluxes

The river fluxes of freshwater and nutrients are derived from different models and there are uncertainties with the nutrient flows we represent. However, this is the only scenario for river inputs available.

We modified the text: "*For the riverine nutrient inputs, scenarios from the MEA report are based on different assumptions from the IPCC SRES scenarios used to compute freshwater runoff in the HIS/A2 simulation. Freshwater discharge from Ludwig et al. (2010) is based on the SESAME model reconstruction and differs
from freshwater runoff in the ARPEGE–Climate model used to force our physical model. This may lead to incoherences between water and nutrient discharges, but the nutrient discharges from Ludwig et al. (2010) are the only ones that are available. Furthermore, the SESAME model is not coupled with NEMO/PISCES. Associated discrepancies and the uncertainties linked with the use of inconsistent scenarios in our simulation should be addressed by developing a more integrated modelling framework to study the impacts of climate change on the Mediterranean Sea biogeochemistry.*" Lines 658 to 674.

Potential effects of aerosol deposition

There is no scenario for the evolution of aerosol deposition during the 21[st] century. It is difficult to forecast the evolution of aerosol emissions and deposition as it is influenced by socio-economic decisions, land-use change, winds and rains. Measurements of atmospheric deposition remain sparse and it is difficult to interpret any tendency from them. In this context of very sparse data and modeling, it is difficult to derive robust scenarios for the evolution of aerosol deposition over the Mediterranean.

However, as we noticed important effects of external nutrient sources on the Mediterranean surface biogeochemistry, atmospheric deposition may also influence on the response of the Mediterranean biogeochemistry to future changes. Our results using present day phosphate deposition from natural dust (see figure 17) show that atmospheric deposition of phosphate enhances surface primary production in the surface Mediterranean. Thus enhanced phosphate fluxes from aerosols may limit the surface decrease of phosphate concentrations and limit phosphorus limitation. However, in the HIS/A2_NALADIN simulation, the surface Mediterranean is still P-limited over most of the Mediterranean because the atmospheric nutrient fluxes are low in comparison to the fluxes from rivers and across the Strait of Gibraltar. Therefore, it appears unlikely that changes in aerosol deposition from natural dust supply will have a larger impact on the Mediterranean biogeochemistry in the future. However, there are multiple sources of aerosols that are not included in atmospheric models such as anthropogenic, volcanic and volatile organic compounds. They may constitute important nutrient fluxes for surface waters of the

Mediterranean Sea. Moreover, aerosols affect radiative forcing over the Mediterranean and hence also climate conditions. It seems therefore important to try to represent this nutrient source in the Mediterranean models.

We added in the text (lines 711 to 723): *"Results from these simulations show that enhanced phosphate fluxes from aerosols may limit the surface decrease of phosphate concentrations and limit phosphorus limitation. However, in the HIS/A2_NALADIN simulation, the surface Mediterranean is still P-limited in most of the Mediterranean because the atmospheric nutrient fluxes are low in comparison to riverine nutrient fluxes from rivers and the nutrient flux through the Strait of Gibraltar (see Richon et al., 2017). Therefore, it appears unlikely that changes in aerosol deposition from natural dust would greatly influence future Mediterranean biogeochemistry. However, there are multiple sources of aerosols that are not included in atmospheric models, e.g., anthropogenic, volcanic and volatile organic compounds (e.g. Wang et al., 2014; Kanakidou et al., 2016). Their combined influence could perhaps constitute an important nutrient flux to the Mediterranean, thus altering the evolution of its biogeochemistry. Moreover, aerosols affect radiative forcing over the Mediterranean and may impact the climate conditions (Nabat et al., 2015b). Thus, efforts should be made to accurately represent this nutrient source in Mediterranean models to assess the effect on Mediterranean Sea biogeochemistry with regards to climate change."*

*2) In the conclusion – between lines 755 and 763 – authors highlighted the differences between the western and eastern basins in their biogeochemical responses to the climate change. This part of the results, which, I believe, is one of the major contribution of the manuscript, was absent in the discussion. There were even some contradictory statements, between the results, discussion and conclusion sections (see below), which make difficult to determine the author's opinion on some key aspects:*
*Line 449 – "But the strong difference between CTRL_R and HIS/A2 at the end of the century indicates that vertical stratification leads to a decrease in surface layer nitrate concentrations, probably linked both with lower winter mixing and nutrient consumption by phytoplankton."*
*Line 739 – "Stratification may lead to increased productivity in the surface because of the nutrient concentration increase (see also Macias et al., 2015)…"*
*Line 751 – "increased stratification, and changes in Atlantic and river inputs, can lead to a significant accumulation of nitrate and a decrease in biological productivity in the surface…"*
*Line 760 – "the eastern basin is more sensitive to vertical mixing and river inputs than the western basin […] and the stratification observed in the future leads to a reduction in surface productivity…"*
*Please, clarify your interpretation of the results in the discussion, in order to support all your statements in the conclusion.*

Our HIS/A2 simulation is the first to consider the joint evolution of physical aspects and external nutrient sources. In some regions, there are antagonistic effects of stratification and nutrient discharge. For instance, in the eastern basin, the increase in nitrate discharge from rivers tends to increase the surface nitrate concentration (see the evolution of nitrate concentrations in CTRL_R). However, changes in physical conditions seem to lower surface nitrate concentrations in the eastern basin by the end of the 21$^{st}$ century (see concentrations in HIS/A2).
See lines 431-440: "*In the eastern basin, the impacts of river discharges of nitrate seem to have large influence on the nitrate accumulation as shown by the similar evolution of HIS/A2 and CTRL_R simulations (Figures 8d, 8e and 8f). Figure 8d shows the contrasted effects of climate and biogeochemical changes. The strong difference between CTRL_R and CTRL concentrations at the beginning of the*

*simulation (almost 0.4 mmol $m^{-3}$) indicates that riverine nutrient discharge has a strong influence on surface nitrate concentrations in the eastern basin and is responsible for an important part of the eastern Mediterranean nitrate budget (see also Table 3). But the strong difference between CTRL_R and HIS/A2 at the end of the century indicates that vertical stratification leads to a decrease in surface layer nitrate concentrations, probably linked both with lower winter mixing and nutrient consumption by phytoplankton."*

Our results suggest that the western Mediterranean basin is more sensitive to changes in nutrient fluxes through the Strait of Gibraltar and to the physical changes linked with climate change. The eastern basin is generally less sensitive to the fluxes through the Strait of Gibraltar, but some regions such as the Adriatic Sea and the Levantine basin are more sensitive to evolution of nutrient fluxes from rivers and to the physical changes.

See lines 742 to 748: *"Results from our different control simulations indicates the extent to which the choice of the biogeochemical forcing scenario may influence the future evolution of the Mediterranean Sea biogeochemistry. Considering only our climate change scenario, less vertical mixing causes a basin-wide decline in surface nutrient concentrations, causing a reduction in primary productivity. But the combined effects of nutrient fluxes from external sources and climate change that lead to the surface accumulation in nitrate (figures 8a and 8d and Table 3). In particular, nutrient inputs through the Strait of Gibraltar have substantial influence on the western basin."*

Our results show 2 main conclusions regarding the effects of climate change versus biogeochemical forcings: 1) surface biogeochemistry is influenced by the evolution of both external nutrient fluxes and climate change, and climate change effects on biogeochemistry are more visible in the intermediate and deep waters; 2) nutrient fluxes through the Strait of Gibraltar primarily influence the western basin whereas riverine nutrient input primarily influences the eastern basin.

We tried to emphasize these observations in the result section by adding a paragraph (lines 447 to 451): *"Evaluating separately the evolution of nutrient concentrations in different layer of the Mediterranean Sea shows that external nutrient fluxes primarily affect the surface in the western basin whereas climate change affects the entire water column. Also, climate and nutrient fluxes may have opposite effects on surface nutrient concentration. This leads to different trends in nutrient concentrations in the surface layer and in the intermediate and deep layers. In particular, surface nitrate in the eastern basin is observed to increase as a result of increased river discharge, but climate change effects lower concentrations in HIS/A2 (see figure 8d). On the other hand, climate and river discharge of nitrate have similar effects on the intermediate and deep eastern layers, leading to the simulated increase in nutrient content (Tables 2 and 3)."*

See also lines 738-741:*" The difference between HIS/A2 and CTRL_RG phosphate concentrations in the intermediate and deep layers (Figure 7b, 7e, 7c, 7f) indicates that variations of phosphate concentrations during the 21st century are primarily driven by climate change while nitrate concentration is equally sensitive to changes in biogeochemical forcings."*

Also, in an effort to make to results section 3.3.5 easier to read, we clearly separated the results concerning phosphate and nitrate in 2 different paragraphs. In each of these paragraphs, we first

describe the results in the western basin (with the order: surface, intermediate and deep waters) and then in the eastern basin.

*Minor comments:*
*1) Line 68 – "Macias et al. (2015) simulated [...]. They found that [...] primary productivity over the eastern Mediterranean basin may increase as a result of density changes (increased stratification isolating the upper layer from the rest of the water column)."*
*Need to be corrected, because in the abstract of Macias et al. (2015),*
*"In the eastern basin, on the contrary, all model runs simulate an increase in surface production linked to a density increase (less stratification) because of the increasing evaporation rate."*
We thank the reviewer for pointing out this mistake, we meant "decreased stratification". The wording in that sentence has been changed accordingly.

*2) Line 81 – "This study aims at understanding the biogeochemical response of the Mediterranean to a "business–as–usual" climate change scenario throughout the 21st century."*
*The objective of the study should be more detailed. Please, highlight, for example, that the study mainly focus on the influence of the external sources of nutrients (rivers, Atlantic).*

We changed this sentence to "This study aims at understanding the biogeochemical response of the Mediterranean to a "business-as-usual" climate change scenario throughout the 21st century**, separating the effects of climate, nutrient inputs from rivers and changes in nutrient fluxes through the Strait of Gibraltar**."

*3) Lines 127 - 139 – "Temperature and salinity increase strongly, leading to a decrease in surface density and an overall increase in vertical stratification. Average sea surface temperature of the Mediterranean rises by up to 3°C by the end of the century. However, the temperature rise is not homogeneous in the basin, regions such as the Balearic, Aegean, Levantine and North Ionian undergo a more intense warming (over 3.4 °C) probably due to the addition of the atmosphere-originated quasi-homogeneous warming with the local effect of surface current changes. The salinity increases by 0.5 (practical salinity units) on average across the basin. In the A2 simulation, the entire Mediterranean basin is projected to become more stratified by 2100 and deep water formation is generally reduced. These variations in hydrological characteristics of the water masses generate important changes in the circulation and in particular in the vertical mixing intensity. The strong reduction in vertical mixing observed in all deep water formation areas of the basin is linked with the changes in salinity and temperature of the water masses."*
*This paragraph needs to be re-write because there are some repetitions, for example, about the stratification and vertical mixing.*
We separated this paragraph and reformulated it:
"*Average sea surface temperature of the Mediterranean rises by up to 3°C by the end of the century. However, that warming is not homogeneous across the basin, with regions such as the Balearic, Aegean, Levantine and North Ionian undergoing greater warming (>3.4 °C) probably due to the addition of the atmosphere-originated quasi-homogeneous warming being combined with local changes in surface currents. The salinity increases by 0.5 (practical salinity units) on average across the basin. These changes in hydrological characteristics generate substantial changes in the circulation and in particular the*

*vertical mixing intensity. Under the A2 scenario, the Mediterranean basin is projected to become more stratified by 2100. Consequently, deep-water formation is generally reduced."*

**4) Line 166 – "Nutrients input from rivers are derived from Ludwig et al. (2010) before 2000, Dissolved inorganic carbon (DIC) and Si are derived from Ludwig et al. (2009)."**
**What are included in "Nutrients"? Replace "Dissolved inorganic carbon" by "Dissolved Inorganic Carbon". "Si" means silicates? DIC and Si inputs are derived from Ludwig et al. 2009?**
The nutrients that are mentioned include nitrate ($NO_3$), phosphate ($PO_4$), and dissolved organic carbon (DOC). We added these precisions.
*"Nutrient inputs from rivers, including $NO_3^-$, $PO4_3^-$ (hereafter noted $NO_3$ and $PO_4$), and dissolved organic carbon (DOC) are derived from Ludwig et al. (2010). Dissolved inorganic carbon (DIC) and Si are derived from Ludwig et al. (2009)."*

**5) Line 190 – "This may result in higher nutrient concentrations at the river mouth. […] However, nutrients from river discharge are consumed rapidly at proximity of the river mouth and we believe these potential higher concentrations don't have a large impact of the results."**
**If all nutrients from river discharge are consumed near the river mouths, why did you study the influence of the river inputs at the scale of the Mediterranean basin?**
At the time that the above-mentioned sentence was written, we meant that the effects of a particular river would be locally confined and would not substantially alter the mean state of the entire basin. However, our results do show that river inputs do have important effects on the regional scale. Hence we have now deleted this confusing sentence. For further clarification we also rearranged other sentences of the same paragraph.

**6) Line 205 – "This period was chosen in order to avoid including in the CTRL years with too important warming such as the 1980s and 1990s." Need to be corrected.**
Warming trends were already observed in the Mediterranean in the 1980s and 1990s (Adloff et al., 2015, HIS period), we chose to use a CTRL basis that does not include this trend.
Figure 12 from Adloff et al (2015), reproduced below, shows the simulated Mediterranean average SST between 1961 and 2100. It illustrates that in the HIS simulation (black line), i.e., the physical forcing we used in this article, the Mediterranean surface waters have a warming trend from 1980 onward. In the revised manuscript, we mention the same figure in this context.

[Figure]

**Fig. 12** Yearly mean time series of sea surface temperature anomalies (vs. 1961–1990) averaged over the Mediterranean basin. A2-F, A2-RF and A2-ARF curves are overlapping. The spread of the ensemble is *shaded* in *grey*

**7) Line 212 – "present–day conditions […] present–day conditions". Need to remove these expressions, or to define them.**
We changed the sentence to "*In order to separately quantify the effects of climate and biogeochemical changes, we made 2 additional control simulations: (1) CTRL_R with climatic and Atlantic conditions corresponding to present–day conditions (no scenario for climate change or nutrient fluxes through the Strait of Gibraltar) and river nutrient discharge following the scenario evolution, and (2) CTRL_RG with present-day climatic conditions, but river nutrient discharge and Atlantic buffer–zone concentrations following the scenario conditions.*"

**8) Line 235 – "from satellite estimations from MyOcean Dataset (http://marine.copernicus.eu)". Which product of MyOcean, quote the full link to access this dataset.**
Done

**9) Line 236 – "All chlorophyll values in the article and the data are chlorophyll–a." I do not understand. Do you mean that the word "chlorophyll" stands for "chlorophyll-a" throughout the manuscript? If so, it needs to be define when you use the word chlorophyll for the first time.**

This is the first time we use chlorophyll in the article, we define here that we only use chlorophyll-a data. "*Whenever we refer to chlorophyll, we always mean chlorophyll–a (hereafter noted chl-a).*"

**10) Line 240 – "with values that", are you talking about the difference in magnitude?**
In the revised manuscript, we have now clarified that sentence, saying "*approximately 50% decrease in average chlorophyll concentration between the western and the eastern basin in the satellite data and 30 to 50% in the model.*"

**11) Line 243 – "Moreover, several studies (see e.g. Claustre et al., 2002; Morel and Gentili, 2009) show that satellite estimates have a systematic positive bias in the coastal regions because of the high concentrations of colored dissolved organic matter and the presence of dust particles in seawater back scattering light."**
**Need to be corrected. The bias in the coastal regions is due to the presence of sediment: turbid water (case-2 water, higher concentration of inorganic particles). The "general bias" over the Mediterranean**

*basin is due to the presence of colored dissolved organic matter in seawater and other components (not well known yet) that modify the optical properties of the seawater.*

Corrected to "*because of the presence of particulate matter (for instance, sediments). The general bias observed in the Mediterranean is linked with colored dissolved organic matter and the presence of dust particles in seawater, which cause light back scattering.*"

*12) Line 247 – "the average chlorophyll surface concentration". That has been normalized?*

Yes, we added precision: "*Figure 2 provides an evaluation of the average normalized chl-a surface concentration evolution over the entire basin for the period 1997–2005.*"

*13) Line 252 – "2 independent datasets" Replace "2" by "two". These datasets are not independent, the original data are from the same satellite sensors.*

Reviewer is right, in the revised manuscript, we now write "two different datasets".

*14) Line 267 – "200 m is 233 ± 146 10$_{-9}$ g L$_{-1}$ (average over the 1991–2005 period), while the model value for the HIS/A2 simulation over the same period is 159 ± 87 10$_{-9}$ g L$_{-1}$." The unit should be modified into: µg L$_{-1}$ or mg m$_{-3}$, and this sentence should be in the previous paragraph.*

We changed all chlorophyll units to ng L$^{-1}$ and put the sentence in the previous paragraph.

*15) Maybe the paragraphs in the method (Section 2.2, lines 122 – 139) should be located into the Section 3.2.*

We placed this paragraph in the Methods because these are not the results from our study but from Adloff et al (2015).

*16) Lines 290 – 295, already mentioned lines 210 – 219.*

These sentences have now been removed.

*17) Line 301 – "Phosphate content in the entire Mediterranean has increased in our simulation by 6 % over the 21$_{st}$ century…"*
*Line 310 – "However, climate change effects lead to a global enhancement of 10 % in phosphate content in 2080–2099 in comparison to 1980–1999."*
*Is it 6 % or 10 %?*

The difference between CTRL and HIS/A2 indicates an increase in phosphate concentrations of 6% ("Phosphate content in the entire Mediterranean has increased in our simulation by 6 % over the 21st century, as determined by the difference between CTRL and HIS/A2 simulations between 1980-1999 and 2080—2099"). This enhancement is the result of both climate and external nutrient fluxes changes (we added this precision in the text).

The effects of climate change alone (difference between HIS/A2 and CTRL_RG) yield a more important increase in phosphate concentrations. This shows that future changes in climate and external nutrient supply may have opposite effects on nutrient concentrations in the Mediterranean.

We modified this paragraph lines (340-355): "*Total phosphate content in the entire Mediterranean grew in our HIS/A2 simulation by 6 % over the 21st century, as determined by the difference between CTRL and HIS/A2 simulations between 1980-1999 and 2080–2099. The increase is larger in the eastern basin than in the western basin. In particular, there is an 8 % increase in phosphate content in the Ionian–Levantine sub–basin. Nutrient content in the HIS/A2 simulation is affected by both climate and nutrient fluxes from external sources (rivers and fluxes via the Strait of Gibraltar). The effects of changes in riverine input of phosphate are derived from the CTRL_R simulation (see also Figure 5). The difference of phosphate*

*content between CTRL and CTRL_R are substantial over the first half of the century. We observe 3 % decrease in phosphate content in the entire Mediterranean between 1980–1999 and 2030–2049 due to river input changes (difference between CTRL_R and CTRL). Changes in phosphate fluxes thought the Strait of Gibraltar seem to have limited effect on the global Mediterranean phosphate content as revealed by the small difference between the beginning and the end of simulation CTRL_RG. Conversely, climate change enhances the basin-wide phosphate content by 10 % in 2080–2099 relative to 1980–1999 (HIS/A2 minus CTRL_RG). Thus future changes in climate and external nutrient supply may have opposite effects on nutrient concentrations in the Mediterranean."*

**18) Line 384 – "P and N", you wrote phosphate and nitrate before, and now P and N. Please, stay consistent throughout the manuscript.**
We corrected to 'phosphorus' and 'nitrogen'. However, the sedimentation fluxes in PISCES are total (inorganic and organic) phosphorus and nitrogen and not just phosphate and nitrate because these fluxes are calculated from organic carbon.

**19) Line 417 – "Nutrient concentrations in the eastern part…" Replace nutrient by phosphate.**
Done

**20) Line 422 – "a large annual variability" Replace by "a large interannual variability".**
Done

**21) Line 427 – In the previous paragraph, about the western basin, some suggestions/interpretations were included. However, in this paragraph about the eastern basin, there were no suggestions to explain your results, why?**
We added a sentence about the eastern basin indicating that its phosphate concentrations are probably influenced primarily by climate change throughout the 21st century: "*These results show that the 21st century evolution of phosphate concentrations in the eastern Mediterranean is mainly driven by climate change. Figure 7 shows that average PO4 concentrations in CTRL, CTRL_R and CTRL_RG are similar for all periods in the eastern basin.*"

**22) Line 459 – "In the Mediterranean Sea, primary productivity is mainly limited by these 2 nutrients and their evolution in the future may impact the productivity of the basin." Not necessary, the sentence can be removed.**
It has now been removed.

**23) Line 461 – "showing an accumulation of nitrate in large zones of the basin," add "by the end of the century".**
Done

**24) Line 463 – "…and a small area in the southeastern Levantine" Also in the Gulf of Lion and south of Crete.**
We have now added references to these two areas mentioned by the reviewer.

**25) Line 465 – "…except near the mouth of the Nile and in the Alboran Sea." Also in the Ionian Sea, in the Tyrrhenian Sea, Algerian basin and between Crete and Cyprus.**
We have now also mentioned these other areas.

**26) Line 466 – "the N:P ratio (i.e. increase in P discharge and decrease in N discharge) in this river in our scenario." Replace by "the N:P ratio in this river in our scenario (i.e. increase in P discharge and decrease in N discharge).".**
Done

**27) Line 471 – "All the most productive zones of the beginning of the century are reduced in size and intensity by the end of the century." This statement does not convince me... Too broad and unclear...**
We have now clarified this sentence, writing as follows: "All the most productive zones at the beginning of the century are reduced in size and intensity by the end of the century. For instance, we observe a 10 to 40 % decrease in primary production in the Gulf of Lion and around the Balearic Islands, and more than a 50 % reduction in the North Adriatic basin, in the Aegean Sea and in the eastern Levantine basin around Cyprus."

**28) Line 481 – You already mentioned this area around Cyprus in the paragraph before.**
We integrated this paragraph to the previous one. See lines 473 to 479: *"For instance, there is a 10 to 40 % decrease in primary production in the Gulf of Lion and around the Balearic Islands, more than 50 % reduction in the North Adriatic basin, in the Aegean Sea and in the eastern Levantine basin around Cyprus. There is also a reduction in primary productivity from 40-50 gCm$^{-2}$ year$^{-1}$ to 20-30 gC m$^{-2}$ year$^{-1}$ around Cyprus. These mesoscale changes may be linked with changes in local circulation (e.g., mesoscale eddies). These observations show that the evolution of the Mediterranean biogeochemistry is influenced by both meso- and large-scale circulations patterns."*

**29) Line 499 – "South Adriatic", North Adriatic?**
We meant more the north eastern Ionian basin. We also observe more P limitation in the coastal South Adriatic.

**30) Line 501 – "One specificity of Mediterranean biology is that most planktonic productivity occurs below the surface at a depth called the deep chlorophyll maximum (DCM). Hence, most of the chlorophyll concentration is not visible by satellites (Moutin et al., 2012)." I do not think that these sentences are necessary, and I do not think that this satellite limitation is discussed in Moutin et al., 2012.**
We removed these sentences.

**31) Line 508 – "as the South Ionian and the Tyrrhenian basin." Not visible in these areas, mainly visible in the south of Crete.**
We changed the sentence to refer to the North Ionian and the South of Crete.

**32) Line 511-524 – statements in this paragraph are not supported by the results. For example,**
☐ **"but surface concentration is enhanced by about 25 10$_{-9}$ g L$_{-1}$", that means 0.025 µg L$_{-1}$, which represents almost nothing.** We agree this is a very small variation and make no conclusions about this value. We changed the sentence to "*but surface concentration is decreased by about 25 10$^{-9}$ g L$^{-1}$, which is negligible.*"

☐ **"This shows that local variability in the Mediterranean circulation and biogeochemistry is important." A general statement not supported by results.** We deleted this sentence
☐ **"the average chlorophyll concentration is reduced by almost 50 %", looking at figure 14a, there is not a 50 % decrease in the chlorophyll concentration.**

We thank the reviewer for this remark and deleted this sentence that indeed was referring to previous results that have been corrected since. We rearranged some sentences in this paragraph in an effort to make it clearer.

See lines 510 to 523: "*Figure 14 shows the average vertical profiles of chl-a at the DYFAMED station (43.25° N, 7.52° E) and the average profiles for the western and eastern basins for the 1980–1999 and 2080–2099 periods. The subsurface chl-a maximum persists through to the end of the century. At the DYFAMED station, the average DCM depth remains unchanged, while the surface chl-a concentration is decreased by about 25 ng L$^{-1}$, which is negligible. Thus the average chl-a profile at DYFAMED changes little throughout the simulation. However, at that station there is approximately 40 % variability in the chl-a concentration profile and depth of DCM for some month (not shown). In the western basin, the subsurface maxima at present and in the future are located at the same depth (100–120 m), but the average chl-a concentration in the DCM increases by about 15 to 20 ng L$^{-1}$ during the simulation. However, where there are small changes in the average chl-a profiles in the 520 western basin, those are often accompanied by greater local changes in the depth and intensity of the DCM (Figure 13). In the eastern basin, subsurface chl-a concentration is reduced by about 50 ng L$^{-1}$ and the subsurface chl-a maximum deepens from 100–120 m to below 150 m.*"

**33) Lines 527, 532, 533 – all percentage quoted in the text are different in Table 4. Please, double-check the values.**
We thank the reviewer. We have now corrected these values that were the surface difference values and not the integrated values.

**34) Line 546 and 556 – lack of quantitative estimates.**
We have no added some numeric values.

See for instance lines 448-450: "*In HIS/A2, the biomass of both phytoplankton classes declines during the simulation (-0.01 mmolC m$^{-3}$ for nanophytoplankton and - 550 0.03 to -0.04 mmolC m$^{-3}$ for diatoms).*"
And lines 558-560: "*In HIS/A2 in all basins there is a decrease in microzooplankton concentration during 1980–2000 (from 0.165 to 0.114 mmol m$^{-3}$), after which it remains stable and consistently below the CTRL values until the end of the simulation in 2100.*"

**35) Section 4.1 Biogeochemical forcings – Not a good title. This section mainly discussed the influences of external sources of nutrients**
We have rearranged the discussion. Section 4.1 is now titled "Sources of uncertainties" and subdivided in 3 subsections: 4.1.1 Climate change scenario, 4.1.2 Uncertainties from the PISCES model and 4.1.3 External nutrient sources.

**36) Line 617 – "these regions" replace by "this region".** Done

**37) Line 624 – "2" replace by "two"** Done

**38) Line 701 – close parenthesis.** Done

**39) Figure A3a – It is impossible to read the colorbar.**
We have changed the colorbar of this figure.

*This revised manuscript is improved from the previous version and the authors have taken on board the comments by the reviewers. There is now quantitative arguments throughout the manuscript and I accept what the authors say regarding the limitations of the model and that this should not stop the manuscript from being published as it is the best model and model inputs that they have available. They have clarified what is in the model and now acknowledge that organic P and N may also affect the budget. In addition they have created a nice figure summarising the inputs and outputs of phosphate and nitrate to the basin. However, I have still found this manuscript relatively difficult to read in places with the keys points lost within the text. There are now places with extremely long paragraphs (i.e lines 166-195, 331-362, 589-620, 696-731) and in these paragraphs it becomes unclear which key point the author wanted the reader to get from the it. I think both the results and discussion sections can still be improved and there should be increased emphasis on the impact of the results in the discussion. I feel that the results the authors present are important and should be published but at present their impact does not come across strongly enough in the discussion.*

We thank the reviewer for their comments. In this revised manuscript, we tried to improve the presentation of the results by separating section 3.3.5 in 2 different paragraphs: one for phosphate, and one for nitrate. In each of these paragraphs, we first describe the results in the western basin (with the order: surface, intermediate and deep waters) and then in the eastern basin.

We reorganized the discussion and in particular emphasize the implications and impact of our results in section 4.2.

*Lines 166-195: There is now a very long paragraph in the method section discussing the riverine input. The authors have addressed my concerns in this paragraph but have additionally added further sentences which in my opinion become too detailed in regards to differences between the different inputs (Ludwig vs Adloff etc ) in different models which I find confusing. In addition, there is now a lot of repetition between this section and the first paragraph of the discussion. Although the authors obviously do need to acknowledge the potential errors in their results I think these sections can be reduced.*

We moved the part of this paragraph describing the uncertainties of the river nutrient fluxes to the discussion as these arguments were repeated.

Lines 665 to 680: "*For the riverine nutrient inputs, scenarios from the MEA report are based on different assumptions from the IPCC SRES scenarios used to compute freshwater runoff in the HIS/A2 simulation. Freshwater discharge from Ludwig et al. (2010) is based on the SESAME model reconstruction and differs from freshwater runoff in the ARPEGE–Climate model used to force our physical model. This may lead to incoherences between water and nutrient discharges, but the nutrient discharges from Ludwig et al. (2010) are the only ones that are available. Furthermore, the SESAME model is not coupled with NEMO/PISCES. Associated discrepancies and the uncertainties linked with the use of inconsistent scenarios in our simulation should be addressed by developing a more integrated modelling framework to study the impacts of climate change on the Mediterranean Sea biogeochemistry. As there is no consensus nor validated scenario for nutrient fluxes from riverine runoff in the Mediterranean, we chose to use one scenario from Ludwig et al. (2010). This scenario has the advantage of being derived from a coherent modeling framework. However, the Ludwig et al. (2010) nutrient discharge transient scenario does not represent the interannual variability of nutrient runoff from rivers. Moreover, according to these authors, the socio–economic decisions made in the 21st century will influence nitrate and phosphate discharge*

*over the Mediterranean. It is difficult to forecast these decisions and the resulting changes in nutrient fluxes are uncertain."*

**Section 3.3: I still find this section hard to follow. The authors have now added quantitative metrics in this section but they are generally put in brackets rather than integrated into the text which is making it awkward to read. In addition although I appreciate that the authors have tried to change this section, the authors still explain trends such as decrease in phosphorus content by decrease in riverine inputs before presenting the river inputs and therefore are having to repeat things. I suggest putting the results on P and N content after discussing the other terms in the budget.**

We moved the 3.1 section after the description of nutrient fluxes in and out of the basin. We also tried to rephrase parts of the section to include the quantitative metrics in the text.

**Discussion: The discussion feels dominated by statements about the limitations/uncertainties of the model with weak statements on the impact and interpretation of results. I think it would improve the manuscript if there was a better integration of the literature with the authors own arguments and conclusions from this study within the discussion. In this revised manuscript the authors have added additional comparisons with other literature which is important and I think was needed, but it currently reads as a list (Lines 696-731). The authors are trying to justify why their results are different than what is in the literature rather than using the literature to put their results into context and strengthen their arguments. For example, in lines 727-731 rather than explaining why you can not observe the effects of temperature on nutrient recycling within this model, you can maybe say how an increase in nutrient cycling due to warmer temperatures may strengthen or weaken your conclusions.**

We rearranged and reformulated the discussion following recommendations from another reviewer. We have separated the discussion on the uncertainties of the study in section 4.1. We try to emphasize the originality and the impact of our results in section 4.2.

In particular, our results show that the effects of climate change, riverine input and nutrient fluxes through the Strait of Gibraltar may have synergistic or antagonistic effects depending on the depth and the region.

See lines 737 to 744: "*Results from our transient simulations show that nutrient concentrations may evolve differently depending on the region and the depth in response to climate change and external nutrient inputs. In the surface western Mediterranean, the effects of climate change and enhanced nutrient fluxes via Gibraltar both concur to the increase in nutrient concentrations (Figures 7a and 8a). In the surface eastern basin, river fluxes of nitrate and stratification have opposing effects on nitrate concentrations whereas phosphate concentrations are mainly driven by climate change effects (Figures 7d and 8d). The difference between HIS/A2 and CTRL_RG phosphate concentrations in the intermediate and deep layers (Figures 7b, 7e, 7c, 7f) indicates that variations of phosphate concentrations during the 21st century are primarily driven by climate change while nitrate concentration is equally sensitive to changes in biogeochemical forcings (Figures 8b, 8e, 8c, 8f)."*

We also report that the changes in nutrient concentration have consequences on the entire ecosystem.
See lines 789 – 794: "*Chust et al. (2014) have shown that regional seas and in particular the Aegean and Adriatic were sensitive to trophic amplification. Our results appear to agree, showing signs of trophic amplification (see Figures 15 and 16). Assessing the sensitivity of the Mediterranean to trophic amplification would require more simulations focused on the evolution of Mediterranean planktonic biomass under different climate change scenarios."*

In general, we tried to use the literature to strengthen our conclusions instead of simply comparing our results. See for instance lines 661-664: "*Herrmann et al. (2014) simulated an increase in chl-a*

concentration associated with climate change effects in a small region of the north western basin by the end of the century. Thus, the separate effects of climate change and external nutrient inputs may have synergetistic effects on the evolution of the western Mediterranean chl-a."

Lines 673-675:" *Lazzari et al. (2014) also conclude that the river mouth regions are highly sensitive because the Mediterranean Sea is influenced by external nutrient inputs. The choice of river runoff scenario will likely influence the evolution of nutrient concentrations and the biogeochemistry in many coastal regions such as the Adriatic Sea (see also Spillman et al., 2007)."*

*Along these lines the discussion on N and P limitation (Lines 713-726) could be a paragraph/section to itself. In this section the authors state the their results are "in contrast with previous literature on the matter" (Lines 715-716 ). However there is evidence within the literature for P and N co-limitation in the Mediterranean. Whilst generally the spring phytoplankton bloom is P limited, N and P co-limitation has been observed, especially during the stratification period (Thingstad et al., 2005; Tanaka et al., 2011) and there is some evidence of the spring phytoplankton bloom being N and P co-limited in the Western Mediterranean (Pasqueron de Fommmervault et al., 2015) or even N limited (Marty et al. 2002). In addition N limitation has been predicted in the Alboran Gyre as well (Ramirez et al., 2005; Lazzari et al., 2016). What time period do you calculate the N and P limitation for (i.e annual mean, spring bloom etc)? This may also affect what you are predicting compared to the literature*

We separated this paragraph from the rest of the text (lines 782-791) and modified in an effort to focus it more on the nutrient co-limitation:

"*The modifications of chl-a production and plankton biomass are linked to changes in nutrient limitation (Figure 12). Finding no clear definition of nutrient co-limitation, we consider that N and P are co-limiting when the difference in limitation factors is less than 1 %. This definition of nutrient co-limitation applies well to the Mediterranean case because of its very low nutrient concentrations. Our results are confirmed by some studies (Thingstad et al 2005, Tanaka et al. 2011). However, our nutrient limitations are calculated from 20–years average nutrient concentrations and nutrient limitation may vary greatly during the seasonal cycle (Marty et al. 2002, Diaz et al 2001)."*

*Finally, at other reviewers suggestion the authors have now included a scenario on atmospheric deposition but only present this within the discussion (Lines 621-638). I feel it should be fully integrated into the text (i.e in the methods and results section) rather than tagged onto the discussion. It does provide some important insite on the effect of climate change despite only considering a climatology of atmospheric inputs rather than potential future ones. The authors could further hypothesise what potential future changes in atmospheric deposition may have on the results in the discussion based upon regional projections of atmospheric inputs into the future (i.e Lambarque et al., 2013). Whilst I appreciate they can't actually run a scenario they may be able to comment on whether it is likely to enhance/dampen the trend they see.*

We initially decided to keep the description of these simulations in the discussion as they are not the main focus of the study, and the aerosol deposition forcings are not evolving during the 21$^{st}$ century.

As suggested, we included these simulations in the main text. In the methods section see lines 175 to 183: "*There is, to our knowledge, no transient scenario for the evolution of atmospheric deposition over the Mediterranean Sea. However, in order to evaluate the potential effects of aerosol deposition on the future Mediterranean Sea, we used deposition fields of total nitrogen deposition ($NO_3$ + $NH_4$) from the global model LMDz-INCA (Hauglustaine et al 2014) and phosphate deposition from natural dust modeled with the regional model ALADIN-Climat (Nabat et al 2015) respectively (see Richon et al 2017 and references therein for the description and evaluation of the atmospheric models) The atmospheric*

*deposition fields represent present-day aerosol deposition fluxes (1997-2012 and 1980-2012 for total nitrogen and dust deposition respectively) that are repeated over the 1980-2099 simulation period."*
And lines 211 to 216: *"We made two supplementary simulations, one with total nitrogen deposition (HIS/A2_N) and another with total nitrogen and natural dust deposition (HIS/A2_NALADIN). These simulations include climate change and nutrient fluxes from rivers and via the Strait of Gibraltar that follow the scenario conditions. The results from these simulations should be considered as exploratory. Nonetheless, they provide insight into the potential effects of future aerosol deposition."*

We added a result section lines 595 to 604: *"Effects of aerosol deposition on surface primary productivity"*
*"Figure 17 shows the relative effects of total nitrogen and natural dust deposition on surface primary production in 1980-1999 and 2080-2099. As shown in Richon et al 2017,* dust deposition enhances surface primary productivity in the southern part of the basin in 1980-1999 whereas nitrogen deposition enhances primary productivity in the northern part of the basin. *As our HIS/A2 simulation shows a decrease in surface $PO_4$ concentrations, thus accentuating phosphate limitation over the Mediterranean basin by the end of the 21st century, the relative impact of phosphate deposition from dust would be enhanced in the 2080–2099 period relative to the 1980–1999 period. Conversely, nitrogen atmospheric deposition has very little effect on Mediterranean primary production at the end of the simulation period because most of the basin is not N-limited."*

In the discussion section, we tried to emphasize the fact that even if phosphate deposition seems to relieve a part of the phosphorus limitations at the end of the century, the Mediterranean is still mainly P-limited in 2100. However, many aerosol sources such as volcanoes, anthropogenic sources and organic phosphorus and nitrogen are not included in our sources. Therefore, it is important to develop models and measurements of these aerosols as they may have important impacts on the Mediterranean surface biogeochemistry.
See lines 706 to 729:
*"The biogeochemistry of the Mediterranean is significantly influenced by aerosol deposition (e.g. Krom et al., 2010; Dulac et al., 1989; Richon et al., 2018, 2017; Guieu et al., 2014). The future evolution of the multiple aerosol sources surrounding the Mediterranean will likely influence the response of the Mediterranean to climate change. Results from the HIS/A2_NALADIN simulation show that enhanced phosphate fluxes from aerosols may limit the surface decrease of phosphate concentrations and limit phosphorus limitation. However, in the HIS/A2_NALADIN simulation, the surface Mediterranean is still P-limited in most of the Mediterranean because the atmospheric nutrient fluxes are low in comparison to riverine nutrient fluxes from rivers and the nutrient flux through the Strait of Gibraltar (see Richon et al., 2017). Therefore, it appears unlikely that changes in aerosol deposition from natural dust would greatly influence future Mediterranean biogeochemistry. However, there are multiple sources of aerosols that are not included in atmospheric models, e.g., anthropogenic, volcanic and volatile organic compounds (e.g. Wang et al., 2014; Kanakidou et al., 2016). Their combined influence could perhaps constitute an important nutrient flux to the Mediterranean, thus altering the evolution of its biogeochemistry. Moreover, aerosols affect radiative forcing over the Mediterranean and may impact the climate conditions (Nabat et al., 2015b). Thus, efforts should be made to accurately represent this nutrient source in Mediterranean models to assess the effect on Mediterranean Sea biogeochemistry with regards to climate change. Our results show that the state of the Mediterranean biogeochemistry at the end of the 21st century is the result of the combined evolutions of both climate and external nutrient fluxes. Therefore, it is very difficult to predict the future evolution of the Mediterranean based on the evolution of one of these components only. This is why it is important, in the case of semi–enclosed basins, to produce reliable estimates of the evolution of all the components influencing the biogeochemistry."*

*Figure 18: I suggest reversing the input and output arrows through the Strait of Gibraltar so that they are the same as the actual water flow. Currently it is suggesting an estuarine flow rather than anti estuarine.*

We thank the reviewer for this suggestion. We changed the arrows.

[revised manuscript text omitted]